# Beyond Continuity: Simulation-free Reconstruction of Discrete Branching Dynamics from Single-cell Snapshots

**Junda Ying** [1]   **Yuxuan Wang** [2]   **Bowen Yang** [3]   **Peijie Zhou** [4 5 6 7]   **Lei Zhang** [1 3 4 5]

## Abstract

Inferring cellular trajectories from destructive snapshots is complicated by the challenges of stochasticity and non-conservative mass dynamics such as cell proliferation and apoptosis. Existing unbalanced Optimal Transport (OT) methods treat mass as a continuous fluid, performing inference at the population level. However, this macroscopic view often fails to capture the discrete, jump-like nature of birth-death events at single-cell resolution, which is essential for understanding lineage branching and fate decisions. We present **Unbalanced Schrödinger Bridge (USB)**, a simulation-free framework for learning underlying dynamics that effectively integrates both stochastic and unbalanced effects which also models the discrete, jump-like birth–death dynamics at single-cell resolution. Theoretically, USB provides a tractable solution to the Branching Schrödinger Bridge (BSB) problem, offering a rigorous microscopic interpretation where individual cells undergo both Brownian motion and discrete birth-death jumps. Technically, the method implements an efficient solver by introducing a simulation-free training objective that effectively scales to high-dimensional omics data. Empirically, we demonstrate on both simulated and real-world datasets that USB not only achieves trajectory reconstruction performance better than or comparable to deterministic baselines but also uniquely enables realistic discrete simulation of birth-death dynamics at single-cell resolution.

---

[1]Beijing International Center for Mathematical Research, Peking University [2]Center for Data Science, Peking University [3]School of Mathematical Sciences, Peking University [4]Center for Quantitative Biology, Peking University [5]Center for Machine Learning Research, Peking University [6]National Engineering Laboratory for Big Data Analysis and Applications, Beijing [7]AI for Science Institute, Beijing. Correspondence to: Lei Zhang <zhangl@math.pku.edu.cn>, Peijie Zhou <pjzhou@pku.edu.cn>.

*Proceedings of the $43^{rd}$ International Conference on Machine Learning*, Seoul, South Korea. PMLR 306, 2026. Copyright 2026 by the author(s).

## 1. Introduction

Recovering continuous underlying dynamics from snapshot observations is a critical challenge in single-cell biology. Due to the destructive nature of single-cell sequencing protocols, methods must infer temporal evolution without access to longitudinal trajectories of individual cells. While Optimal Transport (OT) (Kantorovich, 1958; Benamou & Brenier, 2000) has become a standard framework for this task, traditional OT seeks deterministic ODE flows between balanced probability distributions. Consequently, it fails to capture the inherent stochasticity and unbalanced nature of cellular processes driven by high biological noise and cell proliferation/apoptosis.

The Schrödinger Bridge (SB) problem (Schrödinger, 1932; Léonard, 2014) is employed to model this stochasticity. It models cellular dynamics as SDEs connecting two balanced probability distributions, rather than ODEs, thereby explicitly introducing stochasticity. Parallel to the need for stochastic modeling, addressing varying cell numbers is essential (Sha et al., 2024). Both standard OT and SB assume mass conservation between time points, a condition rarely met in proliferating biological systems. To address this issue, researchers have proposed various extensions of OT for two unbalanced measures (Eyring et al., 2024; Wang et al., 2025; Peng et al., 2026). While effective at handling unbalanced population between snapshots, these standard OT-based methods often lack the capability to naturally incorporate stochasticity.

To simultaneously model stochasticity and unbalanced effects, approaches based on dynamic Regularized Unbalanced Optimal Transport (RUOT) (Zhang et al., 2025) or Schrödinger Bridge with coffin states (Pariset et al., 2023) have been developed. While theoretically capable of modeling both aspects, these methods rely on computationally costly NeuralODE (Chen et al., 2018) or Iterative Proportional Fitting (IPF), rendering them computationally challenging for large data.

To scale dynamical modeling to large data, recent advances have successfully introduced simulation-free paradigms–flow matching (Lipman et al., 2023)–for solving standard OT problems, along with their stochastic or unbalanced ex-

tensions (Tong et al., 2024a;b; Peng et al., 2026). These methods have rendered dynamical modeling efficient and scalable. However, there currently exists no unified framework that is simultaneously stochastic, unbalanced, and simulation-free.

Furthermore, we highlight a critical limitation in current methods modeling unbalanced mass: they uniformly treat mass as a continuously varying quantity. While natural within frameworks like flow matching or NeuralODE, this assumption overlooks a fundamental characteristic of cellular proliferation and apoptosis: cell numbers are discrete values that change through jumps.

To address these challenges, we present Unbalanced Schrödinger Bridge (USB), a simulation-free framework for learning underlying dynamics that accounts for both stochastic and unbalanced effects, and explicitly permits discrete birth-death dynamic simulations. Our contributions are summarized as follows:

- We propose USB, a novel framework that unifies the stochasticity with the unbalance through the lens of Branching Schrödinger Bridge problem (BSB).

- We address the computational bottleneck of stochastic unbalanced modeling by developing a general simulation-free unbalanced score matching framework.

- We demonstrate that USB consistently recovers ground-truth dynamics in complex landscapes and provides discrete, single-cell resolution birth-death simulations of proliferation and apoptosis.

## 2. Related Works

**SB and Unbalanced Extensions.** Many numerical algorithms have been developed to solve Schrödinger Bridge (Schrödinger, 1932) problem (SB) (De Bortoli et al., 2021; Shi et al., 2023; Bunne et al., 2023a;b; Kim et al., 2025; Wang et al., 2021; Tong et al., 2024b; Peluchetti, 2025; Somnath et al., 2023; Gushchin et al., 2024; Garg et al., 2024; De Bortoli et al., 2024; Shen et al., 2025; Noble et al., 2023). Common paradigms include converting SB into a regularized optimal transport (ROT) problem (Föllmer, 1988; Léonard, 2014; Pavon et al., 2021; Tong et al., 2024b), or stochastic optimal control (SOC) problem (Chen et al., 2016; 2022; Liu et al., 2024; Tang et al., 2025). Recently, unbalanced extensions based on coffin state (Chen et al., 2025; Pariset et al., 2023) or branching process (Baradat & Lavenant, 2021) have also been proposed, but **efficient simulation-free solvers are still lacking**. USB is a **simulation-free** solver based on the latter.

**Flow Matching and Score Matching.** Flow matching (Lipman et al., 2023; Liu et al., 2023; Pooladian et al., 2023;

Albergo & Vanden-Eijnden, 2023) is a simulation-free generative framework for learning deterministic ODE flows, and it can be coupled with optimal transport (OT) for efficiently learning dynamics (Tong et al., 2024a; Klein et al., 2024; Rohbeck et al., 2025). To accommodate more complex dynamics, prior works have introduced score-based stochastic extensions (Sohl-Dickstein et al., 2015; Song & Ermon, 2019; 2020; Song et al., 2021; Ho et al., 2020; Winkler et al., 2023; Dhariwal & Nichol, 2021; Tong et al., 2024b; Tang et al., 2025; Lee et al., 2025), unbalanced extensions handling unbalanced marginals (Eyring et al., 2024; Cao et al., 2025; Corso et al., 2025; Wang et al., 2025; Peng et al., 2026), and other generalizations (Kapuśniak et al., 2024; Zhang et al., 2024b; Atanackovic et al., 2025; Petrović et al., 2025). However, existing matching-type methods **do not jointly model stochasticity and unbalanced marginals**. USB introduces an **unbalanced score matching framework** that bridges this gap.

**Single-cell Trajectory Inference in Unbalance and Stochastic Setting.** Single-cell trajectory inference in the multi-time points setting has been tackled with NeuralODEs (Tong et al., 2020; Huguet et al., 2022; Zhang et al., 2024a; Sha et al., 2024; Gu et al., 2025; Choi & Choi, 2024) and optimal transport (OT) (Schiebinger et al., 2019; Klein et al., 2025; Halmos et al., 2025; Banerjee et al., 2025), together with their stochastic (Neklyudov et al., 2023; 2024; Albergo et al., 2025; Zhu et al., 2024; Maddu et al., 2025; Yeo et al., 2021; Chizat et al., 2022; Lavenant et al., 2024; Shi et al., 2023; Koshizuka & Sato, 2023; Bunne et al., 2023a; Chen et al., 2022; Jiang & Wan, 2024; Zhang et al., 2025; Sun et al., 2025) and unbalanced variants (Neklyudov et al., 2023; 2024; Eyring et al., 2024; Wang et al., 2025; Peng et al., 2026). Some also used branching SDEs (Ventre et al., 2024), but require lineage trees information. Flow matching can be applied to these approaches to improve scalability and stability (Tong et al., 2024a; Rohbeck et al., 2025; Klein et al., 2024; Tong et al., 2024b; Lee et al., 2025; Kapuśniak et al., 2024; Atanackovic et al., 2025). However, the field still lacks **a simulation-free matching framework that jointly models stochasticity, unbalance and the discrete birth–death dynamics which need no priors**.

## 3. Preliminaries

**Setup.** Inspired by single-cell dynamics inference, consider two unnormalized measures $\mu_0(\boldsymbol{x})$, $\mu_1(\boldsymbol{x})$, also denoted as $\mu_0, \mu_1$, at time $t = 0$ and $t = 1$ respectively defined over $\mathcal{X} \subseteq \mathbb{R}^d$. Let $\mathcal{M}_+(\mathcal{X})$ represent the set of all absolutely continuous finite measures supported on $\mathcal{X}$. The total mass of $\mu_0$ and $\mu_1$ may be different. We aim to learn the **most likely stochastic process bridging $\mu_0$ and $\mu_1$**.

To characterize changes in mass, two modeling paradigms are commonly used:

- The first employs weighted particles, assigning each particle a continuously varying mass weight and describing the temporal evolution of the measure with a Fokker–Planck equation that includes a source term; we refer to this as a **continuous measure flow**.

- The second employs **branching processes**, using particle division and death to simulate jump-like changes in mass, with the particle count modeling discrete mass. For single-cell birth–death dynamics, it offers superior biological interpretability.

**Continuous Measure Flows.** Taking both stochastic and unbalanced effect into account, consider a time-dependent measure path $\rho : \mathbb{R}^d \times [0,1] \to \mathbb{R}_+$, a time dependent vector field $\boldsymbol{u} : \mathbb{R}^d \times [0,1] \to \mathbb{R}^d$, and a time dependent growth rate $g : \mathbb{R}^d \times [0,1] \to \mathbb{R}$. A measure can be represented by a population of weighted particles $\{(\boldsymbol{x}_i, m_i)\}$. $\boldsymbol{x}$ stands for the position, and $m$ stands for the weight or say mass. Starting from the initial measure $\rho_0$, the measure path is generated by a SDE

$$\begin{cases} \mathrm{d}\boldsymbol{x}_t = \boldsymbol{u}_t(\boldsymbol{x}_t)\mathrm{d}t + \nu\mathrm{d}\boldsymbol{W}_t \\ \mathrm{d}\ln m_t = g_t(\boldsymbol{x}_t) \end{cases} \quad (1)$$

where $\boldsymbol{W}_t$ is the standard Brownian motion in $\mathbb{R}^d$, and $\nu \in \mathbb{R}$ is the diffusion parameter. The measure path satisfies the Fokker-Planck equation with source term

$$\partial_t \rho_t(\boldsymbol{x}) + \nabla_{\boldsymbol{x}} \cdot (\boldsymbol{u}_t(\boldsymbol{x})\rho_t(\boldsymbol{x})) = \frac{\nu^2}{2}\Delta_{\boldsymbol{x}}\rho_t(\boldsymbol{x}) + g_t(\boldsymbol{x})\rho_t(\boldsymbol{x}) \quad (2)$$

**Branching Brownian Motion.** Branching Brownian motion (BBM) is a prototypical example of branching processes. It can be described by a doublet $(\nu, \boldsymbol{q})$ where $\nu \in \mathbb{R}$ is the **diffusion parameter**, $\boldsymbol{q} \in \mathcal{M}_+(\mathbb{N})$ is an unnormalized measure supported on natural numbers with finite total measure called **branching mechanism**. The total measure of branching mechanism $\lambda = \sum_{k \in \mathbb{N}} \boldsymbol{q}_k$ is called **branching rate**, and the normalized branching mechanism $\boldsymbol{p} = \boldsymbol{q}/\lambda$ is called **offspring distribution**. Under BBM, a particle is equipped with an exponential clock of rate $\lambda$. It evolves according to Brownian motion with diffusion parameter $\nu$ until time $\tau \sim Exp(\lambda)$. At time $\tau$, the particle branches into $k \sim \boldsymbol{p}$ particles. ($k = 0$ means that the particle vanishes). After branching, the $k$ new particles undergo BBM independently. Restricting particles to split into two or die makes BBM well aligned with the microscopic dynamics of cell division and apoptosis. Its allowance for mass jumps also makes it a natural choice for single-cell modeling.

### 3.1. Dynamic OT via Flow Matching

Without stochasticity and unbalanced mass, dynamic OT provides a principled framework for interpolating

between two probability measures by $\mathcal{W}_2$ geodesic $\{\rho_t\}_{t \in [0,1]}$ (Benamou & Brenier, 2000). Particles are transported by the ODE flow generated by $\boldsymbol{u}_t$ without growth. The equation (2) reduces to continuity equation $\partial_t \rho_t + \nabla_{\boldsymbol{x}} \cdot (\boldsymbol{u}_t \rho_t) = 0$. By introducing flow matching (Lipman et al., 2023), recent work (Tong et al., 2024a) solves the dynamic OT in a simulation-free manner. It parameterized a neural network $\boldsymbol{v}_\theta(t, \boldsymbol{x})$ to approximate the velocity field by minimizing a regression loss

$$\mathcal{L}_{\mathrm{FM}}(\theta) = \mathbb{E}_{t, \boldsymbol{x} \sim \rho_t}\left[\|\boldsymbol{v}_\theta(t, \boldsymbol{x}) - \boldsymbol{u}_t(\boldsymbol{x})\|_2^2\right].$$

One key observation is that though the marginal probability path $\{\rho_t\}$ is intractable, one can introduce tractable conditional paths $\rho_t(\cdot|\boldsymbol{z})$ and conditional velocity field $\boldsymbol{u}_t(\cdot|\boldsymbol{z})$ w.r.t some conditioning variable $\boldsymbol{z}$. The minimization problem is equivalent to minimizing a conditional version of the loss

$$\mathcal{L}_{\mathrm{CFM}}(\theta) = \mathbb{E}_{t, \boldsymbol{z}, \boldsymbol{x} \sim \rho_t(\cdot|\boldsymbol{z})}\left[\|\boldsymbol{v}_\theta(t, \boldsymbol{x}) - \boldsymbol{u}_t(\boldsymbol{x}|\boldsymbol{z})\|_2^2\right].$$

One can choose $\boldsymbol{z} = (\boldsymbol{x}_0, \boldsymbol{x}_1)$ drawn from the static OT (Kantorovich, 1958) coupling $\gamma(\boldsymbol{x}_0, \boldsymbol{x}_1)$ between $\mu_0$ and $\mu_1$, and use conditional Gaussian path $\mathcal{N}(t\boldsymbol{x}_1 + (1-t)\boldsymbol{x}_0, \nu^2 \mathbf{I})$ for determining $\boldsymbol{u}_t(\cdot|\boldsymbol{z})$. The resulting flow recovers the dynamic OT flow.

We point out that the algorithm consists of two important part – **the coupling** and **the conditional path**. One can design specific coupling and conditional path to approximate other flows instead of dynamic OT flow.

### 3.2. Unbalanced Effect Model with WFR Metric

To interpolate unbalanced source and target, previous works (Chizat et al., 2018a;b; Liero et al., 2018) defined WFR metric as

$$\mathrm{WFR}_\delta^2(\mu_0, \mu_1) =$$

$$\inf_{\rho, g, \boldsymbol{u}} \int_0^1 \int_{\mathcal{X}} \frac{1}{2}(\|\boldsymbol{u}(\boldsymbol{x}, t)\|_2^2 + \delta^2\|g(\boldsymbol{x}, t)\|_2^2)\rho_t(\boldsymbol{x})\mathrm{d}\boldsymbol{x}\mathrm{d}t$$

$$\text{s.t.} \quad \partial_t \rho + \nabla_{\boldsymbol{x}} \cdot (\rho\boldsymbol{u}) = \rho g, \ \rho_0 = \mu_0, \ \rho_1 = \mu_1, \quad (3)$$

This minimization problem is called dynamic WFR. Similar to dynamic OT, it also has a static form

$$\mathrm{WFR}_\delta^2(\mu_0, \mu_1) = 2\delta^2 \{ \inf_{(\gamma_0, \gamma_1)} \int_{\mathcal{X}^2} \left(\gamma_0(\boldsymbol{x}, \boldsymbol{y}) + \gamma_1(\boldsymbol{x}, \boldsymbol{y})\right.$$

$$\left. -2\sqrt{\gamma_0(\boldsymbol{x}, \boldsymbol{y})\gamma_1(\boldsymbol{x}, \boldsymbol{y})}\overline{\cos}(\frac{\|\boldsymbol{x} - \boldsymbol{y}\|_2}{2\delta})\right)\mathrm{d}\boldsymbol{x}\mathrm{d}\boldsymbol{y} \} \quad (4)$$

where $(\gamma_0, \gamma_1) \in \left(\mathcal{M}_+(\mathcal{X}^2)\right)^2$ satisfying marginal constraints $\int_{\mathcal{X}} \gamma_0(\boldsymbol{x}, \boldsymbol{y})\mathrm{d}\boldsymbol{y} = \mu_0(\boldsymbol{x}), \int_{\mathcal{X}} \gamma_1(\boldsymbol{x}, \boldsymbol{y})\mathrm{d}\boldsymbol{x} = \mu_1(\boldsymbol{y})$ is called the **semi-coupling**. It is an unbalanced extension of the OT coupling. For fixed pair $(\boldsymbol{x}, \boldsymbol{y})$, $\gamma_0(\boldsymbol{x}, \boldsymbol{y})$ represents the mass at the initial time sent from $\boldsymbol{x}$, while $\gamma_1(\boldsymbol{x}, \boldsymbol{y})$

represents the corresponding mass at the final time received by $\boldsymbol{y}$. $\overline{\cos}(x) = \cos(\min\{x, \frac{\pi}{2}\})$.

Based on these results, (Peng et al., 2026) designed a flow matching scheme to solve the problem above. They use static WFR coupling as the coupling and travelling Dirac as the conditional path. The resulting flow recovers the WFR geodesic between the source and target.

### 3.3. Stochastic Effect Model with Schrodinger Bridge.

To allow stochastic dynamics bridging two probability measures, (Schrödinger, 1932) proposed the Schrödinger bridge problem. It asks to find a most likely stochastic process $\mathbb{P}^\star$ bridging normalized $\mu_0$ and $\mu_1$ w.r.t a reference stochastic process $\mathbb{Q}$.

$$\mathbb{P}^\star = \underset{\mathbb{P}:p_0=\mu_0,p_1=\mu_1}{\arg\min} \; \mathrm{KL}(\mathbb{P}\|\mathbb{Q}) \qquad (5)$$

where $\mathbb{P}$ is a stochastic process with marginals denoted as $p_t$. A usual choice for $\mathbb{Q}$ is $\nu\mathbb{W}$, i.e. the standard Brownian motion with diffusion parameter $\nu$. The resulting Schrödinger bridge is known as diffusion Schrödinger bridge (DSB) (De Bortoli et al., 2021; Bunne et al., 2023a; Shi et al., 2023).

Following (Föllmer, 1988; Léonard, 2014), one can disintegrate the KL-divergence into two part.

$$\mathrm{KL}(\mathbb{P}\|\mathbb{Q}) = \mathrm{KL}(\mathbb{P}_{01}\|\mathbb{Q}_{01}) + \int \mathrm{KL}(\mathbb{P}^{\boldsymbol{xy}}\|\mathbb{Q}^{\boldsymbol{xy}})\mathbb{P}_{01}(\mathrm{d}\boldsymbol{x}, \mathrm{d}\boldsymbol{y})$$

$$(6)$$

where $\mathbb{P}_{01}$, $\mathbb{Q}_{01}$ are the marginal of $\mathbb{P}$, $\mathbb{Q}$ at time 0, 1, and $\mathbb{P}^{\boldsymbol{xy}}$, $\mathbb{Q}^{\boldsymbol{xy}}$ are the conditional process of $\mathbb{P}$, $\mathbb{Q}$ given start point $\boldsymbol{x}$ and end point $\boldsymbol{y}$. The second term measured the difference of conditional path. It can be minimized to 0 by choosing $\mathbb{P}^{\boldsymbol{xy}} = \mathbb{Q}^{\boldsymbol{xy}}$. Thus, it is sufficient to only minimize the first term

$$\mathbb{P}^\star = \underset{p_0=\mu_0,p_1=\mu_1,}{\arg\min} \; \mathrm{KL}(\mathbb{P}_{01}\|\mathbb{Q}_{01}) \qquad (7)$$

It is also known as the static Schrödinger bridge. When $\mathbb{Q}$ is $\nu\mathbb{W}$ with the same source measure $\mu_0$, it reduces to a regularized OT (ROT) problem. Utilizing these results, (Tong et al., 2024b) proposed SF$^2$M, a simulation-free framework for solving diffusion Schrödinger bridge. It uses ROT coupling as the coupling, and uses $(\nu\mathbb{W})^{\boldsymbol{xy}}$ (which is a Brownian bridge) as conditional path to recover the Schrödinger bridge between $\mu_0$ and $\mu_1$.

### 3.4. Unbalanced Schrödinger Bridge

To take both stochastic effect and unbalanced effect into account, (Baradat & Lavenant, 2021) proposed an unbalanced formulation of Schrödinger bridge. They **replace the reference process with branching Brownian motion**

**(BBM)** to allow growth. The BSB problem is then defined as

$$\mathbb{P}^\star = \underset{\mathbb{P}:p_0=\mu_0,p_1=\mu_1}{\arg\min} \; \mathrm{KL}(\mathbb{P}\|\mathbb{Q}) \qquad (8)$$

where $\mu_0$ and $\mu_1$ are unnormalized measures, and $\mathbb{Q}$ is a BBM.

**The evolution equation of BBM.** According to (Baradat & Lavenant, 2021), the evolution equation of BBM is

$$\partial_t\rho_t = \frac{\nu^2}{2}\Delta_{\boldsymbol{x}}\rho_t + r\rho_t \qquad (9)$$

where $r = \sum_{k\neq 1}(k-1)\boldsymbol{q}_k$. In a weak sense, it is equivalent to evolve according to Brownian motion with diffusion parameter $\nu$ in position, and to grow exponentially in mass.

**The relation to RUOT.** The main result of (Baradat & Lavenant, 2021) is that the BSB problem (8) is ill-posed, and the RUOT problem

$$\mathrm{RUOT}(\mu_0, \mu_1) =$$

$$\inf_{\rho,g,\boldsymbol{u}} \int_0^1 \int_{\mathcal{X}} \left(\frac{1}{2}\|\boldsymbol{u}(\boldsymbol{x}, t)\|_2^2 + \Psi(g)\right)\rho_t(\boldsymbol{x})\mathrm{d}\boldsymbol{x}\mathrm{d}t$$

$$\mathrm{s.t.}\partial_t\rho + \nabla_{\boldsymbol{x}} \cdot (\rho\boldsymbol{u}) = \frac{\nu^2}{2}\Delta_{\boldsymbol{x}}\rho + \rho g, \; \rho_0 = \mu_0, \; \rho_1 = \mu_1,$$

$$(10)$$

is a relaxation of it, i.e. the dual of (8) and (10) happens to be the same. The growth penalization $\Psi$ has a complicated form dependent on $\nu$ and $\boldsymbol{q}$. When $\boldsymbol{p}_0 = \boldsymbol{p}_2 = \frac{1}{2}$ and $\boldsymbol{p}_k = 0, k \neq 0, 2$, i.e. particles split into two or die with no preference, $\Psi$ is determined by its Legendre transform

$$\Psi^*_{\nu,\lambda}(g) = \nu^2\lambda(\cosh(\frac{g}{\nu^2}) - 1) \qquad (11)$$

These results connect continuous measure flow with branching processes, allowing us to realize BSB in continuous measure flow framework.

## 4. Simulation-free Training of USB Problem

In this section, we first establish a general framework for learning unbalanced stochastic dynamics (4.1,4.2,4.3,4.4), and then focus on the specific case of USB (4.5,4.6). Note that USB is not an exact BSB solver, but an approximation. Our main goal is to leverage it to capture complex single-cell dynamics instead of solving it.

### 4.1. Unbalanced Score Matching Loss Design

Given a vector field $\boldsymbol{u}_t$ and a rate function $g_t$ that generate the measure path $\rho_t$ by (2), we can view the Fokker-Planck equation with source term as a continuity equation with source term

$$\partial_t\rho_t(\boldsymbol{x}) + \nabla_{\boldsymbol{x}} \cdot (\boldsymbol{u}_t^\circ(\boldsymbol{x})\rho_t(\boldsymbol{x})) = g_t(\boldsymbol{x})\rho_t(\boldsymbol{x}) \qquad (12)$$

where $\boldsymbol{u}_t^\circ(\boldsymbol{x}) = \boldsymbol{u}_t(\boldsymbol{x}) - \frac{\nu^2}{2}\nabla_{\boldsymbol{x}}\ln\rho_t(\boldsymbol{x})$. It is called the drift of probability flow ODE (Tong et al., 2024b) under deterministic settings. The term $\boldsymbol{s}_t(x) = \nabla_{\boldsymbol{x}}\ln\rho_t(\boldsymbol{x})$ is known as the score function. The equation (12) can be generated by a measure flow ODE instead of SDE

$$\begin{cases} \mathrm{d}\boldsymbol{x}_t = \boldsymbol{u}_t^\circ(\boldsymbol{x}_t)\mathrm{d}t \\ \mathrm{d}\ln m_t = g_t(\boldsymbol{x}_t)\mathrm{d}t \end{cases} \tag{13}$$

and the original SDE (1) generating (2) can be viewed as

$$\begin{cases} \mathrm{d}\boldsymbol{x}_t = (\boldsymbol{u}_t^\circ(\boldsymbol{x}_t) + \frac{\nu^2}{2}\boldsymbol{s}_t(\boldsymbol{x}_t))\mathrm{d}t + \nu\mathrm{d}\boldsymbol{W}_t \\ \mathrm{d}\ln m_t = g_t(\boldsymbol{x}_t)\mathrm{d}t \end{cases} \tag{14}$$

Thus, to recover the unbalanced stochastic dynamics (1), it is sufficient to learn a time-dependent **vector field** $\boldsymbol{v}_{\boldsymbol{\theta}}(\boldsymbol{x}, t)$, a time-dependent **growth rate** $g_{\boldsymbol{\theta}}(\boldsymbol{x}, t)$, and a time-dependent **score function** $\boldsymbol{s}_{\boldsymbol{\theta}}(\boldsymbol{x}, t)$ parametrized by neural networks, with loss function specified by minimizing the **intractable unbalanced score matching objective (USM)**

$$\mathcal{L}_{\mathrm{USM}}(\boldsymbol{\theta}) = \\ \int_0^1 \int_{\mathcal{X}} (\|\boldsymbol{v}_{\boldsymbol{\theta}}(\boldsymbol{x}, t) - \boldsymbol{u}_t^\circ(\boldsymbol{x})\|_2^2 + \|g_{\boldsymbol{\theta}}(\boldsymbol{x}, t) - g_t(\boldsymbol{x})\|_2^2 \\ + \lambda^2(t)\|\boldsymbol{s}_{\boldsymbol{\theta}}(\boldsymbol{x}, t) - \boldsymbol{s}_t(\boldsymbol{x})\|_2^2)\rho_t(\boldsymbol{x})\mathrm{d}\boldsymbol{x}\mathrm{d}t \tag{15}$$

We utilized the weight for score loss $\lambda(t)$ adopted from (Tong et al., 2024b) for numerical stability. The details are left to Appendix B.1.

## 4.2. Conditional Path Construction

**Conditional measure path.** Following the approach of defining a conditional measure path in analogy to unbalanced flow matching (Peng et al., 2026), we define a conditional measure path w.r.t the condition variable $\boldsymbol{z}$ such that $\rho_t(\boldsymbol{x}|\boldsymbol{z}) = m_t(\boldsymbol{z})\tilde{\rho}_t(\boldsymbol{x}|\boldsymbol{z})$ where the time-dependent conditional measure $\rho_t(\boldsymbol{x}|\boldsymbol{z})$ is decoupled into a time dependent mass $m_t(\boldsymbol{z})$ and a time-dependent conditional probability density $\tilde{\rho}_t(\boldsymbol{x}|\boldsymbol{z})$. The conditional velocity field, growth rate satisfy the conditional Fokker-Planck equation with source term

$$\partial_t\rho_t(\boldsymbol{x}|\boldsymbol{z}) + \nabla_{\boldsymbol{x}} \cdot (\boldsymbol{u}_t(\boldsymbol{x}|\boldsymbol{z})\rho_t(\boldsymbol{x}|\boldsymbol{z})) = \\ \frac{\nu^2}{2}\Delta_{\boldsymbol{x}}\rho_t(\boldsymbol{x}|\boldsymbol{z}) + \rho_t(\boldsymbol{x}|\boldsymbol{z})g_t(\boldsymbol{x}|\boldsymbol{z}) \tag{16}$$

and we define the conditional score function as $\boldsymbol{s}_t(\boldsymbol{x}|\boldsymbol{z}) = \nabla_{\boldsymbol{x}}\ln\rho_t(\boldsymbol{x}|\boldsymbol{z})$.

**Marginal measure path.** Assuming $\boldsymbol{z} \sim q(\boldsymbol{z})$, we define the marginal measure path from the conditional measure path

$$\rho_t(\boldsymbol{x}) = \int \rho_t(\boldsymbol{x}|\boldsymbol{z})q(\boldsymbol{z})\mathrm{d}\boldsymbol{z} \tag{17}$$

as well as the marginal vector field and growth rate

$$\begin{cases} \boldsymbol{u}_t(\boldsymbol{x}) = \int \boldsymbol{u}_t(\boldsymbol{x}|\boldsymbol{z})\frac{\rho_t(\boldsymbol{x}|\boldsymbol{z})q(\boldsymbol{z})}{\rho_t(\boldsymbol{x})}\mathrm{d}\boldsymbol{z} \\ g_t(\boldsymbol{x}) = \int g_t(\boldsymbol{x}|\boldsymbol{z})\frac{\rho_t(\boldsymbol{x}|\boldsymbol{z})q(\boldsymbol{z})}{\rho_t(\boldsymbol{x})}\mathrm{d}\boldsymbol{z} \end{cases} \tag{18}$$

The following marginalization theorem connects the conditionals and marginals. The proof is left to Appendix A.1.

**Theorem 4.1.** *The marginal vector field and rate function (18) generates the marginal measure path (17) for any $q(\boldsymbol{z})$ independent of $\boldsymbol{x}$ and $t$. The score function and $\boldsymbol{u}^\circ$ also satisfies the marginalization relation $\boldsymbol{s}_t(\boldsymbol{x}) = \int \boldsymbol{s}_t(\boldsymbol{x}|\boldsymbol{z})\frac{\rho_t(\boldsymbol{x}|\boldsymbol{z})q(\boldsymbol{z})}{\rho_t(\boldsymbol{x})}\mathrm{d}\boldsymbol{z}, \boldsymbol{u}_t^\circ(\boldsymbol{x}) = \int \boldsymbol{u}_t^\circ(\boldsymbol{x}|\boldsymbol{z})\frac{\rho_t(\boldsymbol{x}|\boldsymbol{z})q(\boldsymbol{z})}{\rho_t(\boldsymbol{x})}\mathrm{d}\boldsymbol{z}$.*

**Conditional Gaussian measure path.** As a convenient instance of conditional path, conditional Gaussian mesure path (CGMP) is introduced

$$\tilde{\rho}_t(\boldsymbol{x}|\boldsymbol{z}) = \mathcal{N}(\boldsymbol{x}|\boldsymbol{\eta}_t(\boldsymbol{z}), \sigma_t^2(\boldsymbol{z})\mathrm{I}) \tag{19}$$

The conditional vector field, rate function, and score function of CGMP are easy to compute. (Details in Appendix A.2)

**Proposition 4.2.** *For CGMP (19),*

$$\begin{cases} \boldsymbol{u}_t^\circ(\boldsymbol{x}|\boldsymbol{z}) = \frac{\sigma_t'(\boldsymbol{z})}{\sigma_t(\boldsymbol{z})}(\boldsymbol{x} - \boldsymbol{\eta}_t(\boldsymbol{z})) + \boldsymbol{\eta}_t'(\boldsymbol{z}) \\ g_t(\boldsymbol{x}|\boldsymbol{z}) = \partial_t\ln m_t(\boldsymbol{z}) \\ \boldsymbol{s}_t(\boldsymbol{x}|\boldsymbol{z}) = -\frac{\boldsymbol{x} - \boldsymbol{\eta}_t(\boldsymbol{z})}{\sigma_t^2(\boldsymbol{z})} \end{cases} \tag{20}$$

## 4.3. Conditional Loss Design

We can regress the conditionals which are **tractable** by minimizing the **conditional unbalanced score matching objective (CUSM)**

$$\mathcal{L}_{\mathrm{CUSM}}(\boldsymbol{\theta}) = \\ \mathbb{E}_{t\sim\mathcal{U}[0,1],\boldsymbol{z}\sim q(\boldsymbol{z}),\boldsymbol{x}\sim\tilde{\rho}_t(\boldsymbol{x}|\boldsymbol{z})} m_t(\boldsymbol{z})(\|\boldsymbol{v}_{\boldsymbol{\theta}}(\boldsymbol{x}, t) - \boldsymbol{u}_t^\circ(\boldsymbol{x}|\boldsymbol{z})\|_2^2 \\ + \|g_{\boldsymbol{\theta}}(\boldsymbol{x}, t) - g_t(\boldsymbol{x}|\boldsymbol{z})\|_2^2 + \lambda^2(t)\|\boldsymbol{s}_{\boldsymbol{\theta}}(\boldsymbol{x}, t) - \boldsymbol{s}_t(\boldsymbol{x}|\boldsymbol{z})\|_2^2) \tag{21}$$

Here we weight the regression loss by mass $m_t(\boldsymbol{z})$ to deal with unbalanced mass. In a deterministic setting, the loss reduces to the **CUFM** loss proposed by (Peng et al., 2026). In a balanced setting where $m_t(\boldsymbol{z}) \equiv 1, g_t(\boldsymbol{x}|\boldsymbol{z}) \equiv 0$, (21) naturally reduces to the standard conditional score matching loss (Tong et al., 2024b).

The following theorem recovers the classical results of matching algorithms that one can minimize the intractable marginal objective (15) by minimizing the tractable conditional objective (21). The proof is left to Appendix A.3.

*Table 1.* Examples of coupling and conditional path construction

| Algorithm | coupling | $x_t$ | $m_t$ |
|---|---|---|---|
| OT-CFM | OT coupling $\pi$ | $tx_1 + (1-t)x_0$ | 1 |
| SF$^2$M | ROT coupling $\pi_{2\nu^2}$ | $\mathcal{N}(tx_1 + (1-t)x_0, \nu^2 t(1-t)\mathrm{I})$ | 1 |
| WFR-FM | WFR semi-coupling $(\gamma_0, \gamma_1)$ | $\int_0^t \frac{\omega \mathrm{d}s}{As^2 + Bs + C}$ | $At^2 + Bt + C$ |
| **USB** | RUOT semi-coupling $(\gamma_0, \gamma_1)$ | $\mathcal{N}(tx_1 + (1-t)x_0, \nu^2 t(1-t)\mathrm{I})$ | $m_0^{1-t} m_1^t$ |

**Theorem 4.3.** *If $\rho_t(\boldsymbol{x}) > 0$ for all $\boldsymbol{x} \in \mathcal{X}$ and $t \in [0, 1]$, and $q(\boldsymbol{z})$ is independent of $\boldsymbol{x}$ and $t$, then $\mathcal{L}_{\mathrm{USM}}(\boldsymbol{\theta}) = \mathcal{L}_{\mathrm{CUSM}}(\boldsymbol{\theta}) + C$, where $C$ is independent of $\boldsymbol{\theta}$. Thus they have identical gradients w.r.t $\boldsymbol{\theta}$, i.e.*

$$\nabla_{\boldsymbol{\theta}} \mathcal{L}_{\mathrm{USM}}(\boldsymbol{\theta}) = \nabla_{\boldsymbol{\theta}} \mathcal{L}_{\mathrm{CUSM}}(\boldsymbol{\theta}).$$

### 4.4. A General Framework for Learning Unbalanced Stochastic Dynamics

In sections above, we have established a general framework for learning unbalanced stochastic dynamics. Once $q(\boldsymbol{z})$ and the conditionals $\boldsymbol{u}_t^\circ(\boldsymbol{x}|\boldsymbol{z}), g_t(\boldsymbol{x}|\boldsymbol{z}), \boldsymbol{s}_t(\boldsymbol{x}|\boldsymbol{z})$ are specified, we can regress the marginals $\boldsymbol{u}_t^\circ(\boldsymbol{x}), g_t(\boldsymbol{x}), \boldsymbol{s}_t(\boldsymbol{x})$ by minimizing (21). A common choice for $\boldsymbol{z}$ is $\boldsymbol{z} = (\boldsymbol{x}_0, \boldsymbol{x}_1)$ i.e. the pair of start point and end point from the source and target respectively. In this case, $q(\boldsymbol{z})$ stands for some coupling between the source and target, while the conditionals stand for the conditional path between two Diracs $m_0 \delta_{\boldsymbol{x}_0}, m_1 \delta_{\boldsymbol{x}_1}$. $m_0, m_1$ are determined by the semi-coupling, which reduces to one coupling in balanced cases, resulting in $m_0 = m_1 = 1$. We list the semi-coupling/coupling and conditional path used by some previous methods, as well as USB in Table 1. In the following part of this section, we will focus on the conditional path and semi-coupling of USB.

### 4.5. Conditional Path Construction for USB

We disintegrate the KL-divergence (8)

$$\mathrm{KL}(\mathbb{P}\|\mathbb{Q}) = \mathrm{KL}(\mathbb{P}_{01}\|\mathbb{Q}_{01}) + \int \mathrm{KL}(\mathbb{P}^{\boldsymbol{xy}}\|\mathbb{Q}^{\boldsymbol{xy}})\mathbb{P}_{01}(\mathrm{d}\boldsymbol{x}, \mathrm{d}\boldsymbol{y})$$

$$(22)$$

where $\mathbb{Q}$ is now a BBM. The second term can be minimized to 0 by choosing $\mathbb{P}^{\boldsymbol{xy}} = \mathbb{Q}^{\boldsymbol{xy}}$. Note that it is hard to condition on a BBM in strong sense due to its branching nature. But, the conditional measure path can be constructed based on the evolution equation of BBM (9). The solution of (9) is a weighted Gaussian $\rho_t(\boldsymbol{x}) = e^{rt} \mathcal{N}(\boldsymbol{x}|\boldsymbol{0}, \nu^2 \mathrm{I})$ where the position $\boldsymbol{x}$ follows a Brownian motion with diffusion parameter $\nu$, and the mass varies linearly in log-scale. Intuitively, since BBM branching events follow an exponential distribution and result in a multiplication of the particle count, the particle number can be viewed approximately as a Poisson process in log-scale. The log-linear mass can be viewed as a limit of Poisson process in log-scale with infinitesimal increment. Thus, given a pair of Dirac $(m_0 \delta_{\boldsymbol{x}_0}, m_1 \delta_{\boldsymbol{x}_1})$ as source and target, we bridge them by a **Poisson-Brownian**

**bridge**, i.e. we bridge $(\boldsymbol{x}_0, \boldsymbol{x}_1)$ with Brownian bridge, and bridge the mass with linear interpolation in log-scale, which is the limit of Poission bridge (Conforti et al., 2015) with infinitesimal increment. The details are presented in Appendix B.2.

$$\begin{cases} \mathrm{d}\boldsymbol{x}_t = \dfrac{\boldsymbol{x}_1 - \boldsymbol{x}_t}{1-t} \mathrm{d}t + \nu \mathrm{d}\boldsymbol{W}_t \\ \mathrm{d}\ln m_t = (\ln m_1 - \ln m_0)\mathrm{d}t \end{cases} \quad (23)$$

As a CGMP $\rho_t(\boldsymbol{x}|\boldsymbol{z}) = m_t(\boldsymbol{z})\mathcal{N}(\boldsymbol{\eta}_t(\boldsymbol{z}), \nu^2 t(1-t)\mathrm{I})$ where $m_t(\boldsymbol{z}) = m_0(\boldsymbol{z})^{1-t} m_1(\boldsymbol{z})^t$, $\boldsymbol{\eta}_t(\boldsymbol{z}) = (1-t)\boldsymbol{x}_0 + t\boldsymbol{x}_1$, $\boldsymbol{z} = (\boldsymbol{x}_0, \boldsymbol{x}_1)$, the conditionals of Poisson-Brownian bridge is easy to obtain

$$\begin{cases} \boldsymbol{u}_t^\circ(\boldsymbol{x}|\boldsymbol{z}) = \dfrac{1-2t}{t(1-t)}\big(\boldsymbol{x} - ((1-t)\boldsymbol{x}_0 + t\boldsymbol{x}_1)\big) + (\boldsymbol{x}_1 - \boldsymbol{x}_0) \\ g_t(\boldsymbol{x}|\boldsymbol{z}) = \ln m_1(\boldsymbol{x}_0, \boldsymbol{x}_1) - \ln m_0(\boldsymbol{x}_0, \boldsymbol{x}_1) \\ \boldsymbol{s}_t(\boldsymbol{x}|\boldsymbol{z}) = \dfrac{(1-t)\boldsymbol{x}_0 + t\boldsymbol{x}_1 - \boldsymbol{x}}{\nu^2 t(1-t)} \end{cases}$$

$$(24)$$

### 4.6. Static Coupling Construction for USB

The minimization of the first term of (22) results in a static USB semi-coupling $(\gamma_0, \gamma_1)$. Though hard to obtain, note that RUOT is a relaxation of USB (Baradat & Lavenant, 2021). Thus, we use the semi-coupling induced by the corresponding RUOT problem (10) to approximate the static USB semi-coupling. Here we restrict the branching mechanism of the referencing BBM to $\boldsymbol{p}_0 = \boldsymbol{p}_2 = \frac{1}{2}, \boldsymbol{p}_k = 0, k \neq 0, 2$ to imitate the real cell division and apoptosis.

In practice, it is also hard to calculate the semi-coupling for general RUOT problem due to the complexity of the growth penalty $\Psi$. For efficiency, we approximate the RUOT problem with a WFR problem (3), which is easy to solve, by expanding $\Psi$ to second order. More computational details are presented in Appendix B.3. We also discuss the main difficulty in Appendix E.

## 5. USB Workflow for Trajectory Inference

**Multi-time points USB.** Given samples from unbalanced measures $\mu_i$ at $K + 1$ discrete time points, $t = t_0, t_1, \ldots, t_K$, multi-time points USB tends to find a most likely evolution bridging these marginal measures.

$$\mathbb{P}^\star = \underset{\mathbb{P}:p_i = \mu_i, i=0,1,\cdots,K+1}{\arg\min} \mathrm{KL}(\mathbb{P}\|\mathbb{Q}) \quad (25)$$

where $\mathbb{Q}$ is a BBM. The KL-disintegration yields

$$\mathrm{KL}(\mathbb{P}\|\mathbb{Q}) = \mathrm{KL}(\mathbb{P}_{\{0,1,\cdots,K\}}\|\mathbb{Q}_{\{0,1,\cdots,K\}}) +$$

$$\int \mathrm{KL}(\mathbb{P}^{\boldsymbol{x}_0\cdots\boldsymbol{x}_K}\|\mathbb{Q}^{\boldsymbol{x}_0\cdots\boldsymbol{x}_K})\mathbb{P}_{\{0,1,\cdots,K\}}(\mathrm{d}\boldsymbol{x}_0, \cdots \mathrm{d}\boldsymbol{x}_K)$$

$$(26)$$

Similar to the two-time points case, the second term can be minimized to 0 by choosing $\mathbb{P}^{\boldsymbol{x}_0\cdots\boldsymbol{x}_K} = \mathbb{Q}^{\boldsymbol{x}_0\cdots\boldsymbol{x}_K}$. Due to Markov property, conditioning on multiple time points is equivalent to conditioning on consecutive pairs and stitching the corresponding conditional measure paths together.

The minimization of the first term results in $K$ pairs of semi-couplings. These semi-couplings can be approximated by solving a multi-marginal RUOT problem

$$\text{RUOT}(\mu_0, \cdots, \mu_K) =$$

$$\inf_{\rho, g, \boldsymbol{u}} \frac{t_K - t_0}{2} \int_{t_0}^{t_K} \int_{\mathcal{X}} \left( \frac{1}{2}\|\boldsymbol{u}(\boldsymbol{x},t)\|_2^2 + \Psi(g) \right) \rho_t(\boldsymbol{x}) \mathrm{d}\boldsymbol{x}\mathrm{d}t$$

$$\text{s.t.} \partial_t \rho + \nabla_{\boldsymbol{x}} \cdot (\rho \boldsymbol{u}) = \frac{\nu^2}{2}\Delta_{\boldsymbol{x}}\rho + \rho g, \; \rho_i = \mu_i, i = 0, \cdots, K$$

(27)

Due to Markov property again, the dynamics between different consecutive time pairs are independent. According to (Peng et al., 2026), under this independence assumption, the multi-time problem reduces to solving RUOT between successive time points. The proof is left to Appendix A.4.

**Proposition 5.1.** *The solution to the multi-time RUOT problem (27) is equivalent to the concatenation of the solutions of successive time points.*

**Training workflow.** In practice, the neural networks share the parameters across different pairs of time points. For convenience, we choose the condition variable $\boldsymbol{z} = (\boldsymbol{x}_0.\boldsymbol{x}_1) \sim \gamma_0(\boldsymbol{x}_0.\boldsymbol{x}_1)$, and set $m_0(\boldsymbol{z}) = 1$, $m_1(\boldsymbol{z}) = \frac{\gamma_1(\boldsymbol{x}_0.\boldsymbol{x}_1)}{\gamma_0(\boldsymbol{x}_0.\boldsymbol{x}_1)}$ for each two-time points cases. We described the pseudocode of USB training workflow at multiple time points in 1.

**Inference schemes.** We provide two inference modes for USB, namely, **Continuous Inference** and **Branching Inference**. For **Continuous Inference**, USB simulates the trajectory by following SDE

$$\begin{cases} \mathrm{d}\boldsymbol{x}_t = (\boldsymbol{v}_{\boldsymbol{\theta}}(\boldsymbol{x}_t, t) + \frac{\nu^2}{2}\boldsymbol{s}_{\boldsymbol{\theta}}(\boldsymbol{x}_t, t))\mathrm{d}t + \nu \mathrm{d}\boldsymbol{W}_t \\ \mathrm{d}\ln m_t = g_{\boldsymbol{\theta}}(\boldsymbol{x}_t, t)\mathrm{d}t \end{cases}$$

(28)

The continuous mode aims to match the source and target, and recover the USB dynamics on population-level, while the **Branching Inference** aims to simulate the discrete single-cell birth-death dynamics at single-cell resolution. We start from a cell at $\boldsymbol{x}_0$ whose position evolves according to the upper SDE of (28). We also simulates a non-homogeneous branching process with rate $|g_{\boldsymbol{\theta}}(\boldsymbol{x}_t, t)|$ to decide when the cell undergoes branching. At the branching event, the cell divides into two ($g_{\boldsymbol{\theta}} > 0$) or dies ($g_{\boldsymbol{\theta}} < 0$). After branching, all living cells follow the simulation process above independently. More details are presented in Appendix B.4. Pseudocodes are presented in Appendix F.

**A full model for single-cell dynamics.** We discussed several trajectory inference algorithms related to USB in

*Table 2.* Comparison of trajectory inference algorithms based on Fokker-Planck-like equation(2).

| Algorithm | unbalanced | stochastic | Simulation-free | discrete birth-death dynamics |
|---|---|---|---|---|
| OT-CFM | ✗ | ✗ | ✓ | ✗ |
| SF²M | ✗ | ✓ | ✓ | ✗ |
| WFR-FM | ✓ | ✗ | ✓ | ✗ |
| DeepRUOT | ✓ | ✓ | ✗ | ✗ |
| BranchSBM | ✗ | ✓ | ✗ | ✗ |
| **USB** | ✓ | ✓ | ✓ | ✓ |

Appendix D. Among algorithms based on Fokker-Planck-like equations (2), USB models both the stochastic effect $\frac{\nu^2}{2}\Delta_{\boldsymbol{x}}\rho_t(\boldsymbol{x})$ and the unbalanced effect $g_t(\boldsymbol{x})\rho_t(\boldsymbol{x})$ in a simulation-free manner, which also allows discrete simulation for birth-death dynamics (Table 2).

---

**Algorithm 1** USB training workflow

---

**Require:** Sample-able distributions $\mu_{t_0}, \mu_{t_1}, \ldots, \mu_{t_K}$, diffusion parameter $\nu$, branching rate $\lambda$, training batch size $b$, vector net $\boldsymbol{v}_{\boldsymbol{\theta}}(\boldsymbol{x}, t)$, growth rate net $g_{\boldsymbol{\theta}}(\boldsymbol{x}, t)$, score net $\boldsymbol{s}_{\boldsymbol{\theta}}(\boldsymbol{x}, t)$.
1: **for** $k = 0 \to K - 1$ **do**
2: $\quad (\gamma_0^{(k)}, \gamma_1^{(k)}) \leftarrow$ coupling of $\text{RUOT}_{\nu,\lambda}(\mu_k, \mu_{k+1})$ with penalty $\Psi_{\nu,\lambda}(g)$ determined by (11)
3: **end for**
4: **while** Training **do**
5: $\quad$ **for** $k = 0 \to K - 1$ **do**
6: $\quad\quad (\boldsymbol{x}_{t_k}, \boldsymbol{x}_{t_{k+1}}) \sim \gamma_0^{(k)}$
7: $\quad\quad t \sim \mathcal{U}(0, 1), t^{(k)} \leftarrow t_k + (t_{k+1} - t_k)t$
8: $\quad\quad \boldsymbol{\eta}_{t^{(k)}} \leftarrow \boldsymbol{x}_{t_k} + t(\boldsymbol{x}_1 - \boldsymbol{x}_{t_k})$
9: $\quad\quad \boldsymbol{x}^{(k)} \sim \mathcal{N}(\boldsymbol{\eta}_{t^{(k)}}, \nu^2 t(1 - t)\mathbf{I})$
10: $\quad\quad \boldsymbol{u}^{(k)} \leftarrow \frac{1-2t}{t(1-t)}(\boldsymbol{x}^{(k)} - \boldsymbol{\eta}_{t^{(k)}}) + (\boldsymbol{x}_{t_{k+1}} - \boldsymbol{x}_{t_k})$
11: $\quad\quad g^{(k)} \leftarrow \ln \frac{m_{t_{k+1}}(\boldsymbol{x}_{t_k}, \boldsymbol{x}_{t_{k+1}})}{m_{t_k}(\boldsymbol{x}_{t_k}, \boldsymbol{x}_{t_{k+1}})}/(t_{k+1} - t_k)$
12: $\quad\quad s^{(k)} \leftarrow \frac{\boldsymbol{\eta}_{t^{(k)}} - \boldsymbol{x}^{(k)}}{\nu^2 t(1-t)}$
13: $\quad\quad m^{(k)} \leftarrow m_t(\boldsymbol{x}_{t_k}, \boldsymbol{x}_{t_{k+1}})/\gamma_0^{(k)}(\boldsymbol{x}_{t_k}, \boldsymbol{x}_{t_{k+1}})$ (Defined in 23 and 24)
14: $\quad$ **end for**
15: $\quad$ Concatenate $\{\boldsymbol{x}^{(i)}, t^{(i)}, \boldsymbol{u}^{(i)}, g^{(i)}, s^{(i)}, m^{(i)}\}_{i=1}^K$ into batch tensors $\{\boldsymbol{x}^c, t^c, \boldsymbol{u}^c, g^c, s^c, m^c\}$
16: $\quad \mathcal{L}_{\text{CUSM}}(\boldsymbol{\theta}) \leftarrow (\|\boldsymbol{v}_{\boldsymbol{\theta}}(\boldsymbol{x}^c, t^c) - \boldsymbol{u}^c\|_2^2 + \|g_{\boldsymbol{\theta}}(\boldsymbol{x}^c, t^c) - g^c\|_2^2 + \lambda^2(t)\|\boldsymbol{s}_{\boldsymbol{\theta}}(\boldsymbol{x}^c, t^c) - s^c\|_2^2)m^c$
17: $\quad \boldsymbol{\theta} \leftarrow \text{Update}(\boldsymbol{\theta}, \nabla_{\boldsymbol{\theta}}\mathcal{L}_{\text{CUSM}}(\boldsymbol{\theta}))$
18: **end while**
19: **return** $\boldsymbol{v}_{\boldsymbol{\theta}}, g_{\boldsymbol{\theta}}$, and $\boldsymbol{s}_{\boldsymbol{\theta}}$

---

## 6. Experiment Results

**USB accurately bridges** $\mu_{t_0}, \cdots, \mu_{t_K}$**.** We evaluate how accurate can USB match the measure at observed time points on three synthetic datasets: Simulation Gene (Simulation), Dyngen and the 1000D Gaussian Mixtures (Gaussian).

*Table 3.* Mean $\mathcal{W}_1$ and RME (only for unbalanced methods) on synthetic datasets. For the methods that exhibit randomness in inference, we report the mean value and standard deviation over 5 runs. Best results are in bold, and the second best are underlined.

| Method | Simulation (2D) | | Dyngen (5D) | | Gaussian (1000D) | |
|---|---|---|---|---|---|---|
| | $\mathcal{W}_1 (\downarrow)$ | RME $(\downarrow)$ | $\mathcal{W}_1 (\downarrow)$ | RME $(\downarrow)$ | $\mathcal{W}_1 (\downarrow)$ | RME $(\downarrow)$ |
| MMFM | 0.298 | — | 1.371 | — | 2.833 | — |
| Metric FM | 0.311 | — | 1.767 | — | 3.794 | — |
| SF2M | $0.224_{\pm0.007}$ | — | $1.277_{\pm0.017}$ | — | $3.543_{\pm0.002}$ | — |
| MIOFlow | 0.148 | — | 0.965 | — | 2.858 | — |
| BranchSBM | 0.474 | — | 1.415 | — | 3.438 | — |
| TIGON | 0.045 | 0.014 | 0.512 | 0.047 | 2.263 | 0.127 |
| DeepRUOT | $\underline{0.043}_{\pm0.002}$ | $0.017_{\pm0.001}$ | $0.623_{\pm0.032}$ | $0.065_{\pm0.011}$ | $3.785_{\pm0.009}$ | $0.303_{\pm0.070}$ |
| Var-RUOT | $0.079_{\pm0.003}$ | $0.008_{\pm0.002}$ | $0.522_{\pm0.008}$ | $0.177_{\pm0.007}$ | $2.813_{\pm0.004}$ | $0.041_{\pm0.006}$ |
| UOT-FM | 0.093 | 0.010 | 1.204 | 0.097 | 2.771 | $\underline{0.033}$ |
| VGFM | 0.046 | 0.006 | 0.598 | 0.037 | 3.010 | 0.037 |
| WFR-FM | **0.019** | **0.001** | $\underline{0.135}$ | **0.005** | $\underline{2.233}$ | 0.044 |
| USB | $\mathbf{0.019}_{\pm0.000}$ | $\underline{0.002}_{\pm0.000}$ | $\mathbf{0.131}_{\pm0.001}$ | $\underline{0.007}_{\pm0.000}$ | $\mathbf{2.136}_{\pm0.002}$ | $\mathbf{0.004}_{\pm0.004}$ |

*Table 4.* Mean $\mathcal{W}_1$ over held-out time points on EMT, EB, CITE and Mouse datasets. For the methods that exhibit randomness in inference, we report the mean value and standard deviation over 5 runs. Best results are in bold, and the second best are underlined.

| Method | EMT (10D) | EB (50D) | CITE (50D) | Mouse (50D) |
|---|---|---|---|---|
| MMFM | 0.323 | 11.213 | 38.521 | 8.263 |
| Metric FM | 0.314 | 10.726 | 37.342 | 7.753 |
| SF2M | $0.308_{\pm0.001}$ | $10.986_{\pm0.006}$ | $38.333_{\pm0.002}$ | $8.646_{\pm0.004}$ |
| MIOFlow | 0.325 | 10.960 | 39.574 | 7.779 |
| BranchSBM | 0.369 | 11.988 | 39.125 | 8.586 |
| TIGON | 0.360 | 11.080 | 38.159 | 6.868 |
| DeepRUOT | $0.323_{\pm0.002}$ | $\mathbf{10.075}_{\pm0.004}$ | $37.892_{\pm0.002}$ | $\underline{6.847}_{\pm0.003}$ |
| Var-RUOT | $0.320_{\pm0.003}$ | $11.035_{\pm0.017}$ | $38.393_{\pm0.029}$ | $8.672_{\pm0.040}$ |
| UOT-FM | 0.322 | 11.344 | 38.649 | 9.332 |
| VGFM | $\underline{0.301}$ | 10.370 | 37.386 | 8.496 |
| WFR-FM | **0.298** | $\underline{10.157}$ | $\underline{37.221}$ | **6.586** |
| USB | $\mathbf{0.298}_{\pm0.0001}$ | $10.177_{\pm0.0001}$ | $\mathbf{37.087}_{\pm0.0001}$ | $6.988_{\pm0.0001}$ |

The distribution-matching accuracy is measured by the **1-Wasserstein distance** ($\mathcal{W}_1$) between the normalized true measure and the normalized USB simulated measure, and the mass-matching accuracy is measured by **Relative Mass Error (RME)**, the relative error of predicted total mass (Experiment details in Appendix C.2). USB demonstrates superior performance across all three datasets, achieving best accuracy on distribution-matching task, and best or second-best accuracy on mass-matching task, performing on par with the top-performing baseline, while consistently and significantly outperforming its balanced counterpart, SF²M (Tong et al., 2024b) (Table 3). We also evaluate USB on several real datasets in Appendix C.

**USB accurately interpolates the unobserved time points.** We evaluate how well can USB recover the true underlying dynamics by hold-one-out experiments. For a datasets with $K + 1$ time points $t_0, t_1, \cdots, t_K$, holding out one time point $t_i, 1 \leq i \leq K - 1$, USB is trained on the remaining $K$ time points, and the $\mathcal{W}_1$ is evaluated at the holding out time point $t_i$. The mean performance on all holding out time points are presented in Table 4. Across EMT (Cook & Vanderhyden, 2020), EB (Moon et al., 2019), CITE-seq (Lance et al., 2022), and mouse hematopoiesis data (Mouse) (Weinreb et al., 2020), USB achieves best interpolation accuracy on EMT and CITE, while performs comparable with top baselines on others. This demonstrates that USB successfully captures the underlying cellular dynamics.

**USB recovers the underlying birth-death dynamics.** To evaluate how well can USB recover the underling birth-death dynamics, we calculated the Pearson correlation ($g_{corr}$) between the true growth rate $g = \frac{X_2^2}{2(1+X_2^2)}$ and the predicted growth rate on the Simulation Gene dataset. Averaging on 5 runs, USB gets a mean Pearson correlation of 0.9739, showing high consistency to the ground truth. The growth rate plot on the 1000D Gaussian dataset also shows that USB recovers a more plausible growth rate than WFR, which is designed for capturing unbalanced effect (Figure 1).

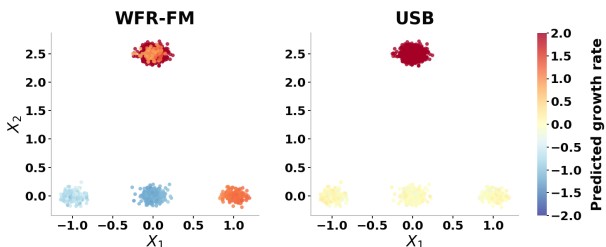

*Figure 1.* Learned growth rate on the Gaussian 1000D dataset. Left panel: WFR-FM; Right panel: USB

In the 1000D Gaussian settings, the upper cluster expands from 100 to 1,000 cells without displacement, whereas the lower cluster maintains its total counts (400), bifurcating into two 200-cell clusters. Consequently, the ground-truth growth rate is high in the upper cluster and zero across the lower clusters; USB recovers this behavior more faithfully than WFR. The difference is because of the different underlying dynamic assumptions of the two algorithms. More detailed discussion is presented in Appendix D.2.

**USB simulates discrete birth-death dynamics at single-cell resolution.** On the Dyngen dataset, we initiated simulations from the same cell to generate 8 trajectories ($\nu = 0.1$) using **Branching Inference**, distinguished by different colors (Fig.2). The total mass of the Dyngen data initially decreases and subsequently increases, evolving into two imbalanced branches, with the lower branch containing a higher cell count than the upper one (0.708 : 0.292). We observe that cell fates diverse significantly even when originating from the same initial state: the majority of cells undergo early apoptosis, while some trajectories traverse towards distinct branches, experiencing subsequent division or death events. These results demonstrate USB's capacity to model not only cell division but also apoptosis and complex bifurcation behaviors.

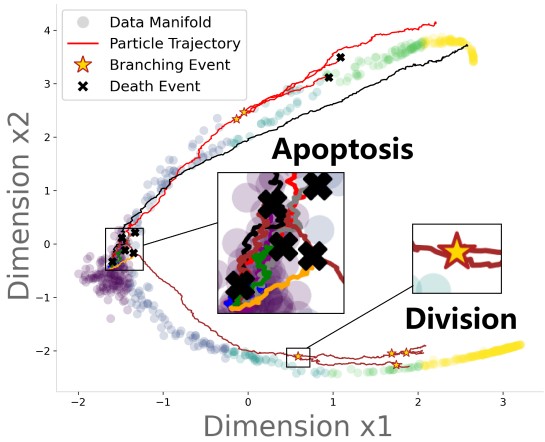

*Figure 2.* Single-cell resolution birth-death dynamics simulation

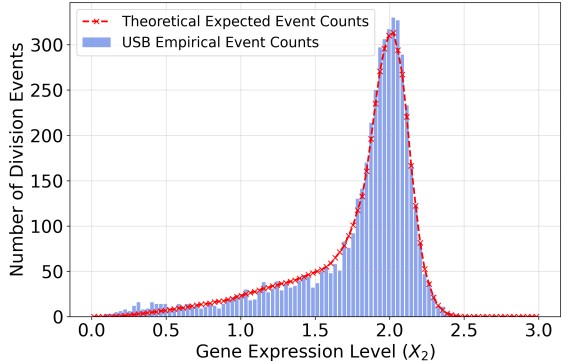

*Figure 3.* Event-Count Statistics of cell division on Simulation data.

**USB captures the discrete dynamics at both microscopic and macroscopic levels.** We rigorously validate USB's discrete birth-death simulations, we conducted evaluations at both microscopic and macroscopic levels. At microscopic level, we calculate the **Event-Count Statistics** of cell division on Simulation dataset. The ground-truth division rate is $g = \frac{X_2^2}{2(1+X_2^2)}$. We simulated 2,000 branching trajectories over 400 steps ($\Delta t = 0.01$). Collecting all coordinates traversed along these paths, we partitioned them into 100 bins. For each bin, we computed the **Theoretical Expected Event Counts** ($\sum g(X_2)\Delta t$) and counted the **Empirical Event Counts** (actual simulated divisions falling in that bin). The Pearson correlation between theoretical and empirical counts across the 100 bins is 0.9950, with a relative error of 10.7% (Figure 3). This rigorously proves USB precisely captures state-dependent discrete dynamics at single-cell resolution.

At macroscopic level, we calculate the **Branch Occupancy Ratios** on Dyngen dataset. We clustered real terminal states

(time point 4) into two branches via K-means to establish ground-truth branch occupancy. We then applied this classifier to the terminal states of 15,700 USB-simulated branching trajectories. The ground-truth ratio is 0.708 : 0.292, while USB predicts 0.650 : 0.350. The accumulation of microscopic discrete events in USB successfully calibrates macroscopic bifurcation structures, faithfully reproducing true biological dynamics.

## 7. Conclusion, Limitation and Discussion

In this work, we introduced USB, a framework rooted in the BSB problem. By employing unbalanced score matching, USB efficiently integrates the stochastic and unbalanced effect inherent in single-cell dynamics, and explicitly permits the discrete simulation of birth-death dynamics. We validated the effectiveness and robustness of the method on both synthetic and real scRNA-seq datasets, demonstrating its significant potential for biological trajectory inference.

A limitation of USB is that the semi-coupling of BSB and its RUOT relaxation are intractable. Consequently, we use WFR for approximation. The underlying mathematics merits further investigation. However, the unbalanced score matching framework we have established is indeed general: provided that the static semi-coupling and the conditional path of the target problem, our approach can be extended to other domains. This generality renders USB applicable to a wider range of dynamical modeling or machine learning scenarios involving stochasticity and unbalance, or featuring microscopic branching and discrete mass variations.

### Code Availability

The code is available at https://github.com/Sirin6/USB.

### Impact Statement

This paper presents work whose goal is to advance the field of machine learning. There are many potential societal consequences of our work, none of which we feel must be specifically highlighted here.

### Acknowledgments

This work was supported by the National Natural Science Foundation of China (NSFC No. 12225102 to L.Z., 8206100646 to P.Z., T2321001 to P.Z. & L.Z., and 12288101 to P.Z. & L.Z.) and the National Key R&D Program of China No.2024YFA0919500 to L.Z. We thank the anonymous referees for their valuable feedback and constructive suggestions.

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

# A. Proofs

Proofs of theorems and propositions

## A.1. Proof of Theorem 4.1

**Theorem 4.1.** *The marginal vector field and rate function (18) generates the marginal measure path (17) for any $q(z)$ independent of $x$ and $t$. The score function and $u^\circ$ also satisfies the marginalization relation $s_t(x) = \int s_t(x|z)\frac{\rho_t(x|z)q(z)}{\rho_t(x)}\,\mathrm{d}z, u_t^\circ(x) = \int u_t^\circ(x|z)\frac{\rho_t(x|z)q(z)}{\rho_t(x)}\,\mathrm{d}z.*$

*Proof.* To show that the marginals generate the marginal measure path (17), it is sufficient to show that the marginal measure $\rho_t(x)$ satisfies the Fokker-Planck equation with source term (2).

$$\partial_t \rho_t(x) = \frac{d}{dt}\int \rho_t(x|z)q(z)\mathrm{d}z$$

$$= \int \partial_t \rho_t(x|z)q(z)\mathrm{d}z$$

$$\stackrel{(1)}{=} \int -\nabla_x \cdot (u_t(x|z)\rho_t(x|z))q(z)\mathrm{d}z + \int \frac{\nu^2}{2}\Delta_x \rho_t(x|z)q(z)\mathrm{d}z + \int g_t(x|z)\rho_t(x|z)q(z)\mathrm{d}z$$

$$= -\nabla_x \cdot \int u_t(x|z)\rho_t(x|z)q(z)\mathrm{d}z + \frac{\nu^2}{2}\Delta_x \int \rho_t(x|z)q(z)\mathrm{d}z + \int g_t(x|z)\rho_t(x|z)q(z)\mathrm{d}z$$

$$= -\nabla_x \cdot \left(\rho_t(x)\int u_t(x|z)\frac{\rho_t(x|z)q(z)}{\rho_t(x)}\mathrm{d}z\right) + \frac{\nu^2}{2}\Delta_x \rho_t(x) + \rho_t(x)\int g_t(x|z)\frac{\rho_t(x|z)q(z)}{\rho_t(x)}\mathrm{d}z$$

$$\stackrel{(2)}{=} -\nabla_x \cdot (\rho_t(x)u_t(x)) + \frac{\nu^2}{2}\Delta_x \rho_t(x) + \rho_t(x)g_t(x)$$

In (1), we use the Fokker-Planck equation with source term of the conditional measure $\rho_t(x|z)$ (16). In (2), we use the definition of marginal vector field and rate function (18). Thus, the marginal measure $\rho_t(x)$ satisfies the Fokker-Planck equation with source term (2).

By direct calculation,

$$s_t(x) = \nabla_x \ln \rho_t(x)$$

$$= \frac{1}{\rho_t(x)}\nabla_x \rho_t(x)$$

$$= \frac{1}{\rho_t(x)}\nabla_x \int \rho_t(x|z)q(z)\,\mathrm{d}z$$

$$= \frac{1}{\rho_t(x)}\int \nabla_x \rho_t(x|z)q(z)\,\mathrm{d}z$$

$$= \frac{1}{\rho_t(x)}\int \nabla_x \ln \rho_t(x|z)\rho_t(x|z)q(z)\,\mathrm{d}z$$

$$= \int s_t(x|z)\frac{\rho_t(x|z)q(z)}{\rho_t(x)}\,\mathrm{d}z$$

$$u_t^\circ(x) = u_t(x) - \frac{\nu^2}{2}s_t(x)$$

$$= \int u_t(x|z)\frac{\rho_t(x|z)q(z)}{\rho_t(x)}\,\mathrm{d}z - \frac{\nu^2}{2}\int s_t(x|z)\frac{\rho_t(x|z)q(z)}{\rho_t(x)}\,\mathrm{d}z$$

$$= \int \left(u_t(x|z) - \frac{\nu^2}{2}s_t(x|z)\right)\frac{\rho_t(x|z)q(z)}{\rho_t(x)}\,\mathrm{d}z$$

$$= \int u_t^\circ(x|z)\frac{\rho_t(x|z)q(z)}{\rho_t(x)}\,\mathrm{d}z$$

The relation between conditionals and marginals is verified. The proof is completed. $\qquad\square$

### A.2. Proof or Proposition 4.2

**Proposition 4.2.** For CGMP (19),

$$
\begin{cases}
\boldsymbol{u}_t^\circ(\boldsymbol{x}|\boldsymbol{z}) = \dfrac{\sigma_t'(\boldsymbol{z})}{\sigma_t(\boldsymbol{z})}\big(\boldsymbol{x} - \boldsymbol{\eta}_t(\boldsymbol{z})\big) + \boldsymbol{\eta}_t'(\boldsymbol{z}) \\[2mm]
g_t(\boldsymbol{x}|\boldsymbol{z}) = \partial_t \ln m_t(\boldsymbol{z}) \\[2mm]
\boldsymbol{s}_t(\boldsymbol{x}|\boldsymbol{z}) = -\dfrac{\boldsymbol{x} - \boldsymbol{\eta}_t(\boldsymbol{z})}{\sigma_t^2(\boldsymbol{z})}
\end{cases}
$$

*Proof.* The CMGP satisfies the following conditional Fokker-Planck equation with source term (16)

$$
\partial_t \rho_t(\boldsymbol{x}|\boldsymbol{z}) + \nabla_{\boldsymbol{x}} \cdot (\boldsymbol{u}_t(\boldsymbol{x}|\boldsymbol{z})\rho_t(\boldsymbol{x}|\boldsymbol{z})) = \frac{\nu^2}{2}\Delta_{\boldsymbol{x}}\rho_t(\boldsymbol{x}|\boldsymbol{z}) + \rho_t(\boldsymbol{x}|\boldsymbol{z})g_t(\boldsymbol{x}|\boldsymbol{z})
$$

Expanding the conditional measure into mass and density, the equation become

$$
m_t(\boldsymbol{z})\partial_t \tilde{\rho}_t(\boldsymbol{x}|\boldsymbol{z}) + \partial_t m_t(\boldsymbol{z})\tilde{\rho}_t(\boldsymbol{x}|\boldsymbol{z}) + m_t(\boldsymbol{z})\nabla_{\boldsymbol{x}} \cdot (\boldsymbol{u}_t(\boldsymbol{x}|\boldsymbol{z})\tilde{\rho}_t(\boldsymbol{x}|\boldsymbol{z})) = \frac{\nu^2}{2}m_t(\boldsymbol{z})\Delta_{\boldsymbol{x}}\tilde{\rho}_t(\boldsymbol{x}|\boldsymbol{z}) + m_t(\boldsymbol{z})\tilde{\rho}_t(\boldsymbol{x}|\boldsymbol{z})g_t(\boldsymbol{x}|\boldsymbol{z}) \quad (29)
$$

Since $m_t(\boldsymbol{z}) > 0$ for all $t$, we devide the both sides of the equation by $m_t(\boldsymbol{z})$ and get

$$
\partial_t \tilde{\rho}_t(\boldsymbol{x}|\boldsymbol{z}) + \partial_t \ln m_t(\boldsymbol{z})\tilde{\rho}_t(\boldsymbol{x}|\boldsymbol{z}) + \nabla_{\boldsymbol{x}} \cdot (\boldsymbol{u}_t(\boldsymbol{x}|\boldsymbol{z})\tilde{\rho}_t(\boldsymbol{x}|\boldsymbol{z})) = \frac{\nu^2}{2}\Delta_{\boldsymbol{x}}\tilde{\rho}_t(\boldsymbol{x}|\boldsymbol{z}) + \tilde{\rho}_t(\boldsymbol{x}|\boldsymbol{z})g_t(\boldsymbol{x}|\boldsymbol{z}) \quad (30)
$$

We further reorganize it to

$$
\partial_t \tilde{\rho}_t(\boldsymbol{x}|\boldsymbol{z}) + \nabla_{\boldsymbol{x}} \cdot (\boldsymbol{u}_t(\boldsymbol{x}|\boldsymbol{z})\tilde{\rho}_t(\boldsymbol{x}|\boldsymbol{z})) = \frac{\nu^2}{2}\Delta_{\boldsymbol{x}}\tilde{\rho}_t(\boldsymbol{x}|\boldsymbol{z}) + \tilde{\rho}_t(\boldsymbol{x}|\boldsymbol{z})\big(g_t(\boldsymbol{x}|\boldsymbol{z}) - \partial_t \ln m_t(\boldsymbol{z})\big) \quad (31)
$$

Set $g_t(\boldsymbol{x}|\boldsymbol{z}) = \partial_t \ln m_t(\boldsymbol{z})$, the equation of the density $\tilde{\rho}_t(\boldsymbol{x}|\boldsymbol{z})$ reduces to the Fokker-Planck equation

$$
\partial_t \tilde{\rho}_t(\boldsymbol{x}|\boldsymbol{z}) + \nabla_{\boldsymbol{x}} \cdot (\boldsymbol{u}_t(\boldsymbol{x}|\boldsymbol{z})\tilde{\rho}_t(\boldsymbol{x}|\boldsymbol{z})) = \frac{\nu^2}{2}\Delta_{\boldsymbol{x}}\tilde{\rho}_t(\boldsymbol{x}|\boldsymbol{z}) \quad (32)
$$

We then absorb the diffusion term into the drift term by introducing $\boldsymbol{u}_t^\circ(\boldsymbol{x}|\boldsymbol{z}) = \boldsymbol{u}_t(\boldsymbol{x}|\boldsymbol{z}) - \frac{\nu^2}{2}\boldsymbol{s}_t(\boldsymbol{x}|\boldsymbol{z})$

$$
\partial_t \tilde{\rho}_t(\boldsymbol{x}|\boldsymbol{z}) + \nabla_{\boldsymbol{x}} \cdot \big(\boldsymbol{u}_t^\circ(\boldsymbol{x}|\boldsymbol{z})\tilde{\rho}_t(\boldsymbol{x}|\boldsymbol{z})\big) = 0 \quad (33)
$$

Note that this equation is exactly the continuity equation of the conditional Gaussian path in conditional flow matching. Thus, the conditional vector field of CGMP shares the same form with the conditional vector field of conditional Gaussian path in balanced conditional flow matching, which is $\boldsymbol{u}_t^\circ(x|z) = \frac{\sigma_t'(\boldsymbol{z})}{\sigma_t(\boldsymbol{z})}\big(\boldsymbol{x} - \boldsymbol{\eta}_t(\boldsymbol{z})\big) + \boldsymbol{\eta}_t'(\boldsymbol{z})$.

The conditional score function of CGMP is just the score function of a family of Gaussian distributions

$$
\begin{aligned}
\boldsymbol{s}_t(\boldsymbol{x}|\boldsymbol{z}) &= \nabla_{\boldsymbol{x}} \ln \rho_t(\boldsymbol{x}|\boldsymbol{z}) \\
&\overset{(1)}{=} \nabla_{\boldsymbol{x}} \ln \tilde{\rho}_t(\boldsymbol{x}|\boldsymbol{z}) \\
&\overset{(2)}{=} \nabla_{\boldsymbol{x}} \ln \mathcal{N}(\boldsymbol{x}|\boldsymbol{\eta}_t(\boldsymbol{z}), \sigma_t^2(\boldsymbol{z})\mathrm{I}) \\
&= -\frac{\boldsymbol{x} - \boldsymbol{\eta}_t(\boldsymbol{z})}{\sigma_t^2(\boldsymbol{z})}
\end{aligned}
$$

In (1), note that the mass term $m_t(\boldsymbol{z})$ is independent of $\boldsymbol{x}$, thus the score function of $\rho_t$ and $\tilde{\rho}_t$ are equal. In (2), we use the definition of CGMP $\tilde{\rho}_t(\boldsymbol{x}|\boldsymbol{z}) = \mathcal{N}(\boldsymbol{x}|\boldsymbol{\eta}_t(\boldsymbol{z}), \sigma_t^2(\boldsymbol{z})\mathrm{I})$ (19).

The proof is completed. □

## A.3. Proof of Theorem 4.3

**Theorem 4.3.** *If $\rho_t(\boldsymbol{x}) > 0$ for all $\boldsymbol{x} \in \mathcal{X}$ and $t \in [0,1]$, and $q(\boldsymbol{z})$ is independent of $\boldsymbol{x}$ and $t$, then $\mathcal{L}_{\mathrm{USM}}(\boldsymbol{\theta}) = \mathcal{L}_{\mathrm{CUSM}}(\boldsymbol{\theta}) + C$, where $C$ is independent of $\boldsymbol{\theta}$. Thus they have identical gradients w.r.t $\boldsymbol{\theta}$, i.e.*

$$\nabla_{\boldsymbol{\theta}} \mathcal{L}_{\mathrm{USM}}(\boldsymbol{\theta}) = \nabla_{\boldsymbol{\theta}} \mathcal{L}_{\mathrm{CUSM}}(\boldsymbol{\theta}).$$

*Proof.* Recall the two objectives

$$\mathcal{L}_{\mathrm{USM}}(\boldsymbol{\theta}) = \int_0^1 \int_{\mathcal{X}} (\|\boldsymbol{v}_{\boldsymbol{\theta}}(\boldsymbol{x},t) - \boldsymbol{u}_t^\circ(\boldsymbol{x})\|_2^2 + \|g_{\boldsymbol{\theta}}(\boldsymbol{x},t) - g_t(\boldsymbol{x})\|_2^2 + \lambda^2(t)\|\boldsymbol{s}_{\boldsymbol{\theta}}(\boldsymbol{x},t) - \boldsymbol{s}_t(\boldsymbol{x})\|_2^2)\rho_t(\boldsymbol{x})\mathrm{d}\boldsymbol{x}\mathrm{d}t$$

$$\mathcal{L}_{\mathrm{CUSM}}(\boldsymbol{\theta}) = \mathbb{E}_{t\sim\mathcal{U}[0,1],\boldsymbol{z}\sim q(\boldsymbol{z}),\boldsymbol{x}\sim\tilde{\rho}_t(\boldsymbol{x}|\boldsymbol{z})} m_t(\boldsymbol{z})(\|\boldsymbol{v}_{\boldsymbol{\theta}}(\boldsymbol{x},t) - \boldsymbol{u}_t^\circ(\boldsymbol{x}|\boldsymbol{z})\|_2^2 + \|g_{\boldsymbol{\theta}}(\boldsymbol{x},t) - g_t(\boldsymbol{x}|\boldsymbol{z})\|_2^2 + \lambda^2(t)\|\boldsymbol{s}_{\boldsymbol{\theta}}(\boldsymbol{x},t) - \boldsymbol{s}_t(\boldsymbol{x}|\boldsymbol{z})\|_2^2)$$

By direct calculation,

$$
\begin{aligned}
\mathcal{L}_{\mathrm{USM}}(\boldsymbol{\theta}) &= \mathbb{E}_{t\sim\mathcal{U}[0,1]} \int_{\mathcal{X}} \Big( \|\boldsymbol{v}_{\boldsymbol{\theta}}(\boldsymbol{x},t)\|_2^2 - 2\langle\boldsymbol{v}_{\boldsymbol{\theta}}(\boldsymbol{x},t),\boldsymbol{u}_t^\circ(\boldsymbol{x})\rangle + \|g_{\boldsymbol{\theta}}(\boldsymbol{x},t)\|_2^2 - 2\langle g_{\boldsymbol{\theta}}(\boldsymbol{x},t),g_t(\boldsymbol{x})\rangle \\
&\qquad + \lambda^2(t)\big(\|\boldsymbol{s}_{\boldsymbol{\theta}}(\boldsymbol{x},t)\|_2^2 - 2\langle\boldsymbol{s}_{\boldsymbol{\theta}}(\boldsymbol{x},t),\boldsymbol{s}_t(\boldsymbol{x})\rangle\big)\Big)\rho_t(\boldsymbol{x})\mathrm{d}\boldsymbol{x} + C. \\
&\overset{(1)}{=} \mathbb{E}_{t\sim\mathcal{U}[0,1]} \int_{\mathcal{X}} \int \Big( \|\boldsymbol{v}_{\boldsymbol{\theta}}(\boldsymbol{x},t)\|_2^2 - 2\langle\boldsymbol{v}_{\boldsymbol{\theta}}(\boldsymbol{x},t),\boldsymbol{u}_t^\circ(\boldsymbol{x}|\boldsymbol{z})\rangle + \|g_{\boldsymbol{\theta}}(\boldsymbol{x},t)\|_2^2 - 2\langle g_{\boldsymbol{\theta}}(\boldsymbol{x},t),g_t(\boldsymbol{x}|\boldsymbol{z})\rangle \\
&\qquad + \lambda^2(t)\big(\|\boldsymbol{s}_{\boldsymbol{\theta}}(\boldsymbol{x},t)\|_2^2 - 2\langle\boldsymbol{s}_{\boldsymbol{\theta}}(\boldsymbol{x},t),\boldsymbol{s}_t(\boldsymbol{x}|\boldsymbol{z})\rangle\big)\Big)m_t(\boldsymbol{z})\tilde{\rho}_t(\boldsymbol{x}|\boldsymbol{z})q(\boldsymbol{z})\,\mathrm{d}\boldsymbol{z}\,\mathrm{d}\boldsymbol{x} + C \\
&= \mathbb{E}_{t\sim\mathcal{U}[0,1]} \int_{\mathcal{X}} \int \Big( \|\boldsymbol{v}_{\boldsymbol{\theta}}(\boldsymbol{x},t) - \boldsymbol{u}_t^\circ(\boldsymbol{x}|\boldsymbol{z})\|_2^2 + \|g_{\boldsymbol{\theta}}(\boldsymbol{x},t) - g_t(\boldsymbol{x}|\boldsymbol{z})\|_2^2 \\
&\qquad + \lambda^2(t)\|\boldsymbol{s}_{\boldsymbol{\theta}}(\boldsymbol{x},t) - \boldsymbol{s}_t(\boldsymbol{x}|\boldsymbol{z})\|_2^2 \Big)m_t(\boldsymbol{z})\tilde{\rho}_t(\boldsymbol{x}|\boldsymbol{z})q(\boldsymbol{z})\mathrm{d}\boldsymbol{z}\,\mathrm{d}\boldsymbol{x} + C \\
&= \mathbb{E}_{t\sim\mathcal{U}[0,1],\boldsymbol{z}\sim q(\boldsymbol{z}),\boldsymbol{x}\sim\tilde{\rho}_t(\boldsymbol{x}|\boldsymbol{z})} m_t(\boldsymbol{z})\Big( \|\boldsymbol{v}_{\boldsymbol{\theta}}(\boldsymbol{x},t) - \boldsymbol{u}_t^\circ(\boldsymbol{x}|\boldsymbol{z})\|_2^2 + \|g_{\boldsymbol{\theta}}(\boldsymbol{x},t) - g_t(\boldsymbol{x}|\boldsymbol{z})\|_2^2 \\
&\qquad + \lambda^2(t)\|\boldsymbol{s}_{\boldsymbol{\theta}}(\boldsymbol{x},t) - \boldsymbol{s}_t(\boldsymbol{x}|\boldsymbol{z})\|_2^2 \Big) + C \\
&= \mathcal{L}_{\mathrm{CUSM}}(\boldsymbol{\theta}) + C
\end{aligned}
$$

where $C$ is independent of $\boldsymbol{\theta}$. In (1), we use the relation between conditionals and marginals (18, 4.1). Since the constant $C$ is independent of $\boldsymbol{\theta}$, we have

$$\nabla_{\boldsymbol{\theta}} \mathcal{L}_{\mathrm{USM}}(\boldsymbol{\theta}) = \nabla_{\boldsymbol{\theta}} \mathcal{L}_{\mathrm{CUSM}}(\boldsymbol{\theta})$$

The proof is completed. □

## A.4. Proof of Proposition 5.1

**Proposition 5.1.** *The solution to the multi-time RUOT problem (27) is equivalent to the concatenation of the solutions of successive time points.*

*Proof. (adopted from (Peng et al., 2026) with modification)* Let $t_0 < t_1 < \cdots < t_K$ be the discrete time points. For each $k = 1, \ldots, K$, denote by $(\rho^{(k)}(\boldsymbol{x},t), \boldsymbol{u}^{(k)}(\boldsymbol{x},t), g^{(k)}(\boldsymbol{x},t))$ the optimal solution of the two-time RUOT problem between $\mu_{t_{k-1}}$ and $\mu_{t_k}$:

$$\mathrm{RUOT}(\mu_{t_{k-1}},\mu_{t_k}) = (t_k - t_{k-1}) \inf_{\rho,u,g} \int_{t_{k-1}}^{t_k} \int_{\mathcal{X}} \Big(\frac{1}{2}\|\boldsymbol{u}(\boldsymbol{x},t)\|_2^2 + \Psi(g(\boldsymbol{x},t))\Big)\rho(\boldsymbol{x},t)\,d\boldsymbol{x}\,dt \tag{34}$$

subject to

$$\partial_t\rho + \nabla_{\boldsymbol{x}} \cdot (\rho\boldsymbol{u}) = \frac{\nu^2}{2}\Delta_{\boldsymbol{x}}\rho + \rho g, \quad \rho_{t_{k-1}} = \mu_{t_{k-1}}, \ \rho_{t_k} = \mu_{t_k}. \tag{35}$$

Now define the concatenated trajectory

$$\tilde{\rho}(\boldsymbol{x},t) = \rho^{(k)}(\boldsymbol{x},t), \quad \tilde{\boldsymbol{u}}(\boldsymbol{x},t) = \boldsymbol{u}^{(k)}(\boldsymbol{x},t), \quad \tilde{g}(\boldsymbol{x},t) = g^{(k)}(\boldsymbol{x},t), \quad \text{for } t \in [t_{k-1},t_k].$$

Since $\rho_{t_k}^{(k)} = \mu_{t_k} = \rho_{t_k}^{(k+1)}$, the concatenated triple $(\tilde{\rho}, \tilde{\boldsymbol{u}}, \tilde{g})$ is admissible for the multi-time RUOT problem.

The corresponding cost is

$$
\begin{aligned}
(t_K - t_0) \int_{t_0}^{t_K} & \int_{\mathcal{X}} \Big( \frac{1}{2} \|\tilde{\boldsymbol{u}}(\boldsymbol{x}, t)\|_2^2 + \Psi(\tilde{g}(\boldsymbol{x}, t)) \Big) \tilde{\rho}(\boldsymbol{x}, t) \, d\boldsymbol{x} \, dt \\
& = \sum_{k=1}^{K} \frac{t_K - t_0}{t_k - t_{k-1}} \operatorname{RUOT}(\mu_{t_{k-1}}, \mu_{t_k}).
\end{aligned}
\tag{36}
$$

Hence

$$
\operatorname{RUOT}(\{\mu_{t_0}, \ldots, \mu_{t_K}\}) \leq \sum_{k=1}^{K} \frac{t_K - t_0}{t_k - t_{k-1}} \operatorname{RUOT}(\mu_{t_{k-1}}, \mu_{t_k}).
\tag{37}
$$

To see that the equality must hold, assume by contradiction that there exists another admissible solution $(\rho', \boldsymbol{u}', g')$ such that

$$
(t_K - t_0) \int_{t_0}^{t_K} \int_{\mathcal{X}} \Big( \frac{1}{2} \|\boldsymbol{u}'(\boldsymbol{x}, t)\|_2^2 + \Psi(g'(\boldsymbol{x}, t)) \Big) \rho'(\boldsymbol{x}, t) \, d\boldsymbol{x} \, dt < \sum_{k=1}^{K} \frac{t_K - t_0}{t_k - t_{k-1}} \operatorname{RUOT}(\mu_{t_{k-1}}, \mu_{t_k}).
$$

Since the total cost is a sum over the disjoint intervals $[t_{k-1}, t_k]$, this strict inequality implies that there must exist at least one index $k^*$ such that

$$
(t_K - t_0) \int_{t_{k^*-1}}^{t_{k^*}} \int_{\mathcal{X}} \Big( \frac{1}{2} \|\boldsymbol{u}'(\boldsymbol{x}, t)\|_2^2 + \Psi(g'(\boldsymbol{x}, t)) \Big) \rho'(\boldsymbol{x}, t) \, d\boldsymbol{x} \, dt < \frac{t_K - t_0}{t_{k^*} - t_{k^*-1}} \operatorname{RUOT}(\mu_{t_{k^*-1}}, \mu_{t_{k^*}}).
$$

Dividing both sides by the positive factor $\frac{t_K - t_0}{t_{k^*} - t_{k^*-1}}$ gives

$$
(t_{k^*} - t_{k^*-1}) \int_{t_{k^*-1}}^{t_{k^*}} \int_{\mathcal{X}} \Big( \frac{1}{2} \|\boldsymbol{u}'(\boldsymbol{x}, t)\|_2^2 + \Psi(g'(\boldsymbol{x}, t)) \Big) \rho'(\boldsymbol{x}, t) \, d\boldsymbol{x} \, dt < \operatorname{RUOT}(\mu_{t_{k^*-1}}, \mu_{t_{k^*}}).
$$

But this contradicts the definition of $\operatorname{RUOT}(\mu_{t_{k^*-1}}, \mu_{t_{k^*}})$ as the minimal cost between $\mu_{t_{k^*-1}}$ and $\mu_{t_{k^*}}$. Thus, no admissible solution can have strictly smaller cost than the concatenated one.

We conclude that the concatenated trajectory

$$
(\rho^*(x, t), \boldsymbol{u}^*(x, t), g^*(x, t)) = \bigcup_{k=1}^{K} (\rho^{(k)}, \boldsymbol{u}^{(k)}, g^{(k)}), \quad t \in [t_{k-1}, t_k],
\tag{38}
$$

achieves the infimum of the multi-time problem and is therefore optimal. The proof is completed. $\square$

**Remark**: This is true only when the dynamics between adjacent time pairs $(t_{k-1}, t_k)$ are independent. In multi-marginal methods, such as MMFM (Rohbeck et al., 2025), 3MSBM (Theodoropoulos et al., 2025) and MMSFM (Lee et al., 2025) where high order continuity or global connections between different time pairs are introduced, proposition (5.1) will never hold.

## B. Implementation details

### B.1. Weighting schedule $\lambda(t)$

For numerical stability, we use the same weighting schedule $\lambda(t) = \nu \sqrt{t(1-t)}$ as (Tong et al., 2024b) to normalize the score loss. With this weighting schedule, the regression target of the weighted score net $\lambda(t) \boldsymbol{s}_{\boldsymbol{\theta}}(\boldsymbol{x}, t)$ is simplified to a standard Gaussian noise $\boldsymbol{\epsilon}_t \sim \mathcal{N}(\boldsymbol{0}, \mathbf{I})$ instead of the conditional score function (24) including a division over $\nu^2 t(1-t)$ which may cause numerical instability when $t$ is near to $0, 1$.

$$
\begin{aligned}
\lambda^2(t) \|\boldsymbol{s}_{\boldsymbol{\theta}}(\boldsymbol{x}, t) - \boldsymbol{s}_t(\boldsymbol{x}|\boldsymbol{z})\|_2^2 & = \left\| \lambda(t) \boldsymbol{s}_{\boldsymbol{\theta}}(\boldsymbol{x}, t) - \frac{\boldsymbol{\eta}_t - \boldsymbol{x}}{\nu \sqrt{t(1-t)}} \right\|_2^2 \\
& \overset{(1)}{=} \|\lambda(t) \boldsymbol{s}_{\boldsymbol{\theta}}(\boldsymbol{x}, t) + \boldsymbol{\epsilon}_t\|_2^2
\end{aligned}
\tag{39}
$$

In (1), note that $\boldsymbol{x} \sim \mathcal{N}(\boldsymbol{\eta}_t, \nu^2 t(1-t)\mathbf{I})$, hence $\frac{\boldsymbol{x} - \boldsymbol{\eta}_t}{\nu \sqrt{t(1-t)}} = \boldsymbol{\epsilon}_t \sim \mathcal{N}(\boldsymbol{0}, \mathbf{I})$.

## B.2. Poisson-Brownian bridge

Since BBM is a branching process, it generates multiple end points from single start point. Thus, it is impossible to find a conditional BBM trajectory given its start point and end point. One can only try to find a conditional process in weak sense, i.e. find a conditional measure path of BBM. According to the evolution equation of BBM (9), we know that BBM undergoes diffusion spatially, while following exponential growth in mass. At a microscopic level, if we approximate that all cells share the same exponential splitting clock, then the logarithm of the total cell count is a Poisson process, the limit of which recovers this exponential growth. For numerical tractability, in what follows we determine the conditional path for this weighted particle approximation.

Given two Diracs $m_0 \delta_{\boldsymbol{x}_0}, m_1 \delta_{\boldsymbol{x}_1}$, the stochastic process bridging $\boldsymbol{x}_0, \boldsymbol{x}_1$ is a Brownian motion fixed on two sides, i.e. a Brownian bridge.

$$\mathcal{N}(t\boldsymbol{x}_1 + (1-t)\boldsymbol{x}_0, \nu^2 t(1-t)\mathrm{I}) \tag{40}$$

The stochastic process bridging $\ln m_0, \ln m_1$ is a Poisson process fixed on two sides, i.e. a Poisson bridge (Conforti et al., 2015). In cases where initial count and final count are two natural numbers $N_0 \leq N_1$, the Poisson bridge between them is

$$\mathcal{P}(t) \sim N_0 + Bernoulli(N_1 - N_0, t) \tag{41}$$

Now, consider adjusting the unit of mass from 1 to $\Delta m$, so that the counts $N_0$ and $N_1$ become total masses $M_0 = \Delta m N_0$ and $M_1 = \Delta m N_1$, respectively. Consequently, the Poisson bridge connecting these states is

$$\mathcal{P}(t) \sim M_0 + \Delta m \cdot Bernoulli(N_1 - N_0, t) \tag{42}$$

When $N_1 - N_0$ is large, one utilize the central limit theorem (CLT)

$$
\begin{aligned}
\mathcal{P}(t) &\approx M_0 + \Delta m \cdot \mathcal{N}((N_1 - N_0)t, (N_1 - N_0)t(1-t)) \\
&= M_0 + \mathcal{N}(\Delta m(N_1 - N_0)t, \Delta m^2(N_1 - N_0)t(1-t)) \\
&= \mathcal{N}(tM_1 + (1-t)M_0, \Delta m(M_1 - M_0)t(1-t)) \\
&= \mathcal{N}(tM_1 + (1-t)M_0, O(\Delta m))
\end{aligned}
\tag{43}
$$

Let $\Delta m \to 0$, we finally recover the linear mass variation on log scale.

$$
\begin{aligned}
\mathcal{P}(t) &\approx \mathcal{N}(tM_1 + (1-t)M_0, O(\Delta m)) \\
&\to \delta_{tM_1 + (1-t)M_0} \quad (\Delta m \to 0)
\end{aligned}
\tag{44}
$$

We point out that this approximation is justified due to the large sample size of single-cell sequencing data. Therefore, given two Dirac measures (representing the precise states at the start and endpoints), we connect their positions using a Brownian bridge, and linearly connect their masses on log scale. We term this the Poisson-Brownian bridge (23).

## B.3. Approximating RUOT semi-coupling

In a word, we approximate the RUOT semi-coupling by a static WFR semi-coupling which can be easily constructed from the coupling of an optimal entropy-transport problem (OET), and the OET problem can be easily solved using the POT package (Flamary et al., 2021).

In order to find the semi-coupling induced by the USB problem (8), we approximate it by solving its relaxation – RUOT (10). When the referencing BBM has diffusion parameter $\nu$, branching rate $\lambda$, and branching mechanism $\boldsymbol{p}_0 = \boldsymbol{p}_2 = \frac{1}{2}, \boldsymbol{p}_k = 0, k \neq 0, 2$, the corresponding RUOT is

$$\mathrm{RUOT}(\mu_0, \mu_1) = \inf_{\rho, \boldsymbol{u}, g} \int_0^1 \int_{\mathcal{X}} \left( \frac{1}{2} \|\boldsymbol{u}(\boldsymbol{x}, t)\|_2^2 + \Psi(g(\boldsymbol{x}, t)) \right) \rho(\boldsymbol{x}, t) \, d\boldsymbol{x} \, dt \tag{45}$$

$$\text{s.t.} \qquad \partial_t \rho + \nabla_{\boldsymbol{x}} \cdot (\rho \boldsymbol{u}) = \frac{\nu^2}{2} \Delta_{\boldsymbol{x}} \rho + \rho g, \; \rho_0 = \mu_0, \; \rho_1 = \mu_1$$

with growth penalty $\Psi_{\nu,\lambda}(g)$ determined by its Legendre transform (Baradat & Lavenant, 2021)

$$\Psi_{\nu,\lambda}^*(g) = \nu^2 \lambda (\cosh(\frac{g}{\nu^2}) - 1) \tag{46}$$

By applying Legendre transform to $\Psi^\star$, we obtain the explicit form of the growth penalty.

$$\Psi(g) = 1 - \sqrt{1+g^2} + g\ln(\sqrt{1+g^2}+g)$$
$$\Psi_{\nu,\lambda}(g) = \nu^2\lambda\Psi(\frac{g}{\lambda}) \tag{47}$$

Since it is hard to find a static form for the RUOT problem or solve it with such a growth penalty, we approximate it by expanding the growth penalty to second order.

$$\Psi(g) = \frac{1}{2}g^2 + o(g^3)$$
$$\Psi_{\nu,\lambda}(g) = \frac{\nu^2}{2\lambda}g^2 + o(g^3) \tag{48}$$

Doing so, we approximate the original RUOT problem with a WFR problem which is easy to solve.

$$\mathrm{RUOT}(\mu_0,\mu_1) \approx \mathrm{WFR}_\delta^2(\mu_0,\mu_1) = \inf_{\rho,\boldsymbol{u},g} \frac{1}{2}\int_0^1\int_{\mathcal{X}} \left(\|\boldsymbol{u}(\boldsymbol{x},t)\|_2^2 + \delta^2|g(\boldsymbol{x},t)|^2\right)\rho(\boldsymbol{x},t)\,d\boldsymbol{x}\,dt$$
$$= 2\delta^2 \inf_{(\gamma_0,\gamma_1)\in(\mathcal{M}_+(\mathcal{X}^2))^2} \int_{\mathcal{X}^2} \left(\gamma_0(\boldsymbol{x},\boldsymbol{y}) + \gamma_1(\boldsymbol{x},\boldsymbol{y}) - 2\sqrt{\gamma_0(\boldsymbol{x},\boldsymbol{y})\gamma_1(\boldsymbol{x},\boldsymbol{y})}\,\overline{\cos}(\frac{\|\boldsymbol{x}-\boldsymbol{y}\|_2}{2\delta})\right)d\boldsymbol{x}d\boldsymbol{y} \tag{49}$$

where $\delta = \nu/\sqrt{\lambda}$, $\overline{\cos}(x) = \cos(\min\{x,0\})$. The second row is known as the static WFR problem which is a minimization problem respect to the semi-coupling $(\gamma_0,\gamma_1)$ instead of $(\rho,\boldsymbol{u},g)$. To solve the static WFR problem, (Chizat et al., 2018b; Liero et al., 2018) proved that the static WFR problem is equivalent to the following optimal entropy-transport (OET) problem.

$$\mathrm{OET}_\delta(\mu_0,\mu_1) = 2\delta^2 \inf_{\gamma\in\mathcal{M}_+(\mathcal{X}^2)} \left\{ \int_{\mathcal{X}^2} -2\ln\overline{\cos}(\frac{\|\boldsymbol{x}-\boldsymbol{y}\|_2}{2\delta})\gamma(\boldsymbol{x},\boldsymbol{y})d\boldsymbol{x}d\boldsymbol{y} \right.$$
$$\left. + \mathrm{KL}(\int_{\mathcal{X}}\gamma(\boldsymbol{x},\boldsymbol{y})d\boldsymbol{y}\|\mu_0(\boldsymbol{x})) + \mathrm{KL}(\int_{\mathcal{X}}\gamma(\boldsymbol{x},\boldsymbol{y})d\boldsymbol{x}\|\mu_1(\boldsymbol{y})) \right\} \tag{50}$$

An OET problem can be easily solved using the generalized Sinkhorn-Knopp matrix scaling algorithm (Chizat et al., 2017) which is well implemented in the POT package (Flamary et al., 2021). To construct the semi-coupling $(\gamma_0,\gamma_1)$ from the OET coupling $\gamma$, we follow the results of (Liero et al., 2018; Peng et al., 2026).

**Theorem B.1.** *(Peng et al., 2026) Let $\gamma$ be the optimal coupling of the OET problem (50), then the semi-coupling $\gamma_0(\boldsymbol{x},\boldsymbol{y}) = \frac{\gamma(\boldsymbol{x},\boldsymbol{y})}{\int_{\mathcal{X}}\gamma(\boldsymbol{x},\boldsymbol{z})d\boldsymbol{z}}\mu_0(\boldsymbol{x}), \gamma_1(\boldsymbol{x},\boldsymbol{y}) = \frac{\gamma(\boldsymbol{x},\boldsymbol{y})}{\int_{\mathcal{X}}\gamma(\boldsymbol{z},\boldsymbol{y})d\boldsymbol{z}}\mu_1(\boldsymbol{y})$ solves the static WFR problem (4).*

We summarize the workflow for the calculation of the semi-coupling $(\gamma_0,\gamma_1)$ as following.

---

**Algorithm 2** Semi-coupling calculation Workflow

---

**Require:** Sample-able distributions $\mu_0,\mu_1$, diffusion parameter $\nu$, branching rate $\lambda$.
1: $\delta = \nu/\sqrt{\lambda}$
2: $\gamma \leftarrow \mathrm{OET}_\delta(\mu_0,\mu_1)$ (Defined in 4)
3: $\gamma_0(\boldsymbol{x},\boldsymbol{y}) \leftarrow \frac{\gamma(\boldsymbol{x},\boldsymbol{y})}{\int_{\mathcal{X}}\gamma(\boldsymbol{x},\boldsymbol{z})d\boldsymbol{z}}\mu_0(\boldsymbol{x}), \gamma_1(\boldsymbol{x},\boldsymbol{y}) \leftarrow \frac{\gamma(\boldsymbol{x},\boldsymbol{y})}{\int_{\mathcal{X}}\gamma(\boldsymbol{z},\boldsymbol{y})d\boldsymbol{z}}\mu_1(\boldsymbol{y})$ (Theorem B.1)
4: **return** $(\gamma_0,\gamma_1)$

---

One can determine the triplet $(\nu,\lambda,\delta)$ by knowing any two of them. We point out that, when fixing the stochastic reference $\nu$, the WFR growth penalty coefficient $\delta$ also has a intuitive meaning of growth reference. Since $\delta = \nu/\sqrt{\lambda}$, the smaller the growth penalty coefficient $\delta$, the larger the growth reference $\lambda$, and vice versa. Thus, for convenience, we implicitly choose the branching rate $\lambda$ by giving $\nu$ and $\delta$. The performance of the approximation is studied in (C.11).

**Remark.** As mentioned above, some RUOT problem e.g. WFR do have an equivalent OET formulation which is easy to solve (Chizat et al., 2018a;b). We tried to follow their approach, but failed. A discussion on their approach and the difficulties here can be found in Appendix E.

## B.4. Branching inference

Branching inference aims to simulate the discrete single-cell birth-death dynamics at single-cell resolution. We start from a cell at $\boldsymbol{x}_0$ whose position evolves according to the marginal SDE.

$$\mathrm{d}\boldsymbol{x}_t = (\boldsymbol{v}_\theta(\boldsymbol{x}_t, t) + \frac{\nu^2}{2}\boldsymbol{s}_\theta(\boldsymbol{x}_t, t))\mathrm{d}t + \nu\mathrm{d}\boldsymbol{W}_t \tag{51}$$

All SDE simulations were implemented using Euler-Maruyama scheme, and all ODE simulations were implemented using forward Euler scheme. Detailed inference workflows can be found in Appendix F.

We also simulates a non-homogeneous branching process to decide when the cell divides or dies. According to the evolution equation of BBM (9), the relation between the learned growth rate $g_\theta(\boldsymbol{x}, t)$ and the corresponding branching rate $\lambda$, offspring distribution $\boldsymbol{p}$ at $(\boldsymbol{x}, t)$ is

$$g_{\boldsymbol{\theta}}(\boldsymbol{x}, t) = \lambda(\boldsymbol{x}, t)(\boldsymbol{p}_2(\boldsymbol{x}, t) - \boldsymbol{p}_0(\boldsymbol{x}, t)) \tag{52}$$

For simplicity, we assume that stochasticity in cells is solely confined to changes in gene expression, while their apoptosis or division is entirely determined by gene expression and time; that is, the offspring distribution $\boldsymbol{p}(\boldsymbol{x}, t)$ is a one-hot vector given $(\boldsymbol{x}, t)$. Hence, we have

$$\begin{cases} \lambda(\boldsymbol{x}, t) = |g_{\boldsymbol{\theta}}(\boldsymbol{x}, t)| \\ \boldsymbol{p}_2 = 1_{g_{\boldsymbol{\theta}}(\boldsymbol{x}, t) \geq 0} \\ \boldsymbol{p}_0 = 1_{g_{\boldsymbol{\theta}}(\boldsymbol{x}, t) < 0} \\ \boldsymbol{p}_k = 0, \quad k \neq 0, 2 \end{cases} \tag{53}$$

where $1_A$ is the indicator function of event $A$. Based on the above, we linked the learned growth rates to the branching dynamics. We simulate the branching event of each cell as a non-homogeneous jump process. In details, for each SDE simulation time step $\Delta t$, the probability of a cell branching within that step is given by $p = |g_\theta(\boldsymbol{x}, t)|\Delta t$. Therefore, we sample a random variable $\alpha \sim \mathcal{U}[0, 1]$. If $\alpha \leq p$, branching occurs, otherwise it does not. When branching occurs: if $g_\theta(\boldsymbol{x}, t) \geq 0$, the cell divides into two, and the resulting daughter cells are subsequently simulated independently according to these aforementioned rules; else, the current cell dies.

We further point out that the mean mass increase of continuous inference and branching inference on log-scale are the same. In any time step $\Delta t$, the log mass increment is $g_\theta(\boldsymbol{x}, t)\Delta t$, while the mean log mass increment of branching inference is

$$\begin{aligned} \lambda(\boldsymbol{x}, t)\Delta t(\boldsymbol{p}_2 - \boldsymbol{p}_0) &= |g_{\boldsymbol{\theta}}(\boldsymbol{x}, t)|\Delta t(1_{g_{\boldsymbol{\theta}}(\boldsymbol{x}, t) \geq 0} - 1_{g_{\boldsymbol{\theta}}(\boldsymbol{x}, t) < 0}) \\ &= |g_{\boldsymbol{\theta}}(\boldsymbol{x}, t)|\Delta t \cdot \mathrm{sgn}(g_{\boldsymbol{\theta}}(\boldsymbol{x}, t)) \\ &= g_{\boldsymbol{\theta}}(\boldsymbol{x}, t)\Delta t \end{aligned}$$

which is the same. Also, since continuous inference and branching inference follow the same SDE spatially, they generate the same measure in weak sense.

# C. Additional results

In this section, we provide a detailed description of the computational resources (C.1), metrics (C.2), and datasets used. We further carry out sensitivity analyses for the growth penalization $\delta$ (C.3), diffusion parameter $\nu$ (C.4), and the mini-batch OT (C.7,C.9), and assess the algorithm's scalability with respect to the number of cells (C.9) and the dimensionality (C.7). In (C.10), we compare the computational cost of USB against current SOTA OT-based algorithms. In (C.11), we study the performance of the WFR approximation.

## C.1. Experimental Details

The experiments were performed on a shared high-performance computing cluster with NVIDIA H800 GPU and 128 CPU cores. The architecture of the neural networks used to parameterize $\boldsymbol{v}_\theta(\boldsymbol{x}, t)$, $g_\theta(\boldsymbol{x}, t)$ and $\boldsymbol{s}_\theta(\boldsymbol{x}, t)$ are Multilayer Perceptrons with 256 hidden channels and 5 layers. We use the LeakyReLU activation function. These networks were optimized using Adam (Kingma & Ba, 2015), and implemented using Pytorch. The OT problems are solved using the Python Optimal Transport (POT) package (Flamary et al., 2021).

## C.2. Evaluation Metrics

To evaluate model performance on measure reconstruction, we decoupled the task into two parts: distribution-matching and mass-matching. Two measures $p, q$ are said to be the same, if and only if they have the same normalized distributions $\tilde{p}, \tilde{q}$, and same total mass $m_p, m_q$, since $p = m_p\tilde{p}, q = m_q\tilde{q}$. In the followings, let $p = m_p\tilde{p}$ be the true measure, and $q = m_q\tilde{q}$ be the predicted measure.

We use the 1-Wasserstein distance ($\mathcal{W}_1$) to measure the similarity between normalized predicted distribution $\tilde{q}$ and true distribution $\tilde{p}$.

$$\mathcal{W}_1(\tilde{p}, \tilde{q}) = \min_{\pi \in \Pi(\tilde{p}, \tilde{q})} \int \|\boldsymbol{x} - \boldsymbol{y}\|_2 d\pi(\boldsymbol{x}, \boldsymbol{y}) \tag{54}$$

where

$$\Pi(\tilde{p}, \tilde{q}) = \{\pi(\boldsymbol{x}, \boldsymbol{y}) \in \mathcal{M}_+(\mathcal{X}^2)| \int_{\mathcal{X}} \pi(\boldsymbol{x}, \boldsymbol{y})\mathrm{d}\boldsymbol{y} = \tilde{p}(\boldsymbol{x}), \int_{\mathcal{X}} \pi(\boldsymbol{x}, \boldsymbol{y})\mathrm{d}\boldsymbol{x} = \tilde{q}(\boldsymbol{y})\}$$

In practice, $\tilde{p}, \tilde{q}$ are empirical distributions consist or weighted particles. The above minimization problem can be easily solved by linear program or Sinkhorn algorithm using POT package.

We use Relative Mass Error (RME) to assess how well the model captures cell population growth. The RME is defined as

$$\text{RME}(p, q) = \frac{|m_p - m_q|}{m_q} \tag{55}$$

In previous literature such as (Sha et al., 2024; Zhang et al., 2025; Wang et al., 2025; Peng et al., 2026), RME is formulated as

$$\text{RME}(t_k) = \frac{|\sum_i w_i(t_k) - n_k/n_0|}{n_k/n_0}$$

where $w_i(t_k)$ is the inferred weight of cell $i$ at time $t_k$, and $n_k$ is the number of cells at time $k$. One can easily check the two formulations are the same.

To generate predicted measures at time $t_k$, we start form the true measure at $t_0$. Each cell is assigned with a mass weight $w_i(0) = 1$. For continuous inference, each cell travels following the learned SDE dynamics, and its weight evolves following the learned birth-death ODE. For branching inference, only the position of each cell follows the SDE, the weight stay invariant, while the cell may vanish or get a copy at the branching event. At $t_k$, the true measure is consists of $n_k$ cells with weight 1 from the dataset, and the predicted measure is consists of weighted cells which are still alive at $t_k$. The total mass is defined as the sum of weights (also the cell number for branching inference). For some datasets, we also perform a hold-out experiment. The model was trained on all but one time point and evaluated on that time point. All simulations were started at the initial time point $t_0$.

To compare USB with other methods, we implement several existed methods on all datasets. For WFR-FM (Peng et al., 2026) and VGFM (Wang et al., 2025), we used their default hyperparameter settings, since the datasets evaluated in our work are also used in their work, and our model sizes are consistent. For branchSBM (Tang et al., 2025), we follow the hyperparameter settings in Table 9 of their paper. For other simulation-free methods (MMFM (Rohbeck et al., 2025), Metric FM (Kapuśniak et al., 2024), SF2M (Tong et al., 2024b), UOT-FM (Eyring et al., 2024)), we trained 3000 epochs with cosine annealing learning rate decay on high-dimensional real datasets (Mouse, Cite, EB50D, EB100D), 1000 epochs with constant learning rate on other datasets. For simulation-based methods (MIOFlow (Huguet et al., 2022), TIGON (Sha et al., 2024) and DeepRUOT (Zhang et al., 2025), Var-RUOT (Sun et al., 2025)), we follow the hyperparameter setting recommended by DeepRUOT, which is the SOTA in simulation-based methods.

## C.3. Simulation Gene Data

The Simulation Gene data is adopted from (Zhang et al., 2025). It simulates a synthetic gene regulatory network via the following equations

$$\frac{dX_1}{dt} = \frac{\alpha_1 X_1^2 + \beta}{1 + \alpha_1 X_1^2 + \gamma_2 X_2^2 + \gamma_3 X_3^2 + \beta} - \delta_1 X_1 + \eta_1 \xi_t$$

$$\frac{dX_2}{dt} = \frac{\alpha_2 X_2^2 + \beta}{1 + \gamma_1 X_1^2 + \alpha_2 X_2^2 + \gamma_3 X_3^2 + \beta} - \delta_2 X_2 + \eta_2 \xi_t$$

$$\frac{dX_3}{dt} = \frac{\alpha_3 X_3^2}{1 + \alpha_3 X_3^2} - \delta_3 X_3 + \eta_3 \xi_t$$

where $X_i$ represents the concentration of gene $i$. The model features a toggle switch system (mutual inhibition and self-activation) between $X_1$ and $X_2$ with inhibition by $X_3$ and activation by an external signal $\beta$. The self-activation rate, inhibition rate, and degradation rate of each gene are denoted as $\alpha_i$, $\gamma_i$, and $\delta_i$, respectively. Also, noise terms $(\eta_i \xi_t)$ are added to each governing ODE to introduce stochasticity. The growth is introduced by probabilistic cell division with probability $g = \alpha_g \frac{X_2^2}{1+X_2^2}\%$. Upon division, each child inherits the gene expression level of its parent, and adds an independent noise $\eta_d \mathcal{N}(0,1)$ to each gene.

The data is generated by simulating the governing equations from two source. One is located at a steady states of the system with no growth, thus exhibits equilibrium during time. The other one exhibits both transition and growth. The data is recorded at five time points $[0, 8, 16, 24, 32]$.

**Sensitivity study of growth penalty $\delta$.** When referencing on a BBM with diffusion parameter $\nu$ and branching rate $\lambda$, we have to solve a RUOT problem (10) to get the semi-coupling. We approximate it by solving a WFR problem with growth penalty $\delta = \nu/\sqrt{\lambda}$ (B.3). The parameter $\delta$ explicitly represents the ratio of the strength of transition and growth. With excessively high $\delta$, cells tend to transit rather than grow. Changes in the relative mass between the two trajectories will not be realized through population growth. Instead, it will be compensated by incorrect transfers between cell states, causing the model to learn an incorrect vector field and underlying dynamics (Fig.4 left panel). With excessively low $\delta$, cells tend to grow rather than transit. When the measure is transported between time points, it tends to first decrease the mass and then increase it after the transfer is completed. Errors in the vector field can be compensated by the unpanelized large growth rate, causing the model to learn an incorrect vector field and dynamics (Fig.4 right panel). To balance the mutually compensating effects of transfer and growth, we chose a moderate $\delta = 1.3$ in the main text to penalize the growth, thereby learning the correct vector field and underlying dynamics (Fig.4 middle panel). Sensitivity analysis results for different $\delta$ are shown in Table 5. Both an excessively high or low $\delta$ ($\delta = 0.5, 2.5$) results in a sub-optimal performance, while $\delta$ in a proper range (1 to 2) exhibit comparable performance, demonstrating the robustness of USB to $\delta$.

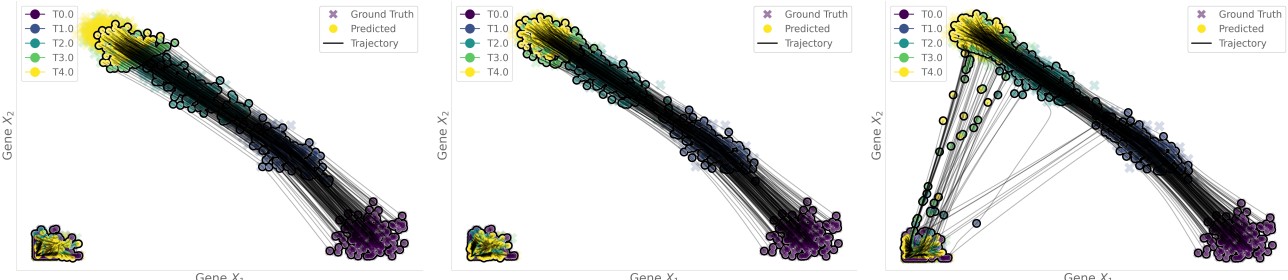

*Figure 4.* Learned trajectories on the Simulation Gene dataset ($\nu = 0.001$). From left to right: $\delta = 0.5, 1.3, 2.5$.

*Table 5.* Sensitivity analysis for parameter $\delta$ on simulation gene dataset.

| Parameter | t=1 | | t=2 | | t=3 | | t=4 | |
|---|---|---|---|---|---|---|---|---|
| | $\mathcal{W}_1$ | RME | $\mathcal{W}_1$ | RME | $\mathcal{W}_1$ | RME | $\mathcal{W}_1$ | RME |
| $\delta = 0.5$ | $0.033_{\pm 6 \times 10^{-5}}$ | $0.020_{\pm 6 \times 10^{-5}}$ | $0.078_{\pm 1 \times 10^{-4}}$ | $0.073_{\pm 1 \times 10^{-4}}$ | $0.096_{\pm 2 \times 10^{-4}}$ | $0.112_{\pm 2 \times 10^{-4}}$ | $0.086_{\pm 2 \times 10^{-4}}$ | $0.130_{\pm 2 \times 10^{-4}}$ |
| $\delta = 1.0$ | $0.022_{\pm 5 \times 10^{-5}}$ | $0.003_{\pm 5 \times 10^{-5}}$ | $0.026_{\pm 3 \times 10^{-5}}$ | $0.006_{\pm 1 \times 10^{-5}}$ | $0.024_{\pm 5 \times 10^{-5}}$ | $0.008_{\pm 2 \times 10^{-5}}$ | $0.022_{\pm 5 \times 10^{-5}}$ | $0.011_{\pm 2 \times 10^{-5}}$ |
| $\delta = 1.3$ | $0.020_{\pm 4 \times 10^{-5}}$ | $0.000_{\pm 2 \times 10^{-5}}$ | $0.021_{\pm 2 \times 10^{-5}}$ | $0.001_{\pm 1 \times 10^{-5}}$ | $0.019_{\pm 1 \times 10^{-5}}$ | $0.002_{\pm 2 \times 10^{-5}}$ | $0.018_{\pm 2 \times 10^{-5}}$ | $0.003_{\pm 3 \times 10^{-5}}$ |
| $\delta = 1.5$ | $0.022_{\pm 5 \times 10^{-5}}$ | $0.002_{\pm 4 \times 10^{-6}}$ | $0.029_{\pm 7 \times 10^{-5}}$ | $0.006_{\pm 5 \times 10^{-6}}$ | $0.026_{\pm 8 \times 10^{-5}}$ | $0.009_{\pm 6 \times 10^{-6}}$ | $0.025_{\pm 3 \times 10^{-5}}$ | $0.009_{\pm 7 \times 10^{-6}}$ |
| $\delta = 1.7$ | $0.025_{\pm 5 \times 10^{-5}}$ | $0.001_{\pm 3 \times 10^{-6}}$ | $0.027_{\pm 8 \times 10^{-5}}$ | $0.004_{\pm 5 \times 10^{-6}}$ | $0.022_{\pm 7 \times 10^{-5}}$ | $0.006_{\pm 8 \times 10^{-6}}$ | $0.021_{\pm 1 \times 10^{-4}}$ | $0.009_{\pm 9 \times 10^{-6}}$ |
| $\delta = 2.0$ | $0.022_{\pm 2 \times 10^{-5}}$ | $0.000_{\pm 3 \times 10^{-6}}$ | $0.026_{\pm 2 \times 10^{-4}}$ | $0.000_{\pm 2 \times 10^{-5}}$ | $0.034_{\pm 6 \times 10^{-4}}$ | $0.002_{\pm 9 \times 10^{-5}}$ | $0.047_{\pm 3 \times 10^{-4}}$ | $0.007_{\pm 1 \times 10^{-4}}$ |
| $\delta = 2.5$ | $0.023_{\pm 3 \times 10^{-5}}$ | $0.000_{\pm 6 \times 10^{-6}}$ | $0.041_{\pm 0.002}$ | $0.004_{\pm 2 \times 10^{-4}}$ | $0.029_{\pm 6 \times 10^{-4}}$ | $0.007_{\pm 4 \times 10^{-4}}$ | $0.026_{\pm 7 \times 10^{-4}}$ | $0.008_{\pm 4 \times 10^{-4}}$ |

## C.4. Dyngen Data

We adopt the same dataset as in (Huguet et al., 2022; Wang et al., 2025). It was simulated by Dyngen (Cannoodt et al., 2021). The dataset contains 728 cells, znd was reduced to 5 dimensions using PHATE (Moon et al., 2019). The data

exhibits a complex dynamics which contains both varying mass across time points and bifurcation. The bifurcation is also unbalanced in some sense that the cell number in the lower branch is larger than the upper branch. Due to the complexity, it is a challenging task to recover its dynamics. USB accurately recovers the bifurcating trajectories, and predicts a higher growth rate in the lower branch, capturing the unbalanced bifurcation dynamics (Fig.5). In the main text, we set $\delta = 1.7$.

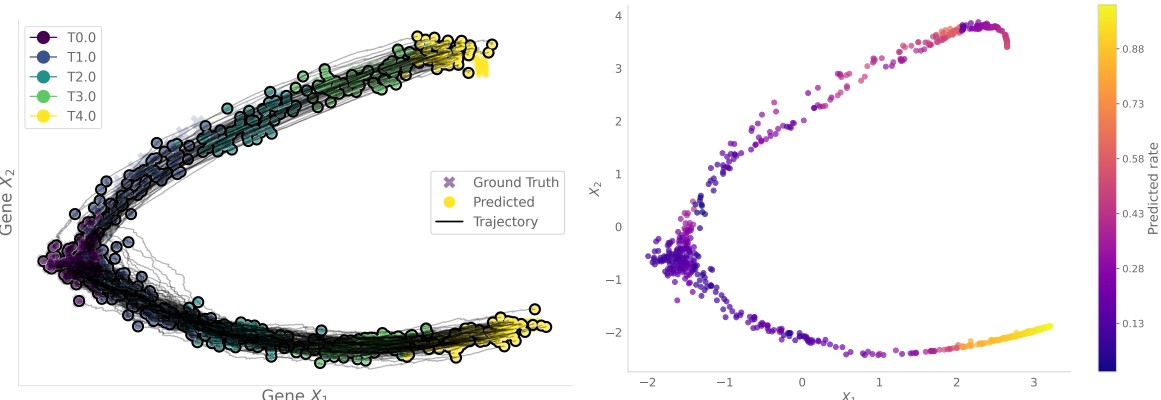

*Figure 5.* Learned trajectories and growth on the Dyngen dataset ($\delta = 1.7, \nu = 0.1$). Left panel: trajectories; Right panel: growth rate.

We also summarized the detailed quantitative results across different time points in Table 6. As shown, USB achieves the top-2 lowest $\mathcal{W}_1$ and RME in all of the time points.

**Sensitivity study of diffusion parameter $\nu$.** We further provide a sensitivity analysis for the diffusion parameter $\nu$ in the bottom of Table 6. As the level pf noise becomes larger, the accuracy of the measure transportation decreases. But, we point out that even for a extremely large noise ($\nu = 0.1$), the $\mathcal{W}_1$ distance and RME of USB remains the second lowest in most experiments, demonstrating the robustness and power of USB. In the main text, if there is no specific notice, we set $\nu = 0.001$.

**Remark.** When setting $\nu = 0.5$, the training diverges, thus $\nu = 0.1$ is a extremely large noise level for Dyngen data.

*Table 6.* Comparison of method performance over time on the Dyngen dataset and sensitivity analysis for parameter $\nu$. Best results are in bold, and the second best are underlined (**results of $\nu = 0.01, 0.05, 0.1$ are not compared**).

| Method | t=1 | | t=2 | | t=3 | | t=4 | |
|---|---|---|---|---|---|---|---|---|
| | $\mathcal{W}_1$ | RME | $\mathcal{W}_1$ | RME | $\mathcal{W}_1$ | RME | $\mathcal{W}_1$ | RME |
| MMFM | 0.574 | — | 1.704 | — | 1.499 | — | 1.706 | — |
| Metric FM | 0.892 | — | 2.347 | — | 2.030 | — | 1.799 | — |
| SF2M | 0.637 | — | 1.266 | — | 1.415 | — | 1.790 | — |
| MIOFlow | 0.420 | — | 0.640 | — | 1.537 | — | 1.263 | — |
| BranchSBM | 0.926 | — | 1.171 | — | 2.081 | — | 1.481 | — |
| TIGON | 0.446 | 0.033 | 0.584 | 0.060 | 0.415 | 0.023 | 0.603 | 0.071 |
| DeepRUOT | 0.454 | 0.011 | 0.481 | 0.070 | 0.870 | 0.104 | 0.688 | 0.074 |
| Var-RUOT | 0.315 | 0.128 | 0.548 | 0.336 | 0.630 | 0.222 | 0.593 | 0.023 |
| UOT-FM | 0.652 | 0.008 | 0.780 | 0.077 | 1.252 | 0.090 | 2.130 | 0.213 |
| VGFM | 0.335 | **0.001** | 0.312 | 0.073 | 1.109 | 0.041 | 0.634 | 0.033 |
| WFR-FM | 0.110 | 0.003 | 0.098 | 0.007 | 0.211 | **0.008** | **0.121** | **0.002** |
| **USB**$_{\nu = 0.001}$ | **0.109**$_{\pm 0.0002}$ | 0.002$_{\pm 0.0000}$ | **0.093**$_{\pm 0.0002}$ | **0.002**$_{\pm 0.0001}$ | **0.180**$_{\pm 0.0013}$ | 0.015$_{\pm 0.0004}$ | 0.142$_{\pm 0.0040}$ | 0.008$_{\pm 0.0008}$ |
| **USB**$_{\nu = 0.01}$ | 0.136$_{\pm 0.001}$ | 0.010$_{\pm 0.0005}$ | 0.144$_{\pm 0.004}$ | 0.005$_{\pm 0.0008}$ | 0.247$_{\pm 0.014}$ | 0.006$_{\pm 0.005}$ | 0.290$_{\pm 0.020}$ | 0.004$_{\pm 0.003}$ |
| **USB**$_{\nu = 0.05}$ | 0.169$_{\pm 0.006}$ | 0.006$_{\pm 0.002}$ | 0.206$_{\pm 0.010}$ | 0.018$_{\pm 0.004}$ | 0.359$_{\pm 0.059}$ | 0.031$_{\pm 0.014}$ | 0.302$_{\pm 0.059}$ | 0.009$_{\pm 0.006}$ |
| **USB**$_{\nu = 0.1}$ | 0.218$_{\pm 0.009}$ | 0.008$_{\pm 0.003}$ | 0.261$_{\pm 0.012}$ | 0.029$_{\pm 0.007}$ | 0.496$_{\pm 0.076}$ | 0.042$_{\pm 0.025}$ | 0.357$_{\pm 0.047}$ | 0.072$_{\pm 0.024}$ |

## C.5. Gaussian Data

We adopt the 1000D Gaussian Mixture Data from (Wang et al., 2025). Following their setup, at the initial time point, there are 500 cells from 2 Gaussians, 100 from the upper Gaussian, and 400 from the lower Gaussian. The upper Gaussian remains

stationary and undergoes pure growth, whereas the lower Gaussian exhibits no growth and instead bifurcates symmetrically into left and right Gaussians. At the terminal time point, the total cell count is 1,400: the upper component increases from 100 to 1,000 cells, and the lower component splits evenly into two 200-cell Gaussians. Since the 1000D Gaussian has 1000 dimensions, it is suitable for testing USB's performance in high dimensional spaces. As shown in Fig.6 and Fig.1, USB faithfully recovers the underlying dynamics. In the main text, we set $\delta = 1.4, \nu = 0.001$.

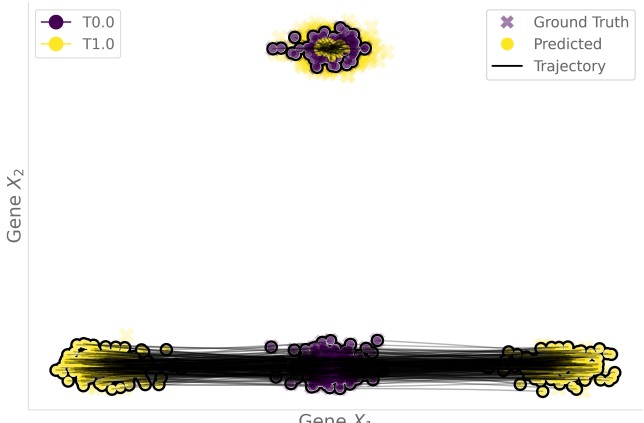

*Figure 6.* Learned trajectories on the Gaussian 1000D dataset ($\delta = 1.3, \nu = 0.001$).

### C.6. Epithelial Mesenchymal Transition Data

We adopt the dataset that captures the epithelial-mesenchymal transition (EMT) in A549 lung cancer cells from (Cook & Vanderhyden, 2020). The dataset includes samples collected at four distinct time points throughout this process. (Sha et al., 2024) reduced the dimension to 10 by an autoencoder. We applied USB on the reduced 10D EMT data. As shown in Table 7, USB achieves the best performance at most time points with growth penalty $\delta = 14$, and diffusion parameter $\nu = 0.001$. We plotted the learned growth rate in Figure 7. As shown, USB predicts higher growth rate in initial and intermediate stages, which is in line with the results predicted by DeepRUOT (Zhang et al., 2025) and WFR-FM (Peng et al., 2026) as these cells exhibit enhanced stemness.

*Table 7.* Comparison of method performance over time on the 10D EMT dataset.

| Method | t=1 | | t=2 | | t=3 | |
|---|---|---|---|---|---|---|
| | $\mathcal{W}_1$ | RME | $\mathcal{W}_1$ | RME | $\mathcal{W}_1$ | RME |
| MMFM | 0.2576 | — | 0.2874 | — | 0.3102 | — |
| Metric FM | 0.2605 | — | 0.2971 | — | 0.3050 | — |
| SF2M | 0.2566 | — | 0.2811 | — | 0.2900 | — |
| MIOFlow | 0.2439 | — | 0.2665 | — | 0.2841 | — |
| BranchSBM | 0.3287 | — | 0.3757 | — | 0.2894 | — |
| TIGON | 0.2433 | 0.002 | 0.2661 | 0.003 | 0.2847 | **0.001** |
| DeepRUOT | 0.2902 | 0.001 | 0.3193 | 0.011 | 0.3291 | 0.002 |
| Var-RUOT | 0.2540 | 0.075 | 0.2670 | 0.014 | 0.2683 | 0.041 |
| UOT-FM | 0.2538 | 0.002 | 0.2696 | 0.013 | 0.2771 | 0.010 |
| VGFM | 0.2350 | 0.016 | 0.2420 | 0.011 | 0.2450 | 0.018 |
| WFR-FM | 0.2099 | 0.001 | 0.2272 | **0.002** | 0.2346 | **0.001** |
| **USB** | **0.1831**±0.0000 | **0.000**±0.0000 | **0.2159**±0.0002 | 0.009±0.0000 | **0.2309**±0.0001 | 0.009±0.0000 |

### C.7. Embryoid Bodies Data

We adopt the EB dataset from (Moon et al., 2019), comprising 16,819 cells collected at five time points over a 27-day differentiation course of human embryoid bodies (EBs), an experimental model of early embryonic development. The

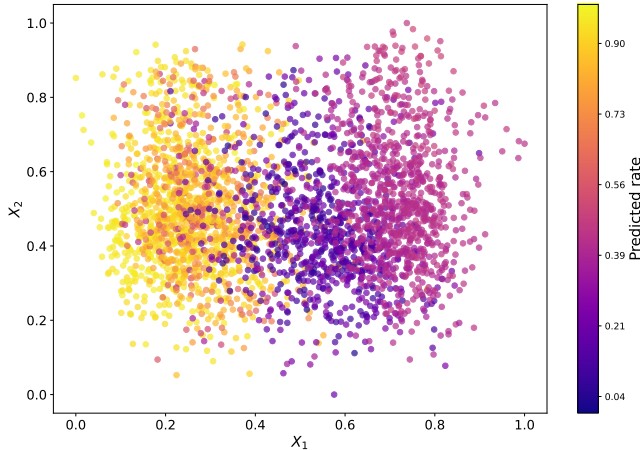

*Figure 7.* Learned growth rate on the EMT dataset ($\delta = 1.4, \nu = 0.001$).

data was preprocessed via Principal Component Analysis (PCA) as in (Wang et al., 2025). We kept the first 100 principal components for downstream analyses.

**Scalability with respect to the dimensionality.** We evaluate USB on EB data with first 5, 50, 100 PCs to test the scalability of USB w.r.t the dimensionality. We set $\delta = 3, 27, 30$ for 5D, 50D, 100D, respectively, and set $\nu = 0.001$. To be consistent to (Peng et al., 2026), the 5D EB is further standardized. We compare the performance of USB with several other methods on the 3 datasets with results in showed Table 8,9,10, respectively. The computation time is recorded in Table 11. Across all experiments, USB achieved performance comparable to the best baseline with no significant computational overhead, indicating that USB scales directly to high-dimensional datasets without compromising performance.

*Table 8.* Comparison of method performance over time on the 5D EB dataset. Best results are in bold, and the second best are underlined.

| Method | t=1 | | t=2 | | t=3 | | t=4 | |
|--------|-----|-----|-----|-----|-----|-----|-----|-----|
| | $\mathcal{W}_1$ | RME | $\mathcal{W}_1$ | RME | $\mathcal{W}_1$ | RME | $\mathcal{W}_1$ | RME |
| MMFM | 0.477 | — | 0.554 | — | 0.781 | — | 0.872 | — |
| Metric FM | 0.449 | — | 0.552 | — | 0.583 | — | 0.597 | — |
| SF2M | 0.556 | — | 0.715 | — | 0.750 | — | 0.650 | — |
| MIOFlow | 0.442 | — | 0.585 | — | 0.651 | — | 0.670 | — |
| BranchSBM | 0.864 | — | 1.355 | — | 1.211 | — | 1.626 | — |
| TIGON | 0.386 | **0.002** | 0.502 | 0.015 | 0.602 | 0.021 | 0.600 | 0.027 |
| DeepRUOT | 0.386 | 0.005 | 0.497 | 0.017 | 0.591 | 0.021 | 0.585 | 0.030 |
| Var-RUOT | 0.416 | 0.111 | 0.486 | 0.144 | 0.509 | 0.054 | 0.511 | 0.022 |
| UOT-FM | 0.544 | 0.032 | 0.670 | 0.029 | 0.729 | 0.016 | 0.852 | 0.041 |
| VGFM | 0.402 | 0.046 | 0.494 | 0.018 | 0.525 | 0.035 | 0.573 | 0.021 |
| WFR-FM | **0.324** | 0.003 | **0.401** | **0.001** | 0.431 | **0.005** | 0.510 | **0.005** |
| **USB** | 0.331$_{\pm 2 \times 10^{-5}}$ | 0.006$_{\pm 2 \times 10^{-6}}$ | 0.408$_{\pm 2 \times 10^{-5}}$ | 0.023$_{\pm 4 \times 10^{-6}}$ | **0.427$_{\pm 2 \times 10^{-5}}$** | 0.018$_{\pm 6 \times 10^{-6}}$ | **0.450$_{\pm 4 \times 10^{-5}}$** | 0.019$_{\pm 1 \times 10^{-5}}$ |

**Sensitivity analysis for mini-batch OT.** For larger datasets, full-batch OT can require substantial memory and long computation time. To avoid these costs, one can use mini-batch OT (Tong et al., 2024a; Fatras et al., 2021). To show that USB is compatible with this strategy—and thus has the potential to handle huge datasets—we evaluated different mini-batch sizes on the EB (100D) and Mouse (C.9) datasets. Results are summarized in Table 12 and Table 15. On EB (100D), we observe a non-monotonic dependence on batch size: using either small batches (500) or full-batch OT leads to slightly inferior performance and longer computation time, while a moderate batch size performs better and runs faster. Nevertheless, the overall impact of batch size is modest, and USB remains on par with the strongest baseline even with small batch size. This indicates that USB can naturally pairs with mini-batch OT and scales to huge datasets without losing performance. For consistency with the baseline using mini-batch OT (WFR-FM) and a favorable accuracy–efficiency balance, we use a mini-batch size of 2,000 on EB (100D), Cite, and Mouse datasets.

*Table 9.* Comparison of method performance over time on the 50D EB dataset. Best results are in bold, and the second best are underlined.

| Method | t=1 | | t=2 | | t=3 | | t=4 | |
|---|---|---|---|---|---|---|---|---|
| | $\mathcal{W}_1$ | RME | $\mathcal{W}_1$ | RME | $\mathcal{W}_1$ | RME | $\mathcal{W}_1$ | RME |
| MMFM | 9.124 | — | 10.474 | — | 11.022 | — | 11.480 | — |
| Metric FM | 8.506 | — | 9.795 | — | 10.621 | — | 12.042 | — |
| SF2M | 9.247 | — | 10.882 | — | 11.650 | — | 12.154 | — |
| MIOFlow | 8.447 | — | 9.229 | — | 9.436 | — | 10.123 | — |
| BranchSBM | 11.085 | — | 12.359 | — | 12.353 | — | 10.813 | — |
| TIGON | 8.433 | 0.067 | 9.275 | 0.022 | 9.802 | 0.179 | 10.148 | 0.101 |
| DeepRUOT | 8.169 | **0.003** | 9.049 | 0.038 | 9.378 | 0.088 | 9.733 | **0.004** |
| Var-RUOT | 9.442 | 0.128 | 9.709 | 0.081 | 10.482 | 0.031 | 10.735 | 0.030 |
| UOT-FM | 8.717 | 0.063 | 10.858 | 0.009 | 11.813 | 0.022 | 12.733 | 0.018 |
| VGFM | 7.951 | 0.089 | 8.747 | 0.042 | 9.244 | 0.019 | 9.620 | 0.044 |
| WFR-FM | **7.664** | 0.008 | **8.659** | **0.006** | **9.182** | **0.004** | 9.914 | **0.004** |
| USB | $7.992_{\pm 2 \times 10^{-5}}$ | $\underline{0.008_{\pm 7 \times 10^{-7}}}$ | $8.973_{\pm 5 \times 10^{-5}}$ | $0.019_{\pm 1 \times 10^{-6}}$ | $9.421_{\pm 2 \times 10^{-5}}$ | $\underline{0.008_{\pm 2 \times 10^{-6}}}$ | $9.963_{\pm 2 \times 10^{-5}}$ | $\underline{0.017_{\pm 3 \times 10^{-6}}}$ |

*Table 10.* Comparison of method performance over time on the 100D EB dataset. Best results are in bold, and the second best are underlined.

| Method | t=1 | | t=2 | | t=3 | | t=4 | |
|---|---|---|---|---|---|---|---|---|
| | $\mathcal{W}_1$ | RME | $\mathcal{W}_1$ | RME | $\mathcal{W}_1$ | RME | $\mathcal{W}_1$ | RME |
| MMFM | 11.460 | — | 13.879 | — | 14.441 | — | 14.907 | — |
| Metric FM | 10.806 | — | 12.348 | — | 13.622 | — | 16.801 | — |
| SF2M | 11.333 | — | 12.982 | — | 13.718 | — | 14.945 | — |
| MIOFlow | 11.387 | — | 12.331 | — | 11.905 | — | 12.908 | — |
| BranchSBM | 12.853 | — | 14.298 | — | 14.419 | — | 13.718 | — |
| TIGON | 10.547 | 0.014 | 12.926 | 0.052 | 13.897 | 0.107 | 14.945 | 0.096 |
| DeepRUOT | 10.256 | **0.002** | 11.103 | 0.074 | 11.529 | 0.136 | **12.406** | 0.047 |
| Var-RUOT | 11.746 | 0.091 | 12.237 | 0.024 | 12.957 | 0.150 | 13.335 | 0.074 |
| UOT-FM | 10.757 | 0.056 | 12.799 | 0.037 | 13.761 | 0.044 | 15.657 | 0.022 |
| VGFM | 10.313 | 0.048 | 11.278 | 0.035 | 11.703 | 0.028 | 12.637 | 0.066 |
| WFR-FM | **9.941** | 0.009 | **11.040** | **0.006** | 11.516 | 0.008 | 12.664 | 0.005 |
| USB | $\underline{10.206_{\pm 5 \times 10^{-5}}}$ | $0.016_{\pm 5 \times 10^{-7}}$ | $11.297_{\pm 6 \times 10^{-5}}$ | $\underline{0.010_{\pm 8 \times 10^{-7}}}$ | $11.758_{\pm 9 \times 10^{-6}}$ | $\mathbf{0.002_{\pm 2 \times 10^{-6}}}$ | $12.888_{\pm 5 \times 10^{-5}}$ | $\mathbf{0.003_{\pm 3 \times 10^{-6}}}$ |

*Table 11.* Computational time across different dimensions on the EB dataset.

| Dimension | Time (s) |
|---|---|
| 5 | 31.36 |
| 50 | 30.15 |
| 100 | 30.67 |

*Table 12.* Sensitivity analysis for batch size of mini-batch WFR-OET on the 100D EB dataset.

| Batch Size | t=1 | | t=2 | | t=3 | | t=4 | | Time (s) |
|---|---|---|---|---|---|---|---|---|---|
| | $\mathcal{W}_1$ | RME | $\mathcal{W}_1$ | RME | $\mathcal{W}_1$ | RME | $\mathcal{W}_1$ | RME | |
| 500 | $10.238_{\pm 3 \times 10^{-5}}$ | $0.015_{\pm 4 \times 10^{-7}}$ | $11.360_{\pm 3 \times 10^{-5}}$ | $0.010_{\pm 4 \times 10^{-7}}$ | $11.877_{\pm 5 \times 10^{-5}}$ | $0.001_{\pm 1 \times 10^{-6}}$ | $12.958_{\pm 6 \times 10^{-5}}$ | $0.006_{\pm 2 \times 10^{-6}}$ | 30.33 |
| 1000 | $10.228_{\pm 2 \times 10^{-5}}$ | $0.014_{\pm 8 \times 10^{-7}}$ | $11.314_{\pm 6 \times 10^{-5}}$ | $0.008_{\pm 2 \times 10^{-6}}$ | $11.789_{\pm 6 \times 10^{-5}}$ | $0.005_{\pm 6 \times 10^{-5}}$ | $12.880_{\pm 2 \times 10^{-4}}$ | $0.001_{\pm 3 \times 10^{-6}}$ | 27.76 |
| 2000 | $10.206_{\pm 5 \times 10^{-5}}$ | $0.016_{\pm 5 \times 10^{-7}}$ | $11.297_{\pm 6 \times 10^{-5}}$ | $0.010_{\pm 8 \times 10^{-7}}$ | $11.758_{\pm 9 \times 10^{-6}}$ | $0.002_{\pm 2 \times 10^{-6}}$ | $12.888_{\pm 5 \times 10^{-5}}$ | $0.003_{\pm 3 \times 10^{-6}}$ | 27.26 |
| 3000 | $10.201_{\pm 4 \times 10^{-5}}$ | $0.015_{\pm 6 \times 10^{-7}}$ | $11.337_{\pm 3 \times 10^{-5}}$ | $0.010_{\pm 1 \times 10^{-6}}$ | $11.803_{\pm 4 \times 10^{-5}}$ | $0.004_{\pm 1 \times 10^{-6}}$ | $12.905_{\pm 5 \times 10^{-5}}$ | $0.001_{\pm 2 \times 10^{-6}}$ | 28.34 |
| 4000 | $10.189_{\pm 1 \times 10^{-5}}$ | $0.016_{\pm 9 \times 10^{-7}}$ | $11.306_{\pm 3 \times 10^{-5}}$ | $0.011_{\pm 1 \times 10^{-6}}$ | $11.756_{\pm 3 \times 10^{-5}}$ | $0.002_{\pm 6 \times 10^{-7}}$ | $12.816_{\pm 4 \times 10^{-5}}$ | $0.003_{\pm 1 \times 10^{-6}}$ | 28.42 |
| w/o mini-batch | $10.216_{\pm 2 \times 10^{-5}}$ | $0.014_{\pm 4 \times 10^{-7}}$ | $11.330_{\pm 5 \times 10^{-5}}$ | $0.011_{\pm 9 \times 10^{-7}}$ | $11.817_{\pm 3 \times 10^{-5}}$ | $0.000_{\pm 1 \times 10^{-6}}$ | $12.916_{\pm 7 \times 10^{-5}}$ | $0.005_{\pm 2 \times 10^{-6}}$ | 31.53 |

## C.8. Cite-seq Data

We adopt the CITE-seq (Cite) dataset from (Lance et al., 2022), consisting of 31,240 cells across four time points. Following the preprocessing in (Wang et al., 2025), we keep only the gene-expression modality and apply PCA to 50 dimensions. We use mini-batch OT (batch size 2,000) and set $\delta = 30, \nu = 0.001$. The performance on distribution matching and mass matching are summarized in Table 13. As shown, USB attains the highest accuracy in half of the experiments and remains comparable to the best baseline on the remainder.

*Table 13.* Comparison of method performance over time on the 50D CITE dataset. Best results are in bold, and the second best are underlined.

| Method | t=1 | | t=2 | | t=3 | |
|---|---|---|---|---|---|---|
| | $\mathcal{W}_1$ | RME | $\mathcal{W}_1$ | RME | $\mathcal{W}_1$ | RME |
| MMFM | 33.971 | — | 36.854 | — | 43.721 | — |
| Metric FM | 28.314 | — | 28.617 | — | 33.212 | — |
| SF2M | 29.543 | — | 32.655 | — | 36.265 | — |
| MIOFlow | 28.290 | — | 28.524 | — | 32.230 | — |
| BranchSBM | 39.908 | — | 37.008 | — | **31.883** | — |
| TIGON | 28.196 | 0.186 | 27.921 | 0.545 | 32.846 | 0.653 |
| DeepRUOT | 28.245 | 0.168 | 27.908 | 0.525 | 32.950 | 0.634 |
| Var-RUOT | 30.219 | 0.331 | 32.702 | 0.325 | 40.613 | 0.486 |
| UOT-FM | 33.531 | 0.009 | 32.795 | 0.046 | 49.751 | 0.097 |
| VGFM | 29.449 | 0.020 | 29.722 | 0.057 | 33.752 | **0.001** |
| WFR-FM | 27.831 | 0.043 | **27.478** | 0.045 | 34.784 | 0.022 |
| **USB** | **27.021**$_{\pm 5 \times 10^{-5}}$ | **0.002**$_{\pm 1 \times 10^{-6}}$ | 28.081$_{\pm 1 \times 10^{-4}}$ | **0.015**$_{\pm 2 \times 10^{-6}}$ | 38.447$_{\pm 0.003}$ | 0.010$_{\pm 5 \times 10^{-5}}$ |

## C.9. Mouse Hematopoiesis Data

We adopt the mouse blood hematopoiesis dataset (Mouse) from (Weinreb et al., 2020), selecting 49,302 lineage-traced cells measured at three time points. The gene-expression matrix were reduced to 50 dimensions using PCA. Owing to its substantial population growth and large cell number, it is well suited for scalability testing and USB. We use mini-bath OT of size 2000, and set $\delta = 15, \nu = 0.001$, the all time points training results is shown in Table 14. USB outperforms most baselines significantly except WFR-FM, and performs comparable to WFR-FM at all time points.

*Table 14.* Comparison of method performance over time on the 50D Mouse dataset. Best results are in bold, and the second best are underlined.

| Method | t=1 | | t=2 | |
|---|---|---|---|---|
| | $\mathcal{W}_1$ | RME | $\mathcal{W}_1$ | RME |
| MMFM | 7.647 | — | 10.156 | — |
| Metric FM | 7.788 | — | 11.449 | — |
| SF2M | 8.217 | — | 11.086 | — |
| MIOFlow | 6.313 | — | 6.746 | — |
| BranchSBM | 7.957 | — | 9.236 | — |
| TIGON | 6.140 | 0.382 | 6.973 | 0.326 |
| DeepRUOT | 6.052 | 0.062 | 6.757 | 0.041 |
| Var-RUOT | 7.951 | 0.131 | 10.862 | 0.154 |
| UOT-FM | 8.114 | 0.035 | 9.170 | 0.011 |
| VGFM | 6.274 | 0.076 | 6.796 | 0.070 |
| WFR-FM | **5.486** | **0.012** | **6.211** | 0.011 |
| **USB** | 5.589$_{\pm 1 \times 10^{-6}}$ | 0.013$_{\pm 6 \times 10^{-8}}$ | 6.548$_{\pm 4 \times 10^{-6}}$ | **0.008**$_{\pm 4 \times 10^{-7}}$ |

**Sensitivity analysis for mini-batch OT.** As mentioned in (C.7), we also test the sensitivity of the batch size of mini-batch OT on Mouse dataset. The result is shown in Table 15. The non-monotonic varying computation time is also observed. Intuitively, for small batch sizes, it take shorter time to compute each mini-batch OT, but the number of batch are larger, thus may also require more computation time. Hence, the computation time exhibits a U-shape variation according to batch size. On Mouse dataset, the $\mathcal{W}_1$ distance and RME decrease when using larger batch size while it slightly increase when using full-batch. The monotonicity is not observed on EB dataset. It may because of that EB is not large enough. Intuitively, the accuracy will be positively correlated to the batch size on sufficiently large datasets. Consistent to the results on EB, the performance and computation time show no significant difference across batch sizes which are not extremely small for Mouse ($>1000$).

*Table 15.* Sensitivity analysis for batch size of mini-batch WFR-OET on the 50D Mouse dataset.

| Batch Size | t=1 | | t=2 | | Time (s) |
|---|---|---|---|---|---|
| | $\mathcal{W}_1$ | RME | $\mathcal{W}_1$ | RME | |
| 500 | $5.644_{\pm 2 \times 10^{-6}}$ | $0.015_{\pm 1 \times 10^{-7}}$ | $6.854_{\pm 1 \times 10^{-5}}$ | $0.001_{\pm 5 \times 10^{-7}}$ | 307.73 |
| 1000 | $5.599_{\pm 4 \times 10^{-6}}$ | $0.014_{\pm 2 \times 10^{-7}}$ | $6.535_{\pm 5 \times 10^{-6}}$ | $0.006_{\pm 6 \times 10^{-7}}$ | 293.61 |
| 2000 | $5.589_{\pm 1 \times 10^{-6}}$ | $0.013_{\pm 6 \times 10^{-8}}$ | $6.548_{\pm 4 \times 10^{-6}}$ | $0.008_{\pm 4 \times 10^{-7}}$ | 292.28 |
| 3000 | $5.539_{\pm 2 \times 10^{-6}}$ | $0.011_{\pm 1 \times 10^{-7}}$ | $6.476_{\pm 7 \times 10^{-6}}$ | $0.007_{\pm 7 \times 10^{-7}}$ | 296.69 |
| 4000 | $5.492_{\pm 3 \times 10^{-6}}$ | $0.012_{\pm 1 \times 10^{-7}}$ | $6.416_{\pm 4 \times 10^{-6}}$ | $0.009_{\pm 4 \times 10^{-7}}$ | 300.50 |
| 5000 | $5.484_{\pm 1 \times 10^{-6}}$ | $0.010_{\pm 6 \times 10^{-8}}$ | $6.396_{\pm 2 \times 10^{-5}}$ | $0.001_{\pm 6 \times 10^{-7}}$ | 306.06 |
| 10000 | $5.502_{\pm 2 \times 10^{-6}}$ | $0.013_{\pm 1 \times 10^{-7}}$ | $6.397_{\pm 2 \times 10^{-6}}$ | $0.013_{\pm 7 \times 10^{-7}}$ | 313.60 |
| 20000 | $5.438_{\pm 1 \times 10^{-6}}$ | $0.009_{\pm 1 \times 10^{-7}}$ | $6.335_{\pm 4 \times 10^{-6}}$ | $0.006_{\pm 5 \times 10^{-7}}$ | 332.33 |
| w/o mini-batch | $5.527_{\pm 8 \times 10^{-7}}$ | $0.008_{\pm 1 \times 10^{-7}}$ | $6.383_{\pm 1 \times 10^{-5}}$ | $0.001_{\pm 5 \times 10^{-7}}$ | 331.22 |

**Scalability with respect to the cell number.** We subsampled the Mouse dataset to 10000, 15000, 20000, 25000, 30000, 35000, 40000, 45000 cells to test the scalability of USB w.r.t the size of the dataset. The training time on each subsampled dataset are scattered in Fig.8. The training time scales linearly with cell numbers, in line with other competing methods, indicating that USB can be applied to larger datasets.

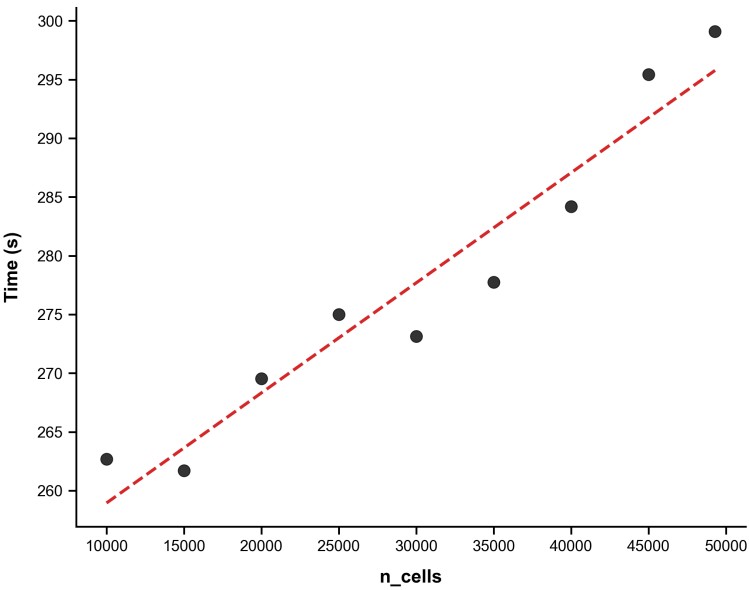

*Figure 8.* Training time v.s. cell numbers

## C.10. Computational cost comparison

We ran DeepRUOT, SF2M, WFR-FM, and USB on a single H200 GPU across varying cell numbers (10000, 20000, 30000, 40000, 49302) using the Mouse Hematopoiesis dataset. Hyperparameter setting: batch size=512, epochs=3000, mini-batch OT with chunk size=2000, networks are MLPs with 256 hidden channels and 5 layers. The runtime and memory results are summarized in Table 16.

Simulation-free algorithms outperform DeepRUOT in runtime. The extra time and memory cost of USB against SF2M and WFR is negligible. Since the memory of SF2M, WFR, USB are very similar, we present an analysis below.

*Table 16.* Runtime and memory results across varying cell numbers

| Metric | Method | N=10000 | N=20000 | N=30000 | N=40000 | N=49302 |
|---|---|---|---|---|---|---|
| **Runtime (s)** | DeepRUOT | 400.03 | 753.90 | 1100.51 | 1596.62 | 2007.85 |
| | WFR-FM | 27.95 | 33.36 | 42.17 | 54.18 | 67.80 |
| | SF2M | 28.35 | 33.84 | 42.83 | 54.74 | 68.78 |
| | **USB** | 29.40 | 35.39 | 44.26 | 56.21 | 70.10 |
| **Peak CPU RAM (MB)** | DeepRUOT | 137.07 | 457.16 | 1010.16 | 1769.52 | 2668.26 |
| | WFR-FM | 1121.32 | 4336.26 | 9681.68 | 17140.84 | 26032.36 |
| | SF2M | 1121.32 | 4336.26 | 9681.68 | 17140.84 | 26032.36 |
| | **USB** | 1121.32 | 4336.25 | 9681.68 | 17140.84 | 26032.36 |
| **Peak GPU VRAM (MB)** | DeepRUOT | 2742.58 | 5904.51 | 9515.35 | 13614.29 | 17586.63 |
| | WFR-FM | 242.21 | 708.72 | 444.44 | 717.98 | 522.33 |
| | SF2M | 242.21 | 708.82 | 444.54 | 718.08 | 522.43 |
| | **USB** | 242.21 | 713.05 | 448.77 | 722.31 | 526.65 |

**Shared OT Pre-computation.** As shown in the table, all three methods exhibit an identical peak CPU memory that grows quadratically with respect to $N$. It is because they share an OT calculation step which causes the dominating $\mathcal{O}(N^2)$ CPU memory peak.

**Small Extra Memory on GPU.** USB's peak VRAM is consistently $\sim 4.3$ MB higher than SF2M and WFR across all sample sizes. This extra $4.3$ MB difference corresponds to the parameter size of our additional neural network. SF2M uses a velocity net and a score net, WFR uses a velocity net and a growth net, while USB uses all 3 networks. The extra network has 5 hidden layers, each has 256 neurons. The total parameter number is about $256 \times (51 + 256 \times 4 + 51) \approx 290000$. We use float32 and Adam (memory $\approx 4 \times$ parameters), thus the expected extra memory is about $290000 \times 4$ Bytes $\times 4/1024^2 \approx 4.4$ MB, which is consistent with the reported extra memory.

These results show that USB allows for more complex dynamics modeling with negligible extra computational cost.

### C.11. WFR approximation

We approximate the intractable RUOT semi-coupling using the efficient WFR semi-coupling (B.3). We studied the performance of this approximation on small numerical examples (RUOT were optimized using scipy packages).

**Eg1. 1D Splitting.** An initial unit mass at $x = 0$ is transported to two targets at $x = -1$ and $x = 1$ (with target masses of $0.4$ and $1.6$, respectively). As shown in Figure 9, while the absolute costs differ, the normalized cost landscapes of WFR and RUOT are highly similar, achieving the exact same global minimum at $0.303$ (the optimal splitting proportion).

**Eg2. 2D unbalanced transport.** We transported mass from 5 source points (total mass 1.0) to 5 target points. The total target mass is scaled by an unbalance factor $\alpha \in \{0.7, 1.0, 1.3, 1.5, 2.0\}$, with individual target masses drawn from a Dirichlet distribution. In Figures 10 to 14, scatter plots comparing the matrix elements of WFR and RUOT show that the vast majority of points align tightly along the 45-degree diagonal ($y = x$). The accompanying heatmaps further visually confirm that the resulting optimal transport plans are very similar.

These results demonstrate that WFR is a highly accurate and efficient approximation of RUOT when the unbalance is moderate. Since real biological mass changes are typically not violently drastic between consecutive, reasonably sampled time points, assuming a moderate unbalance effect is biologically sound and justified.

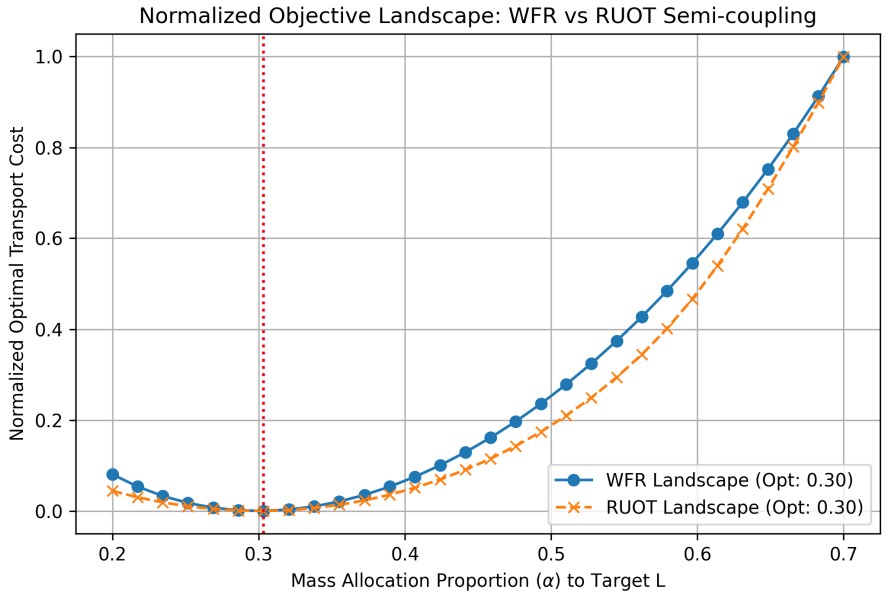

*Figure 9.* WFR approximation on 1D case.

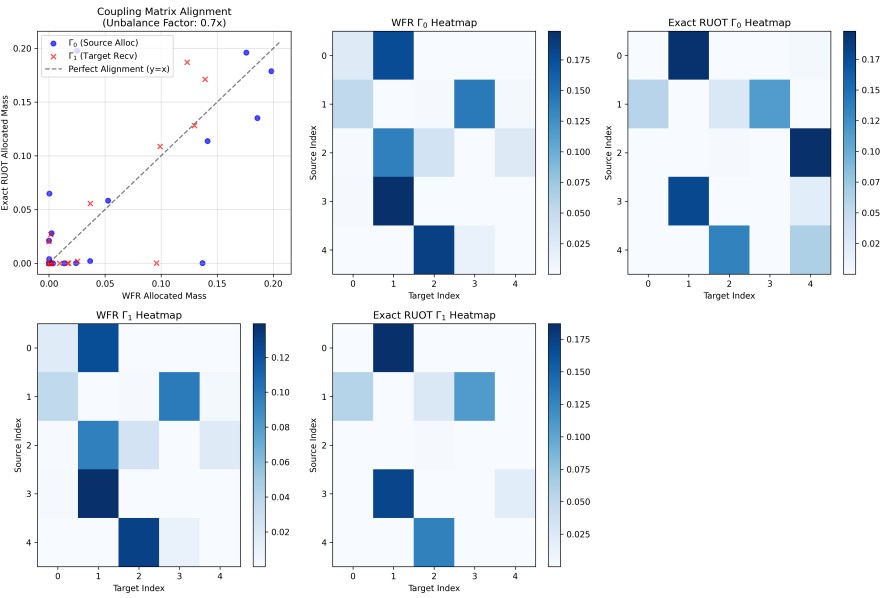

*Figure 10.* WFR approximation on 2D case ($\alpha = 0.7$).

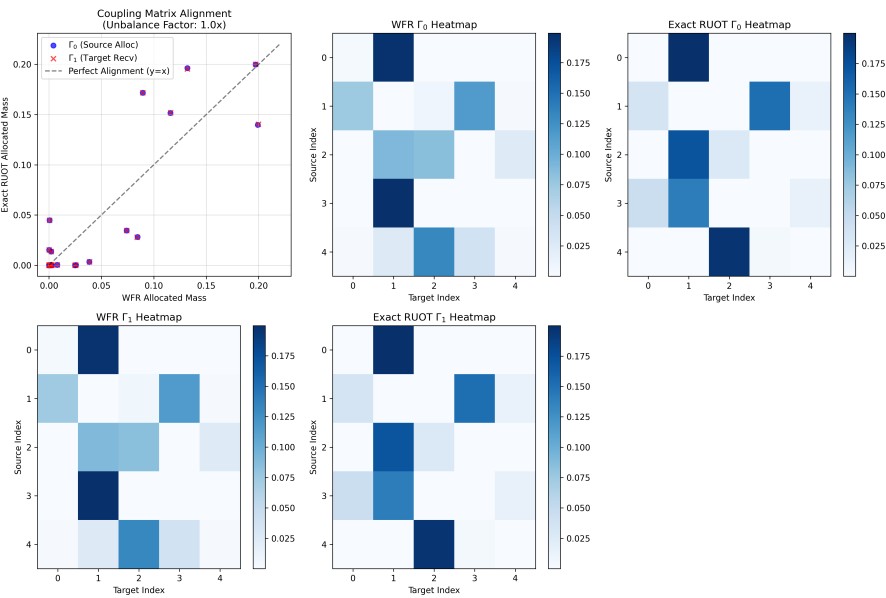

*Figure 11.* WFR approximation on 2D case ($\alpha = 1$).

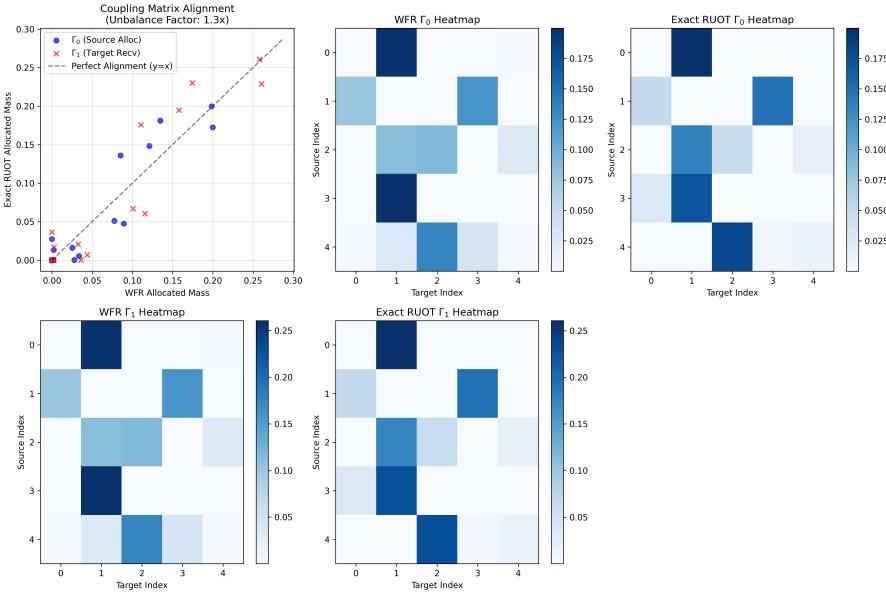

*Figure 12.* WFR approximation on 2D case ($\alpha = 1.3$).

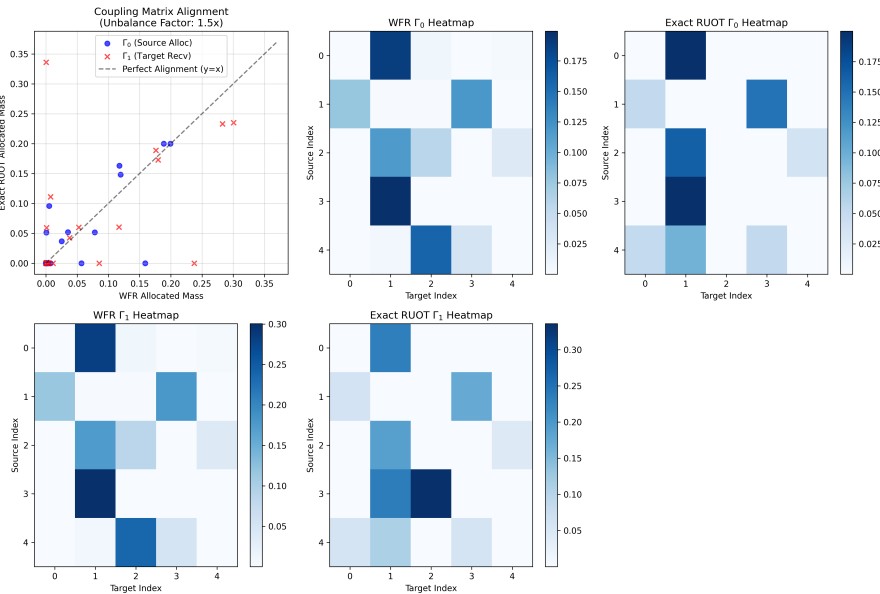

*Figure 13.* WFR approximation on 2D case ($\alpha = 1.5$).

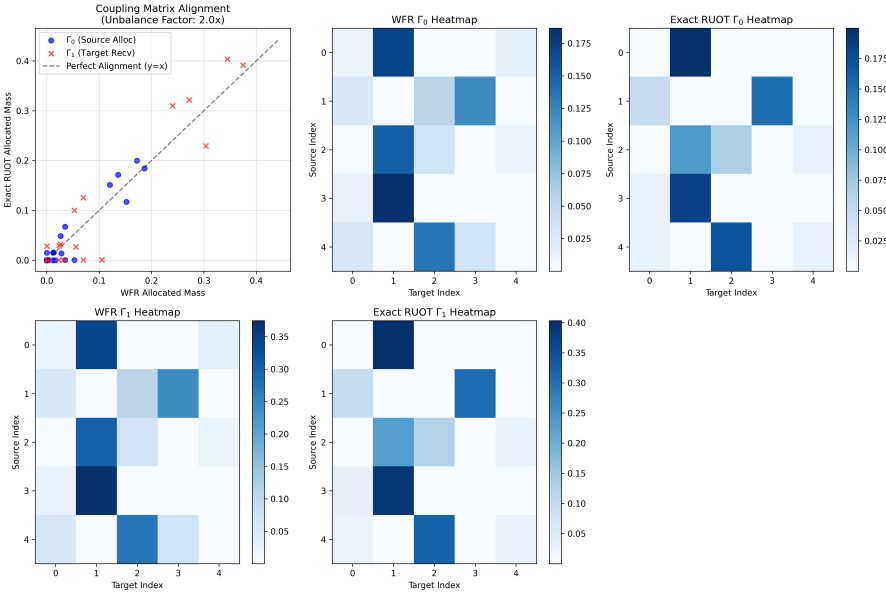

*Figure 14.* WFR approximation on 2D case ($\alpha = 2$).

# D. Relation to other algorithms

## D.1. Relation to SF²M

SF²M ([Tong et al., 2024b](#)) is the first simulation-free framework for learning balanced Schrödinger Bridge between arbitrary source and target distributions. It utilized the KL-disintegration to decoupled the SB to two parts: the regularized OT (ROT) coupling induced by the static SB problem, and the conditional path bridging two Diracs. Following ([Föllmer, 1988](#); [Léonard, 2014](#)), the static SB problem (7) can be written as a regularized OT (ROT) problem

$$\mathrm{KL}(\mathbb{P}_{01}\|\mathbb{Q}_{01}) = \frac{1}{2\nu^2}\int \|\boldsymbol{x}-\boldsymbol{y}\|^2 \mathbb{P}_{01}(\mathrm{d}\boldsymbol{x},\mathrm{d}\boldsymbol{y}) + \mathrm{KL}(\mathbb{P}_{01}\|\mu_0\otimes\mu_1) + C_\nu \tag{56}$$

Thus, SF²M first calculated the optimal coupling $\pi^\star_{2\nu^2}$ of the ROT problem

$$\min_\pi \frac{1}{2\nu^2}\int \|\boldsymbol{x}-\boldsymbol{y}\|^2 \pi(\mathrm{d}\boldsymbol{x},\mathrm{d}\boldsymbol{y}) + \mathrm{KL}(\pi\|\mu_0\otimes\mu_1) \tag{57}$$

then obtained the SB $\mathbb{P}^\star$ by integrating the conditional paths, which are Brownian bridges in this case

$$\mathbb{P}^\star = \int (\nu\mathbb{W})^{\boldsymbol{x}\boldsymbol{y}}\mathrm{d}\pi^\star_{2\nu^2}(\boldsymbol{x},\boldsymbol{y}) \tag{58}$$

SF²M tried to find the most likely stochastic process referencing on a Brownian motion, while USB tries to find the most likely stochastic process referencing on a branching Brownian motion. In comparison, USB extends the ROT coupling to RUOT semi-coupling, and generalizes the Brownian bridge to Poisson-Brownian bridge to deal with unbalanced source and target. Though USB focuses on integrating the stochastic and unbalanced effect of single-cell dynamics, the balanced cases can be naturally included. Under balanced cases, by setting the reference branching rate $\lambda \to 0^+$, USB reduces to SF²M. To see that, one can calculate the no-growth limit of the growth penalty (46). Since the penalty is even, we assume $g \geq 0$.

$$
\begin{aligned}
\lim_{\lambda\to 0^+} \Psi_{\nu,\lambda}(g) &= \lim_{\lambda\to 0^+} \nu^2\lambda\left(1 - \sqrt{1+g^2/\lambda^2} + \frac{g}{\lambda}\ln\left(\frac{g}{\lambda}+\sqrt{1+g^2/\lambda^2}\right)\right) \\
&= \lim_{\lambda\to 0^+} \nu^2\lambda\left(1 - \frac{g}{\lambda} + o(\lambda^{-1}) + \frac{g}{\lambda}\ln\frac{2g}{\lambda} + o(\lambda^{-1})\right) \\
&= \lim_{\lambda\to 0^+} \left(-\nu^2 g + o(1) + \nu^2 g\ln(2g) - \nu^2 g\ln(\lambda)\right) \\
&= \begin{cases} 0, & g = 0 \\ +\infty, & g \neq 0 \end{cases}
\end{aligned}
\tag{59}
$$

Hence, when $\lambda \to 0$, the growth rate is forced to be 0, and the corresponding RUOT problem (10) reduces to

$$
\begin{aligned}
\mathrm{RUOT}_{\text{no grwoth}}(\mu_0,\mu_1) &= \inf_{\rho,g,\boldsymbol{u}} \int_0^1 \int_{\mathcal{X}} \frac{1}{2}\|\boldsymbol{u}(\boldsymbol{x},t)\|_2^2 \rho_t(\boldsymbol{x})\mathrm{d}\boldsymbol{x}\mathrm{d}t \\
\text{s.t.} \quad &\partial_t\rho + \nabla_{\boldsymbol{x}}\cdot(\rho\boldsymbol{u}) = \frac{\nu^2}{2}\Delta_{\boldsymbol{x}}\rho, \ \rho_0 = \mu_0, \ \rho_1 = \mu_1
\end{aligned}
\tag{60}
$$

which is equivalent to the balanced SB problem ([Chen et al., 2016](#); [Pra, 1991](#)). Hence the semi-coupling reduces to the coupling induced by the static SB problem. For conditional path, Poisson-Brownian bridge also reduces to the standard Brownian bridge when there is no growth. Thus, USB can reduce to SF²M in mass conserved cases by referencing on a BBM with zero branching rate, which is a Brownian motion.

## D.2. Relation to WFR-FM

WFR-FM ([Peng et al., 2026](#)) is an unbalanced extension of flow matching based on WFR geometry. It can learn both the velocity and the growth rate simultaneously without ODE simulation, and one can prove that the learned measure trajectory is a geodesic under WFR geometry. The algorithm efficiently addresses unbalanced mass in single-cell dynamics in both theory and practice, but it cannot model stochasticity. In contrast, USB extends WFR-FM's unbalanced flow matching loss to a unbalanced score matching loss, thereby enabling efficient joint modeling of both unbalance and stochasticity in cell

dynamics. Notably, USB cannot be simply regarded as a stochastic extension of WFR-FM; their microscopic dynamics are dissimilar. WFR-FM is based on WFR geometry, whereas USB is based on the BSB problem. The mass variation along WFR-FM's conditional path is quadratic, causing the mass to decrease and then increase even when connecting two equal-mass Dirac measures via a WFR geodesic. By contrast, the mass variation along USB's conditional path is exponential; when two equal-mass Dirac measures are connected by a Poisson–Brownian bridge, the mass does not change. Hence, USB cannot include WFR-FM as a limit case. One interesting question is that is there a stochastic version of WFR which could naturally reduce to WFR?

### D.3. Relation to DeepRUOT

DeepRUOT (Zhang et al., 2025) employs dynamic RUOT to simultaneously model the stochasticity and unbalanced mass inherent in single-cell dynamics, solving the corresponding RUOT problem using a NeuralODE approach. In contrast, USB is proposed based on the BSB problem. It is also capable of simultaneously modeling stochasticity and unbalanced mass, and since it employs unbalanced score matching for solving, it is more efficient than DeepRUOT. To approximate the static semi-coupling within the BSB problem, USB also utilizes RUOT. However, it does not utilize DeepRUOT for the dynamic solution, instead, it approximates its static form using the quickly solvable static WFR.

### D.4. Relation to UDSB

UDSB (Pariset et al., 2023) is also an algorithm for solving stochastic transport between unbalanced measures. It is based on operator theory. UDSB introduces a coffin state outside the ambient space to model changes in total mass, and uses iterative proportional fitting (IPF) to solve the problem. In contrast, USB is designed based on the BSB problem, hence the two in fact tackle different problems. UDSB turns unbalanced stochastic transport into a diffusion Schrödinger bridge problem in an extended state space by introducing the coffin state; USB naturally introduces unbalanced effect by replacing the reference process of Schrödinger bridge from Brownian motion to branching Brownian motion. At the microscopic level, branching Brownian motion better matches the dynamical picture of cell division where new cells born from existing cells instead of a common coffin state, making USB more suitable for single-cell dynamics inference. Moreover, the solution techniques employed by the two methods differ substantially: UDSB is IPF-based and can be viewed as a unbalanced extension of classical Schrödinger bridge numerics, whereas USB is based on unbalanced score matching and should be regarded as a generalization of score matching methods.

### D.5. Relation to BranchSBM

BranchSBM (Tang et al., 2025) is a recently proposed multimodal generalization of the SB problem, aimed at capturing branched or divergent evolution from a common origin to multiple distinct outcomes. It introduces mass weights into the generalized SB (Liu et al., 2024), thereby extending it to an unbalanced generalized SB; by training multiple such unbalanced generalized SB with conserved total mass across branches, BranchSBM models how a population splits from a single source into several branches and ultimately reaches multiple targets. The branching emphasized by BranchSBM is at population-level, focusing on the allocation and flow of mass among branches. Since the total mass is conserved, BranchSBM is inherently unable to model the unbalanced effect in single-cell dynamics. In contrast, USB emphasizes particle-level branching, reflecting the dynamical picture of cell division and apoptosis to model unbalanced effect. How to combine BranchSBM's macroscopic branching structure with USB's microscopic branching structure to build a framework that simultaneously models unbalance, stochasticity, and multimodality is an interesting direction for future research.

### D.6. Relation to VGFM

VGFM (Wang et al., 2025) creatively employs semi-constrained OT and two-period transport to decouple the growth and velocity dynamics, subsequently achieving simultaneous modeling by integrating them through defined joint dynamics. In VGFM, the conditional path between Dirac pairs follows linear interpolation, while the mass evolution follows an exponential trajectory. This conditional path can be viewed as the limit of the USB conditional path as stochastic noise approaches zero; specifically, the reference process corresponds to a branching Brownian motion with a diffusion parameter of zero, exhibiting only birth-death without diffusion.

However, VGFM cannot be regarded as a deterministic degeneration of USB due to differences in their adopted couplings. When calculating the coupling, VGFM utilizes a semi-constrained OT plan, which essentially performs OT between two balanced distributions, thereby treating transitions and mass variations separately at this stage. In contrast, USB is a

principled algorithm based on the BSB problem. It employs RUOT semi-coupling to approximate the BSB problem, integrating transitions with birth-death dynamics within this very step. Consequently, even if the diffusion parameter of the reference process in USB is set to zero, it does not degenerate into VGFM.

Furthermore, VGFM involves a two-stage training process: the first stage uses flow matching for warm-up, while the second requires optimizing an OT loss via NeuralODE, which is simulation-based, to achieve better performance. By comparison, USB is a fully simulation-free algorithm that requires only a single stage of unbalanced score matching training.

### D.7. Relation to biologically motivated trajectory inference methods

Several non-OT-based single-cell trajectory inference methods have been established in the past few years. For example, PAGA (Wolf et al., 2019), Slingshot (Street et al., 2018), Palantir (Setty et al., 2019), RNA velocity-based methods (La Manno et al., 2018; Bergen et al., 2020; Qiu et al., 2022). They tackled the challenge of inferring single-cell dynamics from single scRNA-seq snapshot without any temporal information. In contrast, OT-based trajectory inference methods deal with another problem, i.e. inferring single-cell dynamics from few temporal snapshots where true time information are available. We point out that ignoring the temporal information may lead to unrealistic trajectories.

For example, we run PAGA and Slingshot on Simulation data (Figure 15). Two main issues are observed. 1) Relying on spatial proximity, PAGA fail when temporally adjacent cells are discontinuous on the manifold. For instance, PAGA drops connections between temporally valid but spatially distant clusters. 2) In the Simulation dataset, the Time 0 snapshot originates from two distinct spatial locations (bottom-left and bottom-right), which are naturally identified as separate clusters. without using temporal information, Slingshot forces a sequential, chronological relationship between these two concurrent starting populations, resulting in a temporal mismatch.

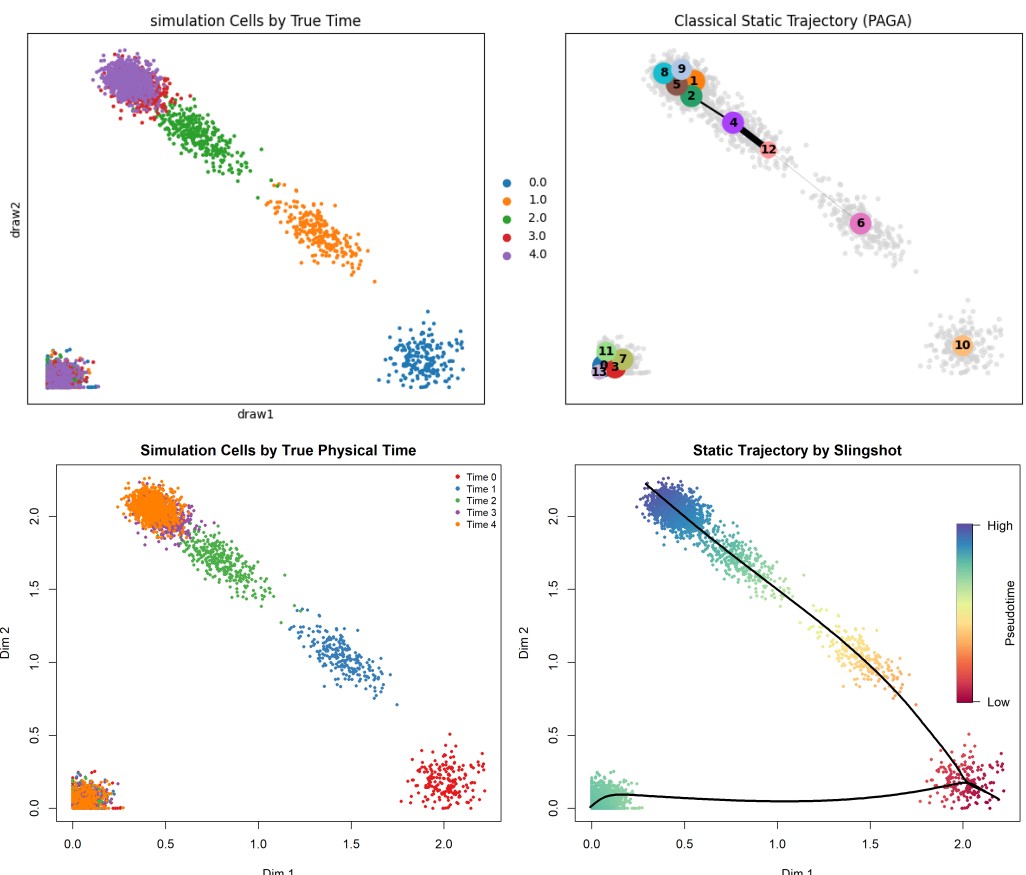

*Figure 15.* PAGA and Slingshot on Simulation data. Upper panel: PAGA; Lower panel: Slingshot

These issues fundamentally stem from the inability to leverage the temporal information. In contrast, OT-based methods like USB explicitly enforce temporal progression between snapshots, naturally avoiding these problems.

## E. Discussion on general travelling dirac

In this section, we first review Chizat's method (Chizat et al., 2018a) for recasting WFR into the OET formulation—namely, by computing the WFR cost between two Dirac measures (the corresponding trajectory is called the travelling Dirac) and expressing the WFR cost between two distributions as an integral of the Dirac WFR cost, thereby converting dynamic WFR into static WFR and then obtaining OET via duality. We then discuss the difficulties this approach encounters in the general RUOT problem.

### E.1. Travelling Dirac

To solve a WFR problem i.e. a RUOT problem with quadratic growth penalty, one can reformulate the WFR problem as a integral of WFR costs between pair of Diracs.

$$\mathrm{WFR}_\delta^2(\mu_0, \mu_1) = \inf_{(\gamma_0, \gamma_1) \in (\mathcal{M}_+(\mathcal{X}^2))^2} \int_{\mathcal{X}^2} \mathrm{WFR\text{-}DD}_\delta^2(\gamma_0(\boldsymbol{x}, \boldsymbol{y})\delta_{\boldsymbol{x}}, \gamma_1(\boldsymbol{x}, \boldsymbol{y})\delta_{\boldsymbol{y}}) \mathrm{d}\boldsymbol{x}\mathrm{d}\boldsymbol{y}, \tag{61}$$

where the WFR cost between two Diracs is

$$\mathrm{WFR\text{-}DD}_\delta^2(m_0\delta_{\boldsymbol{x}_0}, m_1\delta_{\boldsymbol{x}_1}) = \inf_{m, \boldsymbol{x}} \int_0^1 \frac{1}{2}(\|\dot{\boldsymbol{x}}(t)\|_2^2 + \delta^2 |\frac{\dot{m}(t)}{m(t)}|^2)m(t)\mathrm{d}t \tag{62}$$

$$\text{s.t.} \qquad m(0) = m_0, m(1) = m_1, \boldsymbol{x}(0) = \boldsymbol{x}_0, \boldsymbol{x}(1) = \boldsymbol{x}_1$$

Following (Chizat et al., 2018a), one can derive the Euler-Lagrange equation of (62).

$$\begin{cases} \dfrac{\mathrm{d}}{\mathrm{d}t}(m\dot{\boldsymbol{x}}) = \boldsymbol{0} & (\dfrac{\partial}{\partial \boldsymbol{x}} = \dfrac{\mathrm{d}}{\mathrm{d}t}\dfrac{\partial}{\partial \dot{\boldsymbol{x}}}) \\ \dfrac{1}{2}|\dot{\boldsymbol{x}}|^2 - \dfrac{\delta^2\dot{m}^2}{2m^2} = \delta^2\dfrac{\ddot{m}m - \dot{m}^2}{m^2} & (\dfrac{\partial}{\partial m} = \dfrac{\mathrm{d}}{\mathrm{d}t}\dfrac{\partial}{\partial \dot{m}}) \end{cases} \tag{63}$$

The first equation stands for momentum conservation, which yields

$$m\dot{\boldsymbol{x}} = \boldsymbol{\omega} \tag{64}$$

for some constant vector $\boldsymbol{\omega}$. By plugging in $\dot{\boldsymbol{x}} = \frac{\boldsymbol{\omega}}{m}$, the second equation becomes

$$2\ddot{m}m - \dot{m}^2 = |\boldsymbol{\omega}|^2/\delta^2 \tag{65}$$

which describes the evolution of mass. Solutions of the second order ODE are of the form $m(t) = At^2 + Bt + C$. Plugging in this ansatz, one can derive a set of equations of $A, B, C$.

$$\begin{cases} C = m_0 \\ A + B + C = m_1 \\ 4AC - B^2 = |\boldsymbol{\omega}|^2/\delta^2 \end{cases} \tag{66}$$

The existence of the solution has also been proved by (Chizat et al., 2018a). This optimal transport trajectory between two Diracs is called travelling Dirac. Given the closed form of travelling Dirac, one can directly calculate (62) as

$$\mathrm{WFR\text{-}DD}_\delta^2(m_0\delta_{\boldsymbol{x}_0}, m_1\delta_{\boldsymbol{x}_1}) = 2\delta^2(m_0 + m_1 - 2\sqrt{m_0 m_1}\,\overline{\cos}(\frac{\|\boldsymbol{x}_0 - \boldsymbol{x}_1\|_2}{2\delta})) \tag{67}$$

Thank to this explicit form of Dirac-Dirac WFR cost, the WFR cost between two measures can be written in a static form

$$\mathrm{WFR}_\delta^2(\mu_0, \mu_1) = 2\delta^2 \inf_{(\gamma_0, \gamma_1) \in (\mathcal{M}_+(\mathcal{X}^2))^2} \int_{\mathcal{X}^2} \left(\gamma_0(\boldsymbol{x}, \boldsymbol{y}) + \gamma_1(\boldsymbol{x}, \boldsymbol{y}) - 2\sqrt{\gamma_0(\boldsymbol{x}, \boldsymbol{y})\gamma_1(\boldsymbol{x}, \boldsymbol{y})}\,\overline{\cos}(\frac{\|\boldsymbol{x} - \boldsymbol{y}\|_2}{2\delta})\right)\mathrm{d}\boldsymbol{x}\mathrm{d}\boldsymbol{y} \tag{68}$$

which is (4). Hence, the WFR problem (3) is decoupled into two parts: the static WFR semi-coupling and travelling Dirac. The former describes the mass transportation plan i.e. how much mass to be sent from source $\boldsymbol{x}$ and how much to be received at target $\boldsymbol{y}$, while the latter describes what happens during the transportation process i.e. the variation of the position and mass of the specific Dirac given the source and target.

The static WFR is further proved to be equivalent to the OET problem (50) which is easy to solve.

### E.2. The travelling Dirac for general RUOT

For general RUOT problem (10), in order to obtain the semi-coupling, one need to find a static form for it. We tried to follow (Chizat et al., 2018a) to find the travelling Dirac of the general RUOT problem. We first write down the RUOT between two Diracs.

$$\text{RUOT-DD}(m_0\delta_{\boldsymbol{x}_0}, m_1\delta_{\boldsymbol{x}_1}) = \inf_{m,\boldsymbol{x}} \int_0^1 \frac{1}{2}\Big( \|\dot{\boldsymbol{x}}(t)\|_2^2 + \Psi(\frac{\dot{m}(t)}{m(t)}) \Big) m(t)\mathrm{d}t \tag{69}$$

$$\text{s.t.} \qquad m(0) = m_0, m(1) = m_1, \boldsymbol{x}(0) = \boldsymbol{x}_0, \boldsymbol{x}(1) = \boldsymbol{x}_1$$

Then, we derive the Euler-Lagrange equation of it.

$$\begin{cases} \dfrac{\mathrm{d}}{\mathrm{d}t}(m\dot{\boldsymbol{x}}) = \boldsymbol{0} & (\dfrac{\partial}{\partial \boldsymbol{x}} = \dfrac{\mathrm{d}}{\mathrm{d}t}\dfrac{\partial}{\partial \dot{\boldsymbol{x}}}) \\ |\dot{\boldsymbol{x}}|^2 + \Psi - \dfrac{\dot{m}}{m}\Psi' = \dfrac{\ddot{m}m - \dot{m}^2}{m^2}\Psi'' & (\dfrac{\partial}{\partial m} = \dfrac{\mathrm{d}}{\mathrm{d}t}\dfrac{\partial}{\partial \dot{m}}) \end{cases} \tag{70}$$

The first equation stands for the conservation of momentum, while the second equation again describes the evolution of mass. Plugging in $\dot{\boldsymbol{x}} = \frac{\boldsymbol{\omega}}{m}$, the second equation is of the form

$$|\boldsymbol{\omega}|^2 + m^2\Psi - m\dot{m}\Psi' = (\ddot{m}m - \dot{m}^2)\Psi'' \tag{71}$$

The complex second order ODE has no analytic solution for most of the choice of $\Psi$. If we want to derive an analytic solution, $\Psi$ should be well chosen to make the above equation simple enough. To get rid of more complex terms, one may want $\Psi''$ to be a constant instead of a function of $\frac{\dot{m}}{m}$. Hence, for second order differentiable functions, a proper choice for $\Psi$ is a polynomial with degree $\leq 2$. Taking $\Psi(z) = az^2 + bz + c$, we have

$$|\boldsymbol{\omega}|^2 + cm^2 = 2a\ddot{m}m - a\dot{m}^2 \tag{72}$$

When $c = 0$, it is WFR; when $ac < 0$, the mass evolves in cosine law; when $ac > 0$, the mass evolves in hyperbolic cosine law. Note that (72) is independent of the first order term coefficient $b$ since the total cost induced by it is actually a constant $\frac{b}{2}\int_0^1 \dot{m}(t)\mathrm{d}t = \frac{b}{2}(m_1 - m_0)$.

For the RUOT problem with growth penalty like (46) or other general forms, the essential difficulty of deriving a static form for it is that the equation (72) is so hard to solve analytically. Though one can still write the RUOT cost between two measures as the integral of RUOT costs between Diracs

$$\text{RUOT}(\mu_0, \mu_1) = \inf_{(\gamma_0, \gamma_1) \in (\mathcal{M}_+(\mathcal{X}^2))^2} \int_{\mathcal{X}^2} \text{RUOT-DD}(\gamma_0(\boldsymbol{x}, \boldsymbol{y})\delta_{\boldsymbol{x}}, \gamma_1(\boldsymbol{x}, \boldsymbol{y})\delta_{\boldsymbol{y}})\mathrm{d}\boldsymbol{x}\mathrm{d}\boldsymbol{y}, \tag{73}$$

the RUOT cost between two Diracs (RUOT-DD) is intractable, hence it is challenging to solve the semi-coupling through this approach.

# F. Inference workflow

---

**Algorithm 3** Continuous inference workflow

---

**Require:** Data at initial time point $\mathcal{D}_0$, diffusion parameter $\nu$, timestep $\Delta t$, learned vector net $\boldsymbol{v_\theta}(\boldsymbol{x}, t)$, growth rate net $g_{\boldsymbol{\theta}}(\boldsymbol{x}, t)$, score net $\boldsymbol{s_\theta}(\boldsymbol{x}, t)$.

1: **for** $\boldsymbol{x}$ in $\mathcal{D}_0$ **do**
2:     $\omega = 1$
3:     **while** simulation **do**
4:         $\boldsymbol{\xi} \sim \mathcal{N}(\boldsymbol{0}, \Delta t \mathrm{I})$
5:         $\boldsymbol{x} \leftarrow \boldsymbol{x} + \left(\boldsymbol{v_\theta}(\boldsymbol{x}, t) + \frac{\nu^2}{2}\boldsymbol{s_\theta}(\boldsymbol{x}, t)\right)\Delta t + \nu\boldsymbol{\xi}$
6:         $\omega \leftarrow \omega \cdot e^{g_{\boldsymbol{\theta}}(\boldsymbol{x}, t)\Delta t}$
7:     **end while**
8:     $\mathcal{D}_{weight}$ append $(\boldsymbol{x}, \omega)$
9: **end for**
10: **return** $\mathcal{D}_{weight}$

---

**Algorithm 4** Branching inference workflow

---

**Require:** Data at initial time point $\mathcal{D}_0$, diffusion parameter $\nu$, timestep $\Delta t$, learned vector net $\boldsymbol{v_\theta}(\boldsymbol{x}, t)$, growth rate net $g_{\boldsymbol{\theta}}(\boldsymbol{x}, t)$, score net $\boldsymbol{s_\theta}(\boldsymbol{x}, t)$.

1: $\mathcal{D}_{next} \leftarrow \mathcal{D}_0$
2: **while** simulation **do**
3:     $\mathcal{D} \leftarrow \mathcal{D}_{next}$
4:     **for** $\boldsymbol{x}$ in $\mathcal{D}$ **do**
5:         $\boldsymbol{\xi} \sim \mathcal{N}(\boldsymbol{0}, \Delta t \mathrm{I})$
6:         $\boldsymbol{x} \leftarrow \boldsymbol{x} + \left(\boldsymbol{v_\theta}(\boldsymbol{x}, t) + \frac{\nu^2}{2}\boldsymbol{s_\theta}(\boldsymbol{x}, t)\right)\Delta t + \nu\boldsymbol{\xi}$
7:         $\alpha \sim \mathcal{U}[0, 1]$
8:         **if** $\alpha \leq |g_{\boldsymbol{\theta}}(\boldsymbol{x}, t)\Delta t|$ **then**
9:             **if** $g_{\boldsymbol{\theta}}(\boldsymbol{x}, t) \geq 0$ **then**
10:                 $\mathcal{D}_{next}$ append $\boldsymbol{x}$
11:                 $\mathcal{D}_{next}$ append $\boldsymbol{x}$     ($\boldsymbol{x}$ divides into two)
12:             **else**
13:                 pass     ($\boldsymbol{x}$ dies)
14:             **end if**
15:         **else**
16:             $\mathcal{D}_{next}$ append $\boldsymbol{x}$     (No branching)
17:         **end if**
18:     **end for**
19: **end while**
20: **for** $\boldsymbol{x}$ in $\mathcal{D}_{next}$ **do**
21:     $\mathcal{D}_{weight}$ append $(\boldsymbol{x}, 1)$
22: **end for**
23: **return** $\mathcal{D}_{weight}$

---

# G. Advice for practitioners

We provide the following practical advice for future users of USB.

**Hardware Setup.** Due to our simulation-free continuous measure flow design, practitioners do not need expensive multi-GPU clusters. A standard workstation with a moderate CPU (for saving the global coupling matrix) and a basic entry-level GPU is completely sufficient to train USB on tens of thousands of cells (since VRAM cost is less than 1 GB on 49302 cells data; you can also set larger batch size to use more VRAM). For smaller dataset, you can run USB on your laptop. For

detailed computational cost, see C.10.

**Data Suitability.** Practitioners can safely apply USB's WFR approximation to standard single-cell RNA-seq datasets. Since biological mass changes (proliferation/apoptosis) are typically gradual between observed time points, the moderate unbalance assumption always holds (see C.11).

**Growth penalization.** For users with biological background, if you want to explicitly inject biological prior to growth, please tune $\delta$ instead of $\nu, \lambda$. A large $\delta$ means that a small growth rate is preferred.

**Branching inference.** When simulating trajectories, we use one-hot offspring distribution (restricting to binary split or death) as a default (B.4). However, practitioners with specific domain knowledge can flexibly adjust the distribution to simulate different underlying discrete biological events (e.g. when there is a constant apoptosis rate) without retraining the model.

## H. The Use of Large Language Models (LLMs)

LLMs were used only for grammatical correction to enhance the overall readability of this paper.

