# OpenReview forum: "Beyond Continuity: Simulation-free Reconstruction of Discrete Branching Dynamics from Single-cell Snapshots"
_ICML.cc/2026/Conference — ICML 2026 regular_

### Official Review · Reviewer_UBh2 · 2026-02-13

**Soundness:** 3
**Presentation:** 2
**Significance:** 3
**Originality:** 3
**Overall Recommendation:** 5
**Confidence:** 5

**Summary:**

This paper introduces a Schrödinger bridge problem with unbalanced measures, motivated by reconstruction of cellular branching dynamics. As far as I can tell from my expertise in this field, this problem is central and important, hence the paper tackles an important not fully answered problem.

**Compliance With Llm Reviewing Policy:**

Affirmed.

**Key Questions For Authors:**

**Question about a claim on irregular dynamics.**

In their introduction, the authors write "Furthermore, we highlight a critical limitation in current methods modeling unbalanced mass: they uniformly treat mass as a continuously varying quantity. While natural within frameworks like flow matching or NeuralODE, this assumption overlooks a fundamental characteristic of cel- lular proliferation and apoptosis: cell numbers are discrete values that change through jumps" (p.2, c.1, l. 56-62). I do not see this taken into account in the paper's methods. Could you please elaborate on this point? Maybe this is related to the undefined term $m_t$?

** Compute budget and time.**

In your experimental part, could you please report compute budget and times for all different methods, in order to better understand the different trade-offs? Ideally, it would be nice to have an analog of Figure 7 for all different methods.

** Suggestion.**

I suggest moving table 2 up in the text, as it nicely clarifies the differences between the author's method and concurrent approaches.

I also feel that moving some elements of the pseudo-codes given in the Appendix would help increase the readability of the main text.

Could you also please take some time to clarify for unfamiliar readers to which regard your method is simulation free?

**Limitations:**

The paper relies on the WFR approximation, but it seems to me that the good experimental results do not make this a true limitation.

**Strengths And Weaknesses:**

**Strengths.**

The paper bridges unbalanced, dynamical and simulation free approaches to obtain a method that is able to meet all desired criteria. This method performs well in simulated and real-world settings, including several datasets reported in the Appendix. I hence believe this paper to be of high quality.

**Weaknesses.**

The paper can be a bit hard to parse and to understand for the unfamiliar reader.  I have added a few questions for the authors bellow.

**Typos and clarifications.**

- In equation (1), the term $m_t$ is not defined.
- l. 132, c.2: We point out that the algorithm **is consist** of two important part
- l.204, c.1: **a** Brownian bridge
- l.357, c.2: "We evaluate how accurate can USB mathces the measure at observed time points" should read "We evaluate how accurately USB can match the measures at observed time points"

---

> ### Author Rebuttal · Authors · 2026-03-29
>
> We sincerely thank the reviewer for carefully reviewing our manuscript and providing constructive comments. In the revised manuscript, we will double-check and correct the spelling and grammar. Also, we will add the definition of $m_t$. It is the mass of the simulated particle. Generally, the evolution of a probability distribution can be modeled by the motion of a population of particles within a velocity field. By assigning a time-evolving mass to each particle, we can model the evolution of a positive measure. That's why there is $m_t$. Below is our detailed response to key questions.
>
> **Q1.** USB incorporates discrete changes in cell numbers into its theoretical framework by assuming the reference stochastic process is a Branching Brownian Motion (BBM) characterized by jump-like mass dynamics. During the inference phase, we introduce a branching inference procedure, which explicitly simulates discrete cell division and apoptosis events by realizing the learned growth field $g$ as a branching rate. However, USB operates continuously rather than discretely during the training phase. Because the BSB problem is ill-posed and highly intractable, USB approximates it via continuous measure flow framework for efficiency.
>
> **Q2.** We ran DeepRUOT, SF2M, WFR-FM, and USB on a single H200 GPU across varying cell numbers (10000,20000,30000,40000,49302) using the Mouse Hematopoiesis dataset. Hyperparameters: batch size=512, epochs=3000, mini-batch OT with chunk size=2000, networks are MLPs with 256 hidden channels and 5 layers. The runtime and memory results are summarized below:
>
> |Metric|Method|N=10000|N=20000|N=30000|N=40000|N=49302|
> | :--- | :--- | :--- | :--- | :--- | :--- | :--- |
> |**Runtime (s)**|DeepRUOT|400.03|753.90|1100.51|1596.62|2007.85|
> | |WFR-FM|27.95|33.36|42.17|54.18|67.80|
> | |SF2M|28.35|33.84|42.83|54.74|68.78|
> | |**USB**|29.40|35.39|44.26|56.21|70.10|
> |**Peak CPU RAM (MB)**|DeepRUOT|137.07|457.16|1010.16|1769.52|2668.26|
> | |WFR-FM|1121.32|4336.26|9681.68|17140.84|26032.36|
> | |SF2M|1121.32|4336.26|9681.68|17140.84|26032.36|
> | |**USB**|1121.32|4336.25|9681.68|17140.84|26032.36|
> |**Peak GPU VRAM (MB)**|DeepRUOT|2742.58|5904.51|9515.35|13614.29|17586.63|
> | |WFR-FM|242.21|708.72|444.44|717.98|522.33|
> | |SF2M|242.21|708.82|444.54|718.08|522.43|
> | |**USB**|242.21|713.05|448.77|722.31|526.65|
>
> Simulation-free algorithms outperform DeepRUOT in runtime. The extra time and memory cost of USB against SF2M and WFR is negligible. Since the memory of SF2M, WFR, USB are very similar, we present an analysis below.
>
> **Shared OT Pre-computation**
>
> As shown in the table, all three methods exhibit an identical peak CPU memory grows quadratically with respect to $N$. It is because of that they share an OT caculation steep which causes the dominating $O(N^2)$ CPU memory peak.
>
> **Small Extra Memory on GPU**
>
> USB's peak VRAM is consistently ~4.3 MB higher than SF2M and WFR across all sample size. This extra 4.3 MB difference corresponds to the parameter size of our additional neural network. SF2M use a velocity net and a score net, WFR use a velocity net and a growth net, while USB uses the all 3 networks. The extra network has 5 hidden layers, each has 256 neurons. The total parameter number is about $256\times(51+256\times4+51) \approx 290000$. We use float32 and Adam ($memory \approx4\times parameters$), thus the expected extra memory is about $290000\times4Bytes\times4/1024^2\approx4.4MB$, which is consistent to the reported extra memory.
>
> **Suggestion.** We sincerely thank the reviewer for these highly constructive suggestions, which will greatly enhance the readability and accessibility of our paper. We will move the table 2 up and move the training pseudocode into section 5.
>
> **Clarification of the term "Simulation-Free"**
> We will add the following part into appendix to introduce ‘Simulation-Free’ to unfamiliar readers.
>
> The term 'simulation-free' indicates that during the training phase, there is no need to generate trajectories to compute the loss. Suppose we aim to learn a velocity field parameterized by a neural network $f$. An example of a non-simulation-free approach is NeuralODE. When training a NeuralODE model, the loss is defined over a simulated trajectory; consequently, computing the loss at each iteration requires first simulating an entire trajectory using the current velocity field $f$. This numerical simulation is highly time-consuming, as it requires multiple sequential evaluations of $f$. In contrast, an example of a simulation-free approach is Flow Matching. When training a flow model, the loss is constructed directly with respect to the field $f$ itself, meaning each loss computation only requires a single evaluation of $f$. This training paradigm, which circumvents the need for numerical trajectory simulation, is termed 'simulation-free.' By avoiding the repeated evaluations of $f$ required to generate trajectories, it achieves significantly higher computational efficiency.

---

> > ### Author Rebuttal · Reviewer_UBh2 · 2026-04-01
> >
> > Thank you for your answer. I will keep my score at 5, since I feel that is the score that fits your paper, but increase my confidence to 5. I will strongly avocate for acceptance of this paper during the reviewer-AC discussion phase.
> >
> > As a small remark, I believe that the term of "jumps" to describe the fact that the mass $m_t$ is a SDE can induce a confusion with "jump SDEs" (see for example https://en.wikipedia.org/wiki/Jump_diffusion) which combines an SDE and a marked point process. I would avoid it.

---

> > > ### Author Response · Authors · 2026-04-04
> > >
> > > We sincerely thank the reviewer again for their thorough review of our manuscript and for providing such constructive suggestions and positive comments. Sorry for the confusion 'jump'. We use 'jump' since the state space of birth-death dynamics is discrete, just like jump process. We will try to avoid it by replacing 'jump' by 'discrete' or other words in the revised version. Thank you for pointing out this. We are grateful to the reviewer for helping us improve this work.

---

### Official Review · Reviewer_Mkbo · 2026-03-13

**Soundness:** 4
**Presentation:** 3
**Significance:** 3
**Originality:** 4
**Overall Recommendation:** 5
**Confidence:** 5

**Summary:**

The authors develop USB (Unbalanced Schrodinger Bridge), for simulation-free single-cell trajectory inference to model birth-death dynamics and branching dynamics motivated by biological processes. They do this through a general score matching framework, and show recovery of true ground truth dynamics in a wide range of single-cell datasets. As compared to past methods built around neural ODEs, but instead builds from Brownian motion theory and avoids computational expensive simulation-based designs.

**Compliance With Llm Reviewing Policy:**

Affirmed.

**Final Justification:**

I have increased my confidence from my initial review and recommend acceptance. The work is novel and significant, and the authors were very receptive and responsive in the rebuttal.

**Key Questions For Authors:**

- Can you compare to a more biological trajectory inference method, such as Slingshot (Street et. al 2018)? Even if quantitative comparisons are not feasible, qualitative, visual, comparisons on reduced dimension plots would be interesting.
- How sensitive is overall performance to the performance of WFR?

**Limitations:**

Yes

**Strengths And Weaknesses:**

- The authors have several in-depth comparisons on real datasets with appropriate comparison methods.
- I do believe the paper has strong novelty and very sound experimental and simulation analyses. The work is original, clearly delineated from past work with clear empirical and theoretical improvements.
- The work is very well-positioned within the existing literature. The literature review and related work is very well described and well-cited, though I would like to see a few more citations specifically from biologically motivated trajectory inference methods rather than entirely OT related methods.

- The approximation gap for use of WFR needs to studied further.
- The last paragraph of section 2 is too much of a citation dump and makes it unreadable.
- Typos such as "sthocastic", please do a full spell check.

---

> ### Author Rebuttal · Authors · 2026-03-29
>
> We sincerely thank the reviewer for carefully reviewing our manuscript and providing constructive comments. In the revised manuscript, we will simplify the related works, expand our discussion of biological trajectory inference algorithms (such as PAGA, Slingshot), and carefully double-check the spelling and grammar. Below is our detailed response to key questions.
>
> **Q1.** We compared USB against classical trajectory inference (TI) algorithms (**Slingshot** and **PAGA**) on Simulation and Dyngen datasets (hyperparameter: $k=15, resolution=0.3$ and Leiden clustering for PAGA; $cluster=5$ for Slingshot). Visual results are available via the anonymous link: https://osf.io/kevun/overview?view_only=04c7046e071244228912b5116bdfa1e3 (Fig.S1-S2: PAGA; Fig.S3-S5: Slingshot).
>
> While designed for static-snapshot, when applied to time-series data, these methods exhibit 5 critical limitations that highlight the necessity of our USB framework:
>
> * **1. Disconnected Structures:** Relying on spatial proximity, TI methods fail when temporally adjacent cells are discontinuous on the manifold. For instance, PAGA drops connections between temporally valid but spatially distant clusters (Fig. S1, isolating node 10). USB bridges these via time-calibrated SDEs.
> * **2. Ignoring Temporal information:** As shown in Fig.S3, S4, in the Simulation dataset, the Time 0 snapshot originates from two distinct spatial locations (bottom-left and bottom-right), which are naturally identified as separate clusters. We ran Slingshot twice setting each as start cluster. Because Slingshot cannot use temporal information, it forces a sequential, chronological relationship between these two concurrent starting populations, resulting in a temporal mismatch. Furthermore, the cell population in the bottom-left remains spatially stationary across subsequent time points. Static clustering algorithms naively merge these temporally distinct cells into a single cluster, completely failing to identify this region as a dynamic 'attractor' (or sink) and thus missing the true dynamics. These critical errors fundamentally stem from the inability to leverage the temporal information. In contrast, OT-based methods like USB explicitly enforce temporal progression between snapshots, naturally avoiding these problems.
> * **3. Population Averages vs. Single-Cell Resolution:** Classical methods yield coarse macroscopic averages (cluster graphs or principal curves). In contrast, OT-based methods like USB infer true single-cell level trajectories.
> * **4. Lack of Generalization:** PAGA and Slingshot perform static geometric fitting without predictive power. USB learns continuous global vector and growth fields, enabling generalization on unseen cells.
> * **5. Inability to Model Unbalanced Effects:** Classical TI algorithms cannot handle cell population growth and apoptosis.
>
> We will include these comprehensive comparisons in the revised manuscript to further solidify the motivation for OT-based methods.
>
> **Q2.** We approximate the intractable RUOT semi-coupling using the efficient WFR semi-coupling. We compared them on small numerical examples ((RUOT were optimized using scipy) to show the validity of this approximation.
>
> **Eg1 (1D Splitting)** An initial unit mass at $x=0$ is transported to two targets at $x=-1$ and $x=1$ (with target masses of $0.4$ and $1.6$, respectively). As shown in Fig.S8, while the absolute costs differ, the normalized cost landscapes of WFR and RUOT are highly similar, achieving the exact same global minimum at $0.303$ (the optimal splitting proportion).
>
> **Eg2 (2D)** We transported mass from 5 source points (total mass $1.0$) to 5 target points. The total target mass is scaled by an unbalance factor $\alpha \in \{0.7, 1.0, 1.3, 1.5, 2.0\}$, with individual target masses drawn from a Dirichlet distribution. In Fig.S9–S13, scatter plots comparing the matrix elements of WFR and RUOT show that the vast majority of points align tightly along the 45-degree diagonal ($y=x$). The accompanying heatmaps further visually confirm that the resulting optimal transport plans are very similar.
>
> These results demonstrate that WFR is a highly accurate and efficient approximation of RUOT when the unbalance is moderate. Since real biological mass changes are typically not violently drastic between consecutive, reasonably sampled time points, assuming a moderate unbalance effect is biologically sound and justified.

---

> > ### Author Rebuttal · Reviewer_Mkbo · 2026-04-03
> >
> > I am very satisfied with the authors' responses. I maintain my original recommendation of accept, but have increased my confidence. Thank you for the thorough rebuttal with additional results.

---

> > > ### Author Response · Authors · 2026-04-04
> > >
> > > We sincerely thank the reviewer again for their thorough review of our manuscript and for providing such constructive suggestions and positive comments. We will incorporate the corresponding modifications in the revised version (e.g. discussion of biological TI methods and so on). We are grateful to the reviewer for helping us improve this work.

---

### Official Review · Reviewer_eJXQ · 2026-03-13

**Soundness:** 3
**Presentation:** 3
**Significance:** 4
**Originality:** 3
**Overall Recommendation:** 5
**Confidence:** 3

**Summary:**

The paper develops a stochastic simulation-free framework to model single-cell trajectory inference with unbalanced mass evolution. Prior unbalanced OT methods model mass continuously, whereas cell proliferation and apoptosis are discrete phenomena. Therefore, the authors introduce a framework connected to branching Schrödinger bridges and a simulation-free unbalanced score-matching objective.
The simulation-free approach is achieved by regressing on a conditional/marginal unbalanced Poisson-Brownian interpolating path. Empirically, the paper claims strong measure-reconstruction performance on synthetic and real single-cell datasets, recovery of ground-truth growth rates on synthetic data, and the ability to simulate discrete birth/death events at single-cell resolution.

**Compliance With Llm Reviewing Policy:**

Affirmed.

**Final Justification:**

Solid contribution to the field, the rebuttal answered all my questions thus I am increasing my score to a 5. I think this paper is above the acceptance threshold.

**Key Questions For Authors:**

*Which parts of USB should be understood as exact consequences of the BSB formulation, and which parts are pragmatic approximations due to intractable semi-couplings? A cleaner separation would help.

*How sensitive is performance to the WFR approximation of the RUOT semi-coupling? On a small low-dimensional problem, can you compare against a stronger or more exact stochastic-unbalanced baseline to justify this approximation?

*Can the authors provide a quantitative metric specifically for the discrete claim, such as event-count statistics, branch occupancy ratios, fate calibration, or lineage-consistent evaluation on a dataset where such supervision is available?

Can the authors provide a direct runtime and memory comparison against WFR-FM, SF2M, and DeepRUOT on the same hardware/tasks.

**Limitations:**

Yes.

**Strengths And Weaknesses:**

## Strengths

* The problem is important. Modeling stochasticity with unbalanced mass dynamics and branching-like behavior in single-cell dynamics via a simulation-free framework is an important gap in the literature, and the paper frames that gap clearly.

* The simulation-free training formulation is the strongest technical contribution. The conditional/marginal formulation in Section 4 is conceptually clean, and Theorem 4.3 gives a tractable training objective from otherwise intractable marginal quantities.

* The experimentation is broad. On synthetic tasks, the paper shows strong distribution matching and mass matching, and on real datasets, it remains competitive.

----
## Weaknesses
* The paper’s central theoretical framing overstates the extent to which USB is an exact tractable solver for BSB: both the static BSB semi-coupling and its RUOT relaxation are intractable, and the implemented method ultimately relies on a WFR approximation. This makes the practical algorithm feel more like a heuristic BSB-inspired construction than a clean solver for the claimed target problem.

* The discrete branching interpretation is partly imposed at inference time rather than learned end-to-end. In the appendix, branching inference converts the learned growth rate into a non-homogeneous jump process by assuming a one-hot offspring distribution with only death or binary split. Because of that, the paper’s strongest qualitative claim — realistic discrete single-cell birth/death simulation — is not fully validated quantitatively.

---

> ### Author Rebuttal · Authors · 2026-03-29
>
> We sincerely thank the reviewer for carefully reviewing our manuscript and providing constructive comments. Since the two main concerns raised in the Weaknesses section—specifically, that USB is not an exact BSB solver and that the discrete claim lacks full quantitative validation—are directly reflected in Key Questions, we will address them comprehensively in our detailed responses below.
>
> **Q1.** We apologize for any confusion. USB is not intended to be an exact solver for the BSB problem; instead, it uses the BSB framework to simultaneously model unbalanced dynamics, stochasticity, and discrete jumps in cell numbers, efficiently approximating the solution through a continuous measure flow framework. The KL-disintegration (Eq. 22) and all preceding derivations are exact. However, approximations are introduced when minimizing the two terms on the right-hand side of Eq. 22. For the first term (static BSB), we approximate it using the semi-coupling of its convex relaxation, RUOT. Because a simulation-free solution for general RUOT remains an open question, we further approximate it with the WFR semi-coupling for efficiency. or the second term (BSB conditional path), the branching nature of trajectories implies multiple potential endpoints, making it impossible to compute a conditional path with fixed endpoints as a strong solution. Thus, we use a Poisson-Brownian bridge to provide the conditional path in a weak-solution sense, meaning its evolution equation is consistent with the BSB. While these steps make USB an approximate solver, our main goal is not to solve the BSB problem itself, but to leverage it to capture complex single-cell dynamics. These approximations are necessary pragmatic trade-offs to ensure algorithmic efficiency.
>
> **Q2.** As previously discussed, we approximate the intractable RUOT semi-coupling using the efficient WFR semi-coupling. We computed their solutions on small numerical examples (RUOT were optimized using scipy). Visualizations are available via the anonymous link: https://osf.io/kevun/overview?view_only=04c7046e071244228912b5116bdfa1e3
>
> **Eg1 (1D Splitting)** An initial unit mass at $x=0$ is transported to two targets at $x=-1$ and $x=1$ (with target masses of $0.4$ and $1.6$, respectively). As shown in Fig.S8, while the absolute costs differ, the normalized cost landscapes of WFR and RUOT are highly similar, achieving the exact same global minimum at $0.303$ (the optimal splitting proportion).
>
> **Eg2 (2D)** We transported mass from 5 source points (total mass $1.0$) to 5 target points. The total target mass is scaled by an unbalance factor $\alpha \in \{0.7, 1.0, 1.3, 1.5, 2.0\}$, with individual target masses drawn from a Dirichlet distribution. In Fig.S9–S13, scatter plots comparing the matrix elements of WFR and RUOT show that the vast majority of points align tightly along the 45-degree diagonal ($y=x$). The accompanying heatmaps further visually confirm that the resulting optimal transport plans are very similar.
>
> These results demonstrate that WFR is a highly accurate and efficient approximation of RUOT when the unbalance is moderate. Since real biological mass changes are typically not violently drastic between consecutive, reasonably sampled time points, assuming a moderate unbalance effect is biologically sound and justified.
>
> **Q3.** To rigorously validate USB's discrete birth-death simulations, we conducted evaluations at both microscopic and macroscopic levels
>
> **1. Microscopic: Event-Count Statistics (Simulation Dataset)**
> The ground-truth division rate is $g_{true}(X_2) = 0.5 X_2^2/(1+X_2^2)$. We simulated 2,000 branching trajectories over 400 steps ($\Delta t=0.01$). Collecting all $X_2$ coordinates traversed along these paths, we partitioned them into 100 bins. For each bin, we computed the *Theoretical Expected Event Counts* ($\sum g_{true}(X_2)\Delta t$) and counted the *Empirical Event Counts* (actual simulated divisions falling in that bin).
>
> **Result:** The Pearson correlation between theoretical and empirical counts across the 100 bins is **0.9950**, with a relative error of **10.7%** (Fig.S6). This rigorously proves USB precisely captures state-dependent discrete dynamics at single-cell resolution.
>
> **2. Macroscopic: Branch Occupancy Ratios (Dyngen Dataset)**
> To evaluate global fate calibration on unbalanced bifurcations, we clustered real terminal states ($T=4$) into two branches via K-means to establish ground-truth occupancy. We then applied this classifier to the terminal states of 15,700 USB-simulated branching trajectories.
>
> **Result:** The ground-truth ratio is **0.708 : 0.292**, while USB predicts **0.650 : 0.350** (Fig.S7). The accumulation of microscopic discrete events in USB successfully calibrates macroscopic bifurcation structures, faithfully reproducing true biological dynamics.
>
> **Q4.** Due to the strict character limit, and as a similar concern was raised by another reviewer, please see **Q2 from Reviewer Ubh2**.

---

> > ### Author Rebuttal · Reviewer_eJXQ · 2026-04-02
> >
> > I appreciate the authors' detailed rebuttal. I will increase my score.

---

> > > ### Author Response · Authors · 2026-04-04
> > >
> > > We sincerely thank the reviewer again for their thorough review of our manuscript and for providing such constructive suggestions and positive comments. We will incorporate the experiments and clarify the exact and unexact part of the algorithm in the revised version. We are grateful to the reviewer for helping us improve this work.

---

### Official Review · Reviewer_8aQ1 · 2026-03-13

**Soundness:** 3
**Presentation:** 3
**Significance:** 3
**Originality:** 4
**Overall Recommendation:** 5
**Confidence:** 3

**Summary:**

This paper introduces an unbalanced, stochastic, simulation-free framework for modeling single-cell trajectories. The main advantage of this framework over previous works is that it is able to model single cell birth-death dynamics via jumps. The frameworks does so by solving the Unbalanced Schrodinger Bridge problem which replaces the standard reference stochastic process with branching Brownian motion.

**Compliance With Llm Reviewing Policy:**

Affirmed.

**Final Justification:**

My initial and final score are both 5. The rebuttal reinforced my prior assessment and the additional comments will be valuable additions to the paper in my opinion, especially the advice to practitioners. Overall I believe this work is novel and should be accepted.

**Key Questions For Authors:**

1. In the setup in section 3, in lines 119-130, the variable $m$ is not defined. Is $m$ meant to be the same as the time dependent weight $w_t(\mathbf{x}_t)$ from section 3.1 of [1]?
2. Should $\nu^2$ be instead $\nu$ in equation 9? That way it would match the form from [2] (Corollary 2.39,).

[1] Joint Velocity-Growth Flow Matching for Single-Cell Dynamics Modeling - https://arxiv.org/abs/2505.13413

[2] Regularized unbalanced optimal transport as entropy minimization with respect to branching Brownian motion - https://arxiv.org/abs/2111.01666

**Limitations:**

Yes

**Strengths And Weaknesses:**

Strengths:
- The paper is well written and clearly explains the previous related works leading up to its contribution.
- The approach in solving the Unbalanced Schrodinger Bridge problem is novel.
- The proposed framework unifies the important aspects of simulation-freeness, stochasticity, and unbalanced dynamical modelling for single cell trajectory inference. The branching inference procedure should become a useful analysis tool for practitioners.

Weaknesses:
- Given that one of the main contributions of this paper is the ability to model cell apoptosis and division, additional analysis into the capabilities of the framework and advice for practitioners would be useful.

Typos:
- Line 119 - “sthocastic” -> “stochastic”
- Line 132 - “the algorithm is consist of” -> “the algorithm consists of”
- Line 155 - in the definition of the semi-coupling there seems to be missing brackets
- Line 217 - “unnomarlized” -> “unnormalized”
- Lines 350-351 - “For convenient” -> “For convenience”
- Line 333, Table 2 - “dearth” -> “death”
- Line 357 - “mathces” -> “match”

---

> ### Author Rebuttal · Authors · 2026-03-29
>
> We sincerely thank the reviewer for carefully reviewing our manuscript and providing constructive comments. Below is our detailed response to key questions and weaknesses.
>
> **Q1.** We apologize for the missing definition. You are absolutely correct; the term $m_t$ shares the exact same meaning as the $\omega_t$ in [1]. We will add a definition in the revised manuscript. Generally, the evolution of a probability distribution can be modeled by the motion of a population of particles within a velocity field. By assigning a time-evolving mass to each particle, we can model the evolution of a positive measure. That's why there is $m_t$.
>
> **Q2.** Thank you for carefully checking the mathematical details! The notation $\nu$ in our paper carries a slightly different meaning than in [2]. In [2], the variance of the Brownian motion at time $t$ is $\nu t$, whereas in our paper, it is $\nu^2 t$ (see Line 179, where the reference process is defined as $\nu$ multiplied by the standard Brownian motion). This notational difference is the reason the equations appear distinct. To see this more clearly, when there is no growth (i.e., $r=0$), Equation 9 should reduce to the standard Fokker-Planck equation, hence the $\nu^2$ term.
>
> **Typos.** Thank you for point out the typos, we have thoroughly proofread the full text and corrected all identified spelling errors.
>
> **Additional analysis.**
> Following the advices from all reviewers, we have done some additional analysis on USB.
> * **1. Capability of Discrete Modeling:** We rigorously validated USB's branching inference. On a 400-step simulation, the Pearson correlation between our predicted discrete division event counts and the theoretical expected counts reached **0.9950**. On the Dyngen dataset, USB accurately reconstructed the macroscopic branch occupancy ratio (Predicted 0.650:0.350 vs. Ground-truth 0.708:0.292). For details, please see **Q3 from reviewer eJXQ**.
> * **2. Scalability and Memory Efficiency:** We benchmarked USB against DeepRUOT, SF2M, WFR-FM on up to 49302 cells. USB processed 50k cells in **~70 seconds** and requires only **~520 MB of GPU VRAM**, making it highly scalable. For details, please see **Q2 from reviewer UBh2**.
> * **3. Validity of WFR Approximation:** We analyzed the cost landscapes of our WFR approximation against the exact RUOT on 1D and 2D toy examples. The results confirm that WFR tightly matches the RUOT as long as the unbalance effects between different timepoints are not violently drastic (unbalance factor in $0.7\sim2$). For details, please see **Q2 from reviewer eJXQ**.
>
> **Advice for Practitioners** Based on the analyses above, we provide the following practical advice for future users of USB. We will write a new appendix G for these advice, and move the old appendix G to appendix H in the revised manuscript.
> * **Hardware Setup:** Due to our simulation-free continuous measure flow design, practitioners do not need expensive multi-GPU clusters. A standard workstation with a moderate CPU (for the $\sim$26GB global coupling matrix) and a basic entry-level GPU (since VRAM cost is $<1$ GB; you can also set larger batch_size to use more VRAM) is completely sufficient to train USB on tens of thousands of cells. For smaller dataset, you can run USB on your laptop.
> * **Data Suitability:** Practitioners can safely apply USB's WFR approximation to standard single-cell RNA-seq datasets. Since biological mass changes (proliferation/apoptosis) are typically gradual between observed time points, the moderate unbalance assumption always holds.
> * **Growth penalization $\delta$:** For users with biological background, if one wants to explicitly inject biological prior to growth, please tune $\delta$. A large $\delta$ means small growth prior.
> * **Branching inference:** When simulating trajectories, we use one-hot offspring distribution (restricting to binary split or death) as a default. However, practitioners with specific domain knowledge can flexibly adjust the distribution to simulate different underlying discrete biological events l (e.g. when there is a constant apoptosis rate) without retraining the model.

---

> > ### Author Rebuttal · Reviewer_8aQ1 · 2026-04-03
> >
> > Thank you for the response. I maintain my score and confidence as I still recommend acceptance. Please make sure to include these revisions in the final version of the paper, especially the advice for practitioners.

---

> > > ### Author Response · Authors · 2026-04-04
> > >
> > > We sincerely thank the reviewer again for their thorough review of our manuscript and for providing such constructive suggestions and positive comments. We will incorporate the corresponding modifications in the revised version. We are grateful to the reviewer for helping us improve this work.

---

### Decision · Program_Chairs · 2026-04-30

**Decision:**

Accept (regular)

**Comment:**

**Summary**

In this paper, the authors introduce a new generative modeling framework to take into account the unbalanced nature of some modeling. In particular, they focus on the reconstruction of Single-cell Snapshots. This allows to model non-conservative mass perturbations (cell proliferation, apoptosis). They generalize the concept of Diffusion Schrodinger Bridge to this unbalanced setting. While the unbalanced setting had been investigated before, they propose a novel point of view on the problem relying on Branching Brownian Motion. They solve the Schrodinger Bridge problem in two parts. First, a coupling is recovered by solving a Wasserstein-Fisher-Rao OT problem. Once the coupling is obtained, they consider a Poisson-Brownian bridge (contrary to classical Flow Matching which would consider a Brownian Bridge only) and solve the drift, score and mass variation parameters.

**Reviewer concerns**

All reviewers were convinced by the novelty of the approach. The experimental results are convincing. One of the main objection raised by the reviewers is the dependency on the WFR approximation "The paper’s central theoretical framing overstates the extent to which USB is an exact tractable solver for BSB: both the static BSB semi-coupling and its RUOT relaxation are intractable, and the implemented method ultimately relies on a WFR approximation. This makes the practical algorithm feel more like a heuristic BSB-inspired construction than a clean solver for the claimed target problem." (see Reviewer eJXQ). In order to overstate the claim, I would like the authors to include a broader discussion along the lines of "While these steps make USB an approximate solver, our main goal is not to solve the BSB problem itself, but to leverage it to capture complex single-cell dynamics. These approximations are necessary pragmatic trade-offs to ensure algorithmic efficiency." (as  stated in their rebuttal) in the final version of the paper.